# Flash-Searcher: Fast and Effective Web Agents via DAG-Based Parallel Execution

**Tianrui Qin**[1,2], **Qianben chen**[2], **Sinuo Wang**[2,3], **Xinyu Yang**[4], **Kang Zhu**[2],
**He Zhu**[2], **Dingfeng Shi**[2], **Xinxin Liu**[2,5,9], **Ge Zhang**[6], **Jiaheng Liu**[7],
**Xitong Gao**[5,8*], **Yuchen Eleanor Jiang**[1], **Wangchunshu Zhou**[2*]

[1]OPPO AI Center    [2]OPPO Reseach Institute    [3]The University of Adelaide
[4]Carnegie Mellon University    [5]Shenzhen Institutes of Advanced Technology
[6]ByteDance    [7]Nanjing University    [8]Shenzhen University of Advanced Technology
[9]Southern University of Science and Technology
{qintianrui,zhouwangchunshu}@oppo.com,xt.gao@siat.ac.cn

## Abstract

Large language models (LLMs) have demonstrated remarkable capabilities in complex agent reasoning tasks when equipped with external tools. However, current frameworks predominantly rely on sequential processing, leading to inefficient execution particularly for tasks requiring extensive tool interaction. This paper introduces Flash-Searcher, a novel parallel agent reasoning framework that fundamentally reimagines the execution paradigm from sequential chains to directed acyclic graphs (DAGs). Flash-Searcher decomposes complex tasks into subtasks with explicit dependencies, enabling concurrent execution of independent reasoning paths while maintaining logical constraints. Through dynamic workflow optimization, our framework continuously refines the execution graph based on intermediate results, effectively integrating summary module. Comprehensive evaluations across multiple benchmarks demonstrate that Flash-Searcher consistently outperforms existing approaches. Specifically, it achieves **67.7%** accuracy on BrowseComp and **83%** on xbench-DeepSearch, while reducing agent execution steps by up to **35%** compared to current frameworks. Furthermore, when distilling this parallel reasoning pipeline into single models, we observe substantial performance gains across diverse backbone architectures, underscoring the generalizability of our methodology. We propose a scalable and efficient paradigm for complex reasoning, advancing agent architecture design with our source code publicly available at https://github.com/OPPO-PersonalAI/Flash-Searcher.

## 1 Introduction

Recent advances in tool-augmented agents and multi-agent systems (MAS) (Dorri et al., 2018; Canese et al., 2021; Zhou et al., 2023c; 2024; Zhu et al., 2025a;b; Qiu et al., 2025; Roucher et al., 2025; Tang et al., 2025; Team, 2025) have demonstrated remarkable capabilities in complex problem-solving tasks, showcasing how collaborative agent frameworks can effectively address challenges requiring diverse reasoning abilities and tool manipulation. These systems leverage specialized agents with distinct roles, enabling sophisticated planning, reasoning, and tool utilization to solve tasks that would be challenging for single-agent approaches. Concurrently, research efforts have focused on Tool-Integrated Reasoning (TIR) (Jin et al., 2025a; Li et al., 2025c;d; Wu et al., 2025a; Sun et al., 2025; Zhang et al., 2025a; Zheng et al., 2025; Xue et al., 2025) approaches, which aim to incorporate the capabilities of tool execution or multi-agent systems into a single model through specialized training methodologies.

Despite their impressive performance, both MAS and TIR approaches face significant limitations when addressing general complex tasks. Multi-agent systems suffer from inefficient tool utilization, excessively long reasoning chains, and prolonged execution times due to sequential processing and redundant communication, while TIR methods encounter reasoning efficiency bottlenecks with chains

---

* Corresponding Author.

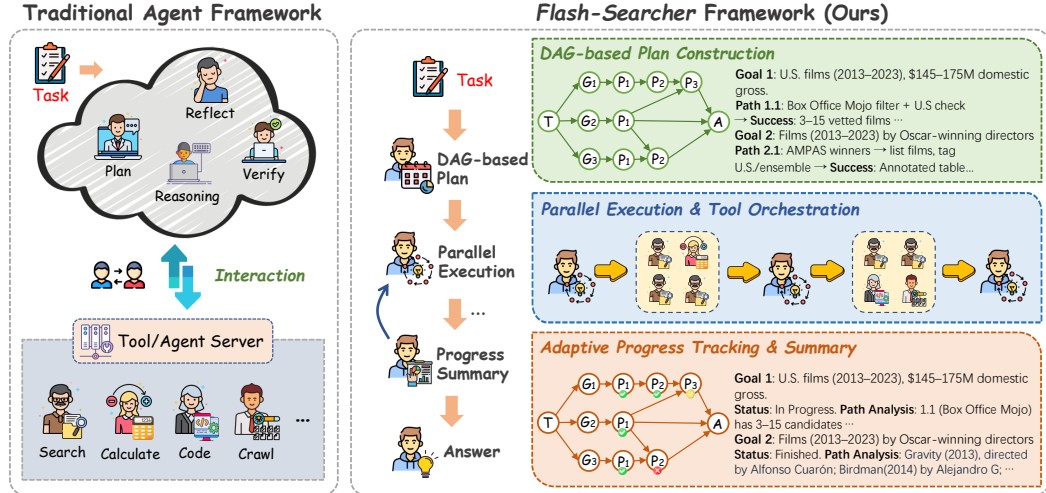

Figure 1: Overview of FLASH-SEARCHER: Framework and Key Components.

frequently exceeding context window limitations. These issues become even more pronounced in complex scenarios that require deep research capabilities. In such cases, MAS and TIR systems integrate additional verification mechanisms, such as reflection, self-critique, and iterative refinement, to enhance reliability. However, these improvements come at the cost of significantly increased computational overhead when solving complex tasks. Deep research tasks in current agent frameworks often require more than 20 interaction steps (Wang et al., 2025b; Roucher et al., 2025; Hu et al., 2025), with execution times extending to several hours. This creates a sharp tension between solution quality and computational efficiency, severely limiting practical viability in user-responsive applications. *When confronted with complex tasks inducing unavoidable latency, do users deem the better performance necessary enough to justify tolerating or paying for these delays?*

To address these critical challenges, we introduce FLASH-SEARCHER, a novel parallel agent reasoning framework that fundamentally reimagines how agents collaborate to solve complex tasks. Building upon recent empirical advances in reasoning models, our approach leverages these models' enhanced capabilities in simultaneously managing multiple cognitive threads. As illustrated in Figure 1, unlike traditional approaches that adhere to strict sequential processing, FLASH-SEARCHER decomposes the original task into multiple parallel execution paths, orchestrated via carefully designed agent workflows. This parallelization allows multiple reasoning paths to progress simultaneously while intelligently managing tool calls across different execution branches. The FLASH-SEARCHER framework redefines the efficiency-effectiveness frontier in complex task solving through key innovations: 1) adaptive decomposition and parallelization of tasks into concurrent subtasks with dynamic strategy adjustment, 2) dependency-aware reasoning graph management to model information dependencies and 3) optimize critical paths/information flow, and proactive information retrieval with knowledge sharing to anticipate downstream needs and reduce redundant interaction steps.

Our extensive evaluations demonstrate that FLASH-SEARCHER achieves state-of-the-art performance across multiple challenging benchmarks. Our FLASH-SEARCHER (with GPT-5-mini) reduces the average agent execution steps by **35%** (11.2 → 7.4 steps on GAIA) and shortens the overall execution time by ~**65%** (27.4 → 9.6 mins on BrowseComp) compared to OAgents (Zhu et al., 2025a). Despite this dramatic efficiency improvement, FLASH-SEARCHER (with GPT-5) achieves an impressive average performance of **82.5%** on GAIA benchmark. Furthermore, on more challenging benchmarks such as xbench, HLE and BrowseComp, FLASH-SEARCHER achieves performance metrics of **83.0**, **44.0** and **67.7** respectively, surpassing current state-of-the-art methods. Furthermore, to validate the generalizability of our approach, we constructed FLASH-SEARCHER execution trajectories based on collected web agent data and conducted post-training on the Qwen-2.5 family of open-source models. This lightweight adaptation achieves a performance score of **68.0** on the xbench-DeepSearch benchmark, representing a **29.3** improvement over WebDancer. This demonstrates the effective transfer of the parallel agent paradigm to open-source models with minimal additional training.

In summary, our contributions are as follows:

- We present a novel parallel agent reasoning framework that substantially reduces execution steps while achieving SOTA performance across various benchmarks.

- High-quality parallel reasoning trajectories, systematically curated and constructed for model post-training, significantly boost performance on complex evaluation tasks.

- Experimental results demonstrate the effectiveness of lightweight post-training in propagating parallel agent strategies to open-source models, achieving comparable results to multi-agent systems.

- We fully open-source pipeline and datasets of FLASH-SEARCHER to catalyze research on search agents and models.

## 2 RELATED WORK

### 2.1 MULTI-AGENT SYSTEM

Recent research has highlighted the effectiveness of multi-agent systems in addressing complex real-world challenges through collaborative agent frameworks. These systems typically employ multiple specialized agents with distinct roles, thereby supporting advanced planning, multi-turn reasoning, tool utilization, and environment interaction (Zhou et al., 2023c; 2024; Jin et al., 2025b; Zhu et al., 2025a;b; Mai et al., 2025; Hu et al., 2024; Tang et al., 2025; Shi et al., 2025; Tang et al., 2025; Zhou et al., 2023b). Early multi-agent systems such as CAMEL (Li et al., 2023) showed that dialog between agents can elicit stepwise reasoning through role-playing. Subsequent frameworks, including MetaGPT (Hong et al., 2024) and ChatDev (Qian et al., 2023), formalized this approach by implementing structured execution pipelines with dedicated roles such as manager, designer, and coder. Other approaches, like Magnetic-One (Fourney et al., 2024) and Smolagents (Roucher et al., 2025), incorporate a central planner that dynamically delegates subtasks to specialized tool-based agents. AgentVerse (Chen et al., 2023) refines collaborative reasoning via a recruitment–decision–execution–evaluation cycle, enhancing reflection and coordination. Workforce (Hu et al., 2025) decouples planning, coordination, and execution into modular agents, enabling efficient domain transfer through plug-and-play workers. Alita (Qiu et al., 2025) proposes autonomous tool exploration via iterative trial-and-error, expanding capabilities by transforming multi-attempt tasks into single-attempt ones. However, beyond performance, the latency in these complex multi-agent frameworks remains understudied.

### 2.2 EFFICIENT FRAMEWORK

Efficiency optimization has long been a pervasive focus in deep learning, pursued through various strategies ranging from model compression (e.g., knowledge distillation (Li et al., 2025h;f; Gou et al., 2021; Yang et al., 2025; Li et al., 2025b; Fang et al., 2026; Xu et al., 2024), structured pruning (Wang et al., 2019; Gao et al., 2018; Zhao et al., 2019; Liu et al., 2018; Ma et al., 2023; Li et al., 2025i;g)) to efficient data utilization via dataset distillation (Wang et al., 2018; Lei & Tao, 2023; Sun et al., 2024; Li et al., 2025e; Wang et al., 2025a). To address the efficiency bottlenecks inherent in existing agent frameworks, Tool-Integrated Reasoning (TIR) has recently emerged as a prominent research direction. Early efforts primarily adopted prompt-based strategies, such as Search-o1 (Li et al., 2025c), which employ static templates to instantiate fixed *Thought–Action–Observation* loops, thereby enabling rudimentary tool-augmented reasoning. More recent work has pivoted toward post-training paradigms (Jin et al., 2025a; Li et al., 2025d; Wu et al., 2025a; Li et al., 2025a; Tao et al., 2025; Sun et al., 2025; Xue et al., 2025; Li et al., 2025b; Nguyen et al., 2025), where agents are refined via task-specific fine-tuning to enhance performance. Despite their empirical gains, these approaches typically enforce narrowly scoped execution workflows, which severely limit their adaptability and scalability in open-domain, real-world environments. These challenges have motivated a broader effort to improve the efficiency and scalability of reasoning-enabled agents. Recent advances have focused on two key directions: optimizing agent pipelines and parallelizing search processes. Efficient Agents (Wang et al., 2025b) conducts a comprehensive analysis of core agent modules (workflow design, tool invocation, and memory architecture) to systematically balance performance and cost. Similarly, ParallelSearch (Zhao et al., 2025) trains models to detect parallelizable query structures, decomposing complex queries into independent sub-queries for retrieval tasks, resulting in significant performance gains in search-based tasks. However, existing systems remain constrained by

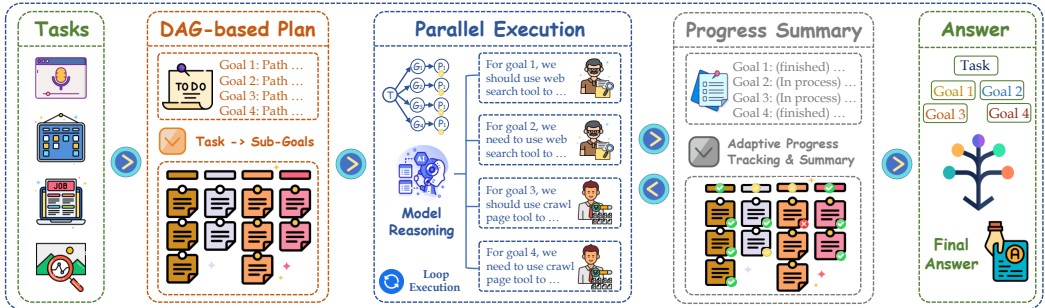

Figure 2: The pipeline of FLASH-SEARCHER.

isolated reasoning-execution loops or and the prolonged cycles introduced by multi-step verification, highlighting the need for more efficient approaches to agent.

# 3 METHOD

## 3.1 PRELIMINARIES

**Tool-Augmented Agents.** Tool-augmented agents enhance the capabilities of LLMs by seamlessly integrating external tools to perform actions such as information retrieval, mathematical computation, and code execution. This paradigm mitigates the inherent limitations of parametric knowledge through a structured tool-calling pipeline. Formally, the agent–environment interaction is modeled as a Markov decision process, wherein each tool invocation induces a state transition driven by environmental feedback. At timestep $t$, the agent selects a tool-calling action $a_t \in \mathcal{A}$, where $\mathcal{A}$ denotes the action space comprising available tools based on the current state $s_t$, and receives an observation $o_t \sim \mathcal{P}(\cdot \mid s_t, a_t)$ from the tool environment. The state transition function is defined as:

$$s_{t+1} = g(s_t, a_t, o_t),\tag{1}$$

where $g : \mathcal{S} \times \mathcal{A} \times O \to \mathcal{S}$ represents a state update function incorporating task, action history, and structured tool outputs into the new state representation $s_{t+1} \in \mathcal{S}$.

**Multi-Agent Systems.** Consider a set of agents indexed by $\mathcal{I}$, where each agent is denoted as $a_i$ for $i \in \mathcal{I}$. Each agent maintains a local state $s_t^i$ and possesses specialized capabilities. The global system state at time $t$ is defined as $S_t = \{s_t^1, s_t^2, \dots, s_t^n, c_t\}$, which aggregates all local states along with a shared context $c_t$. Agents coordinate through inter-agent communication protocols to optimize a common objective function $\mathcal{U}(S_t)$. The evolution of the global state follows:

$$S_{t+1} = f\left(\{a_t^i\}_{i \in \mathcal{I}}, S_t, O_t\right),\tag{2}$$

where $a_t^i \in \mathcal{A}$ denotes the action executed by agent $i$ at timestep $t$, $O_t = \{o_t^i\}_{i \in \mathcal{I}}$ represents the collection of observations from all agents, and $f$ integrates individual actions, the current global state, and observations to produce the next global state.

Existing approaches often adopt sequential execution with reflection and verification, prolonging task completion. Complex tasks may require 40+ interactions, introducing substantial latency. This sequential dependency creates a fundamental quality–efficiency trade-off, hindering real-world deployment.

## 3.2 FLASH-SEARCHER: PARALLEL AGENT REASONING FRAMEWORK

To overcome the inherent inefficiencies of sequential execution in conventional agent frameworks, we introduce FLASH-SEARCHER, a novel parallel reasoning framework that reformulates complex task solving as structured concurrency. Our approach transcodes the traditional linear workflow into a dynamic directed acyclic graph (DAG) plan, achieving substantial efficiency gains while preserving execution coherence. The full pipeline of FLASH-SEARCHER is illustrated in Figure 2.

**DAG-based Plan Construction.** Given a composite task $T$, FLASH-SEARCHER employs a decomposition function $\mathcal{D}$ that identifies constituent subtasks and their interdependencies, yielding a

DAG-based plan:

$$\mathcal{D}(T) = G_{\mathrm{plan}} = (V, E), \tag{3}$$

where $V = \{t_1, t_2, \ldots, t_n\}$ denotes subtasks and $E \subseteq V \times V$ captures prerequisite relations. Each directed edge $(t_i, t_j) \in E$ encodes that $t_i$ must precede $t_j$.

**Parallel Inferential Execution & Tool Orchestration.** At execution step $t$, FLASH-SEARCHER selects candidate subtasks from the pending set $\mathcal{P}_t \subseteq V$:

$$\mathcal{E}(G_t, \mathcal{P}_t) = \{v_i \in \mathcal{P}_t \mid \varphi(v_i, G_t, s_t) = 1\}, \tag{4}$$

where $\varphi(\cdot)$ is a readiness predicate. Unlike strict topological scheduling, $\varphi$ permits *aggressive parallelization*: a subtask $v_i$ may be scheduled if either (i) all its prerequisites are complete, or (ii) partial execution can provide auxiliary signals for dependency verification. Thus, $\varphi$ formalizes cross-validation as a hybrid criterion, blending dependency satisfaction and heuristic consistency checks. During execution, multiple subtasks $\mathcal{E}(G_t, \mathcal{P}_t)$ are processed in parallel via tool or agent invocations. The system integrates observations into the reasoning state:

$$s_{t+1} = \mathcal{F}\left(s_t, \{a_t^{(k)}\}_{k=1}^m, \{o_t^{(k)}\}_{k=1}^m\right), \tag{5}$$

where $a_t^{(k)}$ and $o_t^{(k)}$ denote the action and observation of the $k$-th parallel execution, and $\mathcal{F}$ integrates the results via structured aggregation and performs state transitions based on the aggregated information.

**Adaptive Progress Tracking & Summarization.** To reflect execution progress, FLASH-SEARCHER periodically updates the DAG-based plan every $\Delta$ steps:

$$G_{\mathrm{plan}}^{t+\Delta} = \mathcal{R}\left(G_{\mathrm{plan}}^t, C_t, \mathcal{P}_t, s_t\right), \tag{6}$$

where $C_t$ is the set of completed subtasks. The refinement rule $\mathcal{R}$ eliminates resolved nodes, revalidates unresolved dependencies based on cross-validation outcomes, and dynamically inserts new decomposition nodes if needed. The interval $\Delta$ can be flexibly specified: a smaller $\Delta$ increases the frequency of plan updates, ensuring faster task adaptation and responsiveness; a larger $\Delta$ suppresses excessive optimization, reducing computational overhead in complex or stable tasks.

By integrating DAG-based decomposition, controlled aggressive parallelization, and periodic DAG optimization, FLASH-SEARCHER mitigates the sequential bottleneck of existing reasoning architectures. This design provides a scalable and efficient alternative to sequential reasoning architectures, maintaining logical coherence through its structured, parallel approach. The full FLASH-SEARCHER pipeline is formally presented in Algorithm 1.

---

**Algorithm 1** FLASH-SEARCHER Framework

---

**Require:** Composite task $T$
 1: $G_{\mathrm{plan}} \leftarrow \mathcal{D}(T)$
 2: Initialize $s_0, \mathcal{P}_0 \leftarrow V, C_0 \leftarrow \emptyset$
 3: $t \leftarrow 0$
 4: **while** $\mathcal{P}_t \neq \emptyset$ **do**
 5: $\quad \mathcal{E}_t \leftarrow \{v \in \mathcal{P}_t \mid \varphi(v, G_t, s_t) = 1\}$
 6: $\quad$ Execute subtasks in $\mathcal{E}_t$ in parallel
 7: $\quad$ Collect results $\{o_t^{(k)}\}$ and update $s_{t+1} = \mathcal{F}(s_t, \{a_t^{(k)}\}, \{o_t^{(k)}\})$
 8: $\quad C_{t+1} \leftarrow C_t \cup$ completed subtasks
 9: $\quad \mathcal{P}_{t+1} \leftarrow \mathcal{P}_t \setminus C_{t+1}$
10: $\quad$ **if** $t \bmod \Delta = 0$ **then**
11: $\quad\quad G_{t+1} \leftarrow \mathcal{R}(G_t, C_{t+1}, \mathcal{P}_{t+1}, s_{t+1})$
12: $\quad$ **end if**
13: $\quad t \leftarrow t + 1$
14: **end while**
15: **return** Final state $s_T$

---

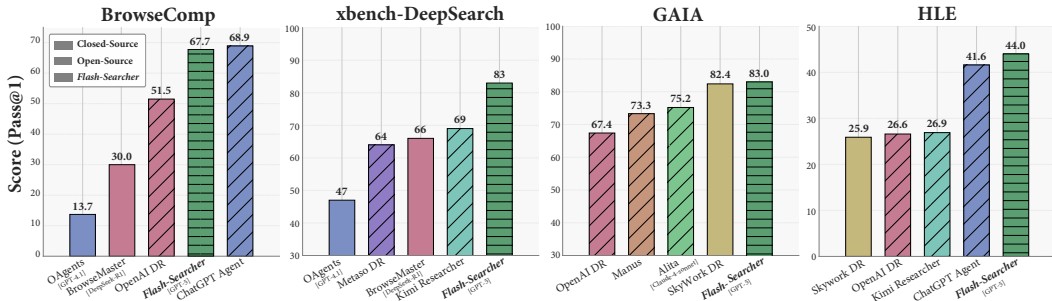

Figure 3: Performance comparison of agent frameworks on BrowseComp, xbench-DeepSearch, GAIA and HLE benchmarks. All results are reported using Pass@1 metric.

# 4 EXPERIMENT

## 4.1 FLASH-SEARCHER FRAMEWORK

### 4.1.1 SETUP

**Benchmarks.** We evaluate FLASH-SEARCHER on four challenging benchmarks for information retrieval and reasoning:

- **GAIA** (Mialon et al., 2023): A comprehensive benchmark for evaluating complex task-solving capabilities. For this benchmark, we mainly use the text-only validation set (103 tasks), which requires deep information retrieval and complex reasoning. Notably, the full validation set is solely used in Figure 3 for fair comparison; all other evaluations are based on the text-only validation subset.

- **BrowseComp** (Wei et al., 2025): Large-scale benchmark comprising 1,266 tasks designed to test internet-scale information retrieval with hard-to-find information needs and sophisticated browsing strategies.

- **xbench-DeepSearch** (Xbench-Team, 2025): Professional benchmark with 100 tasks simulating real-world search scenarios, emphasizing multi-round refinement and cross-source information integration.

- **HLE** (Phan et al., 2025): A frontier benchmark covering over a hundred subjects, designed to address the limited difficulty of existing benchmarks. We follow the setting in AFM (Li et al., 2025b) and use HLE-500 for evaluations.

**Framework Configuration.** FLASH-SEARCHER employs a minimalist yet powerful tool configuration optimized for parallel execution. Our framework integrates two core components: a Search Tool implemented with the Serper API (Serper, 2025) for retrieving structured search results, and a Crawl Tool leveraging the Jina Reader (Jina, 2025) for content extraction. The crawl tool incorporates automatic summarization using the same backbone language model, ensuring consistent information representation while significantly reducing cognitive load. This streamlined design enables efficient parallel tool orchestration across reasoning branches while maintaining trajectory simplicity and operational coherence. More details can be found in Section C.

**Metrics.** We employ the LLM-as-Judge paradigm (Zheng et al., 2023; Wu et al., 2025a) for automated evaluation, utilizing GPT-4.1-mini as the judge model. Each agent output of different benchmarks receives a binary correctness assessment from the judge model. We report Pass@1 results, which measure the proportion of tasks solved correctly on the first attempt, based on these binary correctness scores. The standardized prompt for judgment is detailed in Section G.1. The standardized prompt for judgment is detailed in Section G.1.

### 4.1.2 MAIN RESULTS

We present a comprehensive evaluation of FLASH-SEARCHER against state-of-the-art closed-source and open-source agent frameworks across four challenging benchmarks: BrowseComp, xbench-DeepSearch, GAIA, and HLE. As illustrated in Figure 3, our method achieves highly competitive

performance, matching or exceeding existing approaches while demonstrating superior efficiency and scalability. These results underscore the effectiveness of our DAG-based architecture in handling diverse task complexities.

In Figure 4, our FLASH-SEARCHER when integrated with GPT-5 achieves a competitive performance of **67.7%** on the BrowseComp benchmark. This result not only demonstrates a substantial advantage over state-of-the-art open-source frameworks (e.g., BrowseMaster (Pang et al., 2025), which attains 30.0%) but also approaches the performance of the leading closed-source solution, specifically the OpenAI ChatGPT agent (68.9%). Even with less powerful backbone models such as GPT-5-mini, our framework achieves **35.3%**, demonstrating the effectiveness of our parallel reasoning approach regardless of the underlying model. For xbench-DeepSearch, FLASH-SEARCHER also shows remarkable performance, with our GPT-5 variant achieving **83%**, surpassing both Browse-Master (66%) and Metaso DeepResearch (64%). This

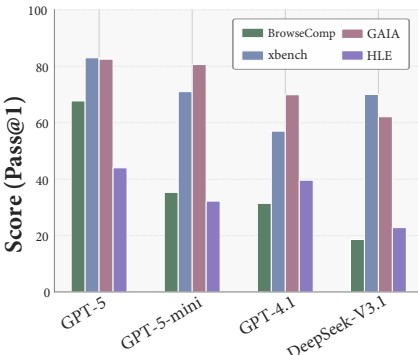

Figure 4: Performance of FLASH-SEARCHER with different backbones.

substantial improvement highlights the particular strength of our approach in deep research scenarios that demand extensive information gathering and complex reasoning. Besides, On the GAIA benchmark, FLASH-SEARCHER with lightweight, resource-efficient GPT-5-mini backbone achieves **80.6%**, exceeding even strong closed-source systems like Alita (75.2%) and Manus (73.3%). Additionally, our method demonstrates exceptional capability on the HLE benchmark, achieving a state-of-the-art **44.0%** with GPT-5, substantially outperforming all other frameworks. These results demonstrate that our parallel reasoning framework effectively handles diverse information retrieval challenges. The framework's consistent performance across different backbone models validates the robustness of our approach.

## 4.2 FLASH-SEARCHER LEARNING

### 4.2.1 SETUP

**Dataset.** To train our parallel reasoning agent, we construct a high-quality dataset derived from multiple sources including WebWalker (Wu et al., 2025b), ASearcher (Gao et al., 2025), WebShaper (Tao et al., 2025), and CoA (Li et al., 2025b). Our final dataset consists of **3354** effective DAG-based reasoning trajectories. Each trajectory incorporates periodic DAG workflow reviews and is formatted as a multi-turn dialogue, enabling effective context window extrapolation and long-range dependency modeling. This format specifically enhances the model's ability to manage complex reasoning graphs while maintaining coherent conversation flow. More details can be found in Section E.1

**Training Configurations.** We maintain consistent evaluation metrics and benchmarks with the framework experiments in Section 4.1.1. All training is implemented using the Llama-Factory framework (Zheng et al., 2024). We employ supervised fine-tuning to develop robust parallel reasoning capabilities. Specifically, for all trained models, the maximum dialogue length is set to 131,072 tokens, the learning rate is set to $10^{-5}$, and training is conducted for four epochs. The full training parameters and detailed data formatting specifications are comprehensively documented in Section E.2.

### 4.2.2 AGENT MODEL RESULTS

To validate the effectiveness of our parallel reasoning approach beyond framework implementation, we distilled FLASH-SEARCHER's parallel reasoning capabilities into standalone agent models through lightweight supervised fine-tuning. Table 1 presents a comprehensive comparison of these agent models against existing state-of-the-art methods across four challenging benchmarks.

Our experimental analysis demonstrates that lightweight supervised fine-tuning effectively facilitates the transfer of FLASH-SEARCHER's parallel reasoning capabilities to standalone agent models, consistently achieving state-of-the-art (SOTA) performance across diverse benchmarks and model backbone scales. Specifically, on the Qwen-2.5-32B backbone, FLASH-SEARCHER establishes a new

Table 1: Performance comparison of agent models on BrowseComp, xbench-DeepSearch, and GAIA benchmarks. All results are reported using Pass@1 metric. Gray-font values correspond to results reported in the associated reports.

| Method | Backbone | BrowseComp | xbench-DeepSearch | GAIA | HLE |
|---|---|---|---|---|---|
| Cognitive Kernel-Pro | Qwen-3-8B | - | - | 43.7 | - |
| WebDancer | | 3.8 | 39.0 | 50.5 | 7.2 |
| WebThinker-RL | | 2.8 | 24.0 | 48.5 | - |
| SimpleDeepSearcher | QwQ-32B | - | - | 50.5 | - |
| WebShaper | | - | - | 53.3 | 12.2 |
| SFR-DR | | - | - | 52.4 | 17.1 |
| WebDancer | | 2.5 | 38.7 | 40.7 | - |
| SimpleDeepSearcher | | - | - | 40.8 | - |
| WebShaper | Qwen-2.5-32B | - | - | 52.4 | - |
| WebSailor | | 10.5 | 53.3 | 53.2 | 10.8 |
| AFM-RL | | 11.1 | 58.0 | 55.3 | 18.0 |
| **FLASH-SEARCHER** | | **14.4** | **63.0** | **57.3** | **19.4** |
| WebSailor | | 12.0 | 55.0 | 55.4 | - |
| WebShaper | Qwen-2.5-72B | - | - | 60.1 | - |
| **FLASH-SEARCHER** | | **18.9** | **68.0** | **61.2** | **20.2** |

performance ceiling. It outperforms the strongest prior method by 3.3% on BrowseComp, 5.0% on xBench-DeepSearch, and 2.0% on GAIA. Despite forgoing code interpreter tools, FLASH-SEARCHER achieves state-of-the-art performance at 19.4% on HLE, surpassing tool-augmented baselines and affirming the general effectiveness of FLASH-SEARCHER in handling general complex tasks. This result underscores FLASH-SEARCHER's inherent reasoning robustness, as it delivers strong performance without relying on extensive tools.

Scaling FLASH-SEARCHER to 72B yields consistent and meaningful performance gains across all benchmarks, demonstrating that our parallel reasoning framework scales gracefully with model capacity. Notably, the most substantial improvements occur on complex, multi-step reasoning tasks such as BrowseComp and xbench-DeepSearch, with 5% gains, suggesting that increased parameter scale enhances the model's ability to coordinate and refine reasoning steps. Even on HLE, the performance affirms that FLASH-SEARCHER internalizes structured reasoning without relying on external tools. This behavior confirms that our lightweight fine-tuning paradigm not only transfers reasoning capabilities effectively but also unlocks deeper potential as backbone capacity grows, making it suited for scalable, general-purpose agent deployment.

Notably, these results are achieved through lightweight supervised fine-tuning without RL or tool reliance. This confirms that parallel reasoning is a learnable and scalable inductive bias, efficiently transferred via minimal supervision. FLASH-SEARCHER thus emerges as a simple, robust, and parameter-efficient solution for real-world agents.

## 5 EFFICIENCY ANALYSIS

We present a comprehensive efficiency analysis of FLASH-SEARCHER using the GPT-5-mini backbone, evaluating its execution efficiency and framework improvements compared to existing agent systems. The distribution plot in Figure 5a demonstrates BrowseComp benchmark requiring the highest number of both metrics. This reflects the varying complexity demands across different benchmark types. Figure 5b reveals FLASH-SEARCHER's operational efficiency through tool calls per execution step. The tight interquartile range, particularly evident in the GAIA benchmark, indicates consistent and predictable tool utilization patterns.These results support our core claim that the DAG-based architecture optimizes tool efficiency and reduces execution steps. By invoking complementary tools in parallel, our approach eliminates the sequential bottlenecks that cause redundant steps in linear pipelines.

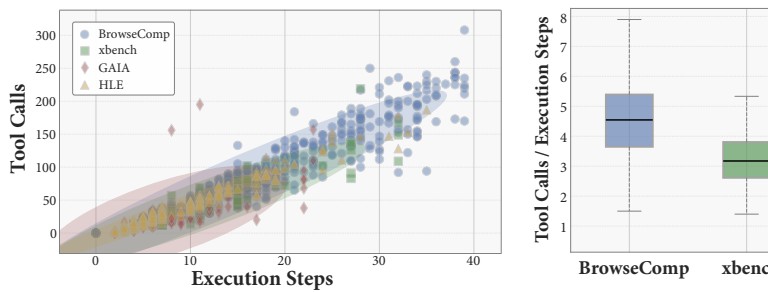
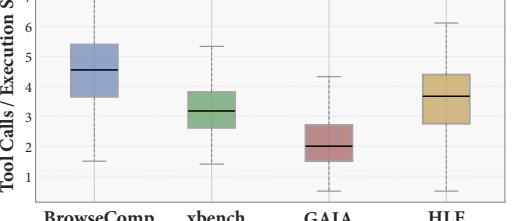

(a) Tool calls vs. execution steps.   (b) Tool calls per execution step of FLASH-SEARCHER.

Figure 5: Efficiency analysis of FLASH-SEARCHER on four benchmarks: (a) shows the correlation between tool calls and steps; (b) characterizes the distribution of tool calls per step.

To fairly evaluate the execution efficiency of FLASH-SEARCHER, we compare FLASH-SEARCHER against OAgents (Zhu et al., 2025a) and OWL-Roleplaying (Hu et al., 2025) with their original configurations (Details in Section F). The experimental results are presented in Figure 6, which demonstrates significant efficiency improvements of our approach across four benchmarks.

As shown in Figure 6a, FLASH-SEARCHER outperforms OAgents on all four benchmarks, achieving higher task success rates and efficiency gains, with this advantage growing more pronounced as task complexity increases (*BrowseComp ¿ xbench-DeepSearch ¿ HLE ¿ GAIA*). This validates FLASH-SEARCHER's adaptability to complex scenarios, laying the foundation for subsequent efficiency analysis. Figure 6b further demonstrates that FLASH-SEARCHER (with GPT-5-mini backbone) reduces agent steps by **35%** versus OAgents and **30%** versus OWL-Roleplaying on GAIA benchmark, enabled by its parallel reasoning architecture. This efficiency gain stems from the DAG-based workflow's ability to execute concurrent reasoning paths, which effectively mitigates the sequential bottleneck of traditional methods. Figure 5 illustrates the distribution of tool calls and steps for FLASH-SEARCHER: despite fewer total steps, our approach maintains higher per-step tool utilization efficiency (average **3.00** tool calls per step, compared to **0.83** for OAgents and **0.85** for OWL-Roleplaying), confirming more productive and effective reasoning iterations.

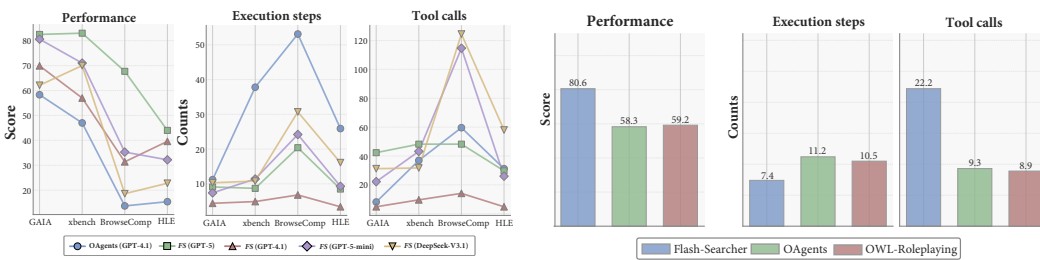

(a) Execution steps and tool calls comparison.   (b) Efficiency comparison of frameworks on GAIA.

Figure 6: Efficiency comparison of agent frameworks on four benchmarks.

The core innovation lies in our DAG-based parallel execution mechanism, which directly addresses the fundamental limitation of redundant tool invocation cycles in sequential reasoning approaches. By coordinating information needs across parallel branches, we eliminate duplicate searches while maintaining reasoning diversity. As in Figure 6a, our framework simultaneously enhances both efficiency and performance, effectively resolving the longstanding efficiency-effectiveness trade-off in agent systems.

Although agent execution duration is inherently influenced by external factors such as API rate limits, FLASH-SEARCHER consistently achieves a **35%** reduction in execution steps under comparable environmental conditions. This reduction directly translates into lower end-to-end latency and improved throughput, offering a significant efficiency advantage. This efficiency is critical for applications requiring low latency and high throughput, where sequential agents often face scalability bottlenecks. More detailed analysis of execution steps, time, and cost overheads can be found in Section D.3.

## 6 CONCLUSION

In this work, we introduce FLASH-SEARCHER, a novel parallel agent reasoning framework that overcomes the sequential bottlenecks of conventional tool-augmented agents through structured concurrency. By reformulating task solving as dynamic scheduling over DAGs, FLASH-SEARCHER enables fine-grained parallel execution while rigorously preserving logical coherence and correctness. Extensive experiments across BrowseComp, xbench-DeepResearch, GAIA, and HLE demonstrate that FLASH-SEARCHER achieves state-of-the-art performance, attaining a score of **67.7%** on BrowseComp, alongside substantial gains in computational efficiency through reduced latency and improved resource utilization. Our results, further corroborated by distilled agent variants, establish parallel reasoning as a foundational paradigm for building efficient, scalable, and robust AI systems capable of mastering complex real-world tasks.

## 7 ETHICS STATEMENT

The research focuses on the development of agent frameworks and model architectures for web-based autonomous agents, aiming to create more effective systems that can assist users in completing complex tasks. We conduct rigorous evaluation on controlled benchmarks and ensure transparency in our experimental procedures. This work does not involve any risks related to ethics issues and is intended to advance research in web agent systems.

## 8 REPRODUCIBILITY STATEMENT

We have made significant efforts to ensure the reproducibility of FLASH-SEARCHER. The code and data will be made publicly available at https://github.com/OPPO-PersonalAI/Flash-Searcher. Detailed experimental configurations, hyperparameters, and procedural specifications are documented in Section C. We note that exact reproduction may be challenging due to factors such as API updates, inference non-determinism, and reliance on closed source models, which could lead to minor variations in results.

## ACKNOWLEDGMENTS

We would like to express our sincere gratitude to the anonymous reviewers for their thoughtful and constructive feedback. Xitong Gao is supported by National Natural Science Foundation of China (62376263), Natural Science Foundation of Guangdong (2024A1515030209), and Shenzhen Science and Technology Innovation Commission (JCYJ20230807140507015).

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

## A  DISCUSSION OF DAG/GRAPH REASONING METHODS

### A.1  RELATIONSHIP TO GRAPH/TREE-STRUCTURED REASONING

While our work builds upon recent graph-based reasoning frameworks, key distinctions exist in purpose and implementation. Graph of Thoughts (GoT) (Besta et al., 2024) models reasoning steps as graph structures but emphasizes symbolic reasoning rather than tool execution. Tree of Thoughts (ToT) (Yao et al., 2023) explores branching reasoning paths through tree structures but prioritizes depth-first search over parallelization. Algorithm of Thoughts (AoT) (Sel et al., 2023) provides algorithmic guidance for reasoning, whereas our framework optimizes parallel tool execution specifically.

### A.2  DISTINCTIONS FROM PARALLEL REASONING AND PLANNING FRAMEWORKS

Recent works explore parallel reasoning and planning for LLM-based agents, yet differ significantly from our approach. Learning Adaptive Parallel Reasoning (LAPR) (Pan et al., 2025) introduces parallelization for language model reasoning but focuses primarily on model-internal computation rather than coordinating external tool calls. Plan-over-Graph (PoG) (Zhang et al., 2025b) shares conceptual similarities with our DAG approach but emphasizes strict dependency enforcement, whereas our framework intentionally relaxes these constraints to maximize parallel execution efficiency while ensuring result validity through cross-validation.

While Language Agent Tree Search (LATS) (Zhou et al., 2023a) and LLM+P (Liu et al., 2023) effectively integrate planning and acting through tree search and optimal planning techniques, they lack explicit mechanisms for parallel tool execution. Our framework complements these efforts by specializing DAG structures specifically for efficient information retrieval across multiple sources.

### A.3  COMPARISON OF VARIOUS FRAMEWORKS

In the context of DAG-based reasoning, several related frameworks offer alternative approaches, each with distinct characteristics in terms of parallelism, dependency handling, and dynamic refinement:

Table 2: Comparison of DAG/Graph-Based Reasoning Methods.

| Method | Parallel Execution | Dependency Handling | Dynamic Refinement |
|---|---|---|---|
| FLASH-SEARCHER | Tool-level parallelism | Relaxed | Yes |
| ParallelSearch | Search parallelism | N/A | No |
| Plan-over-Graph (PoG) | Limited | Strict dependencies | No |
| LATS | No | Strict | No |
| Graph of Thoughts | No | Strict | No |

Our primary contribution is an efficient dynamic DAG-based planning framework that optimizes execution trajectories in real time. It enables two core capabilities: (**1**) **parallel tool invocation for faster computation**, and (**2**) **cross-validation across dependent subtasks to preserve result integrity, resolving the key efficiency-accuracy trade-off in complex workflows**. Beyond efficiency, the framework addresses a critical LLM limitation: context length constraints. By continuously summarizing intermediate states and refining paths via real-time outcomes, it ensures lengthy multi-step workflows remain tractable without losing information fidelity.

## B  RELAXED CONSTRAINTS DETAILS

**Auxiliary Signals.** In web-agent scenarios, we observe that certain prerequisite information can often be *anticipated* before it is formally produced by upstream sub-goal. To exploit this property, our framework issues an *auxiliary early retrieval* whenever a subtask requires information that is not yet available. This early retrieval allows subsequent sub-goal to proceed under relaxed dependency constraints, thereby reducing idle waiting time and improving overall execution efficiency.

**Heuristic Consistency Checks.** Once the true prerequisite becomes available, the agent performs a second retrieval to validate or revise the earlier auxiliary signal. This *dual-retrieval mechanism*

functions as a heuristic consistency check. If the validated result differs from the auxiliary one, downstream states dependent on the earlier signal are corrected. This design preserves correctness while enabling more aggressive parallelization.

**Failure Handling.** Failure handling is integrated into the *Adaptive Progress Tracking & Summarization* module. During execution, if the agent detects that its current dependency path is blocked, inconsistent, or no longer leads toward the task objective, the framework triggers a path update. All outdated intermediate progress associated with the invalid path is cleared, and a revised plan graph is reconstructed based on the latest state. This ensures forward progress even under noisy signals or partially incorrect auxiliary retrievals.

Below, we provide an algorithm of the relaxed-constraint execution process supported by auxiliary-signal validation and failure handling.

---

**Algorithm 2** Relaxed-Constraint Execution with Auxiliary-Signal Validation and Failure Handling

---

**Require:** Composite task $T$
1:   $G_{\text{plan}} \leftarrow \mathcal{D}(T)$
2:   Initialize state $s_0$; active subtasks $\mathcal{P}_0 \leftarrow V$; completed set $C_0 \leftarrow \emptyset$
3:   $t \leftarrow 0$
4:   **while** $\mathcal{P}_t \neq \emptyset$ **do**
5:      $\mathcal{E}_t \leftarrow \{v \in \mathcal{P}_t \mid \varphi(v, G_t, s_t) = 1\}$ {**Eligible subtasks**}
6:      For each $v \in \mathcal{E}_t$, check prerequisite availability
7:      **if** prerequisite missing **then**
8:         Issue early retrieval $r_v^{\text{aux}}$ {**Auxiliary signal**}
9:      **end if**
10:     Execute all subtasks in $\mathcal{E}_t$ in parallel and collect outputs $\{o_t^{(k)}\}$
11:     Update state $s_{t+1} = \mathcal{F}(s_t, \{a_t^{(k)}\}, \{o_t^{(k)}\})$
12:     $C_{t+1} \leftarrow C_t \cup$ completed subtasks
13:     $\mathcal{P}_{t+1} \leftarrow \mathcal{P}_t \setminus C_{t+1}$
14:     **for** each recently satisfied prerequisite **do**
15:        Issue validation retrieval $r_v^{\text{chk}}$ {**Heuristic consistency check**}
16:        **if** $r_v^{\text{chk}} \neq r_v^{\text{aux}}$ **then**
17:           Correct downstream state using $r_v^{\text{chk}}$
18:        **end if**
19:     **end for**
20:     **if** Inconsistency or blocked path detected **then**
21:        Clear outdated intermediate progress
22:        $G_{t+1} \leftarrow \mathcal{R}(G_t, C_{t+1}, \mathcal{P}_{t+1}, s_{t+1})$ {**Failure handling**}
23:     **end if**
24:     $t \leftarrow t + 1$
25:   **end while**
26:   **return** Final state $s_T$

---

## C  Experiment Details

### C.1  Benchmarks.

We evaluate the effectiveness of FLASH-SEARCHER on four challenging benchmarks that require sophisticated information retrieval and reasoning capabilities:

- **GAIA** (Mialon et al., 2023): As a milestone benchmark for General AI Assistants, it constructs real-world questions that necessitate fundamental capabilities including reasoning, multi-modality handling, web browsing, and tool-use proficiency. To ensure rigorous and comparable evaluation, we conduct experiments primarily on the text-only validation subset of GAIA, which consists of 103 carefully curated cases. This subset specifically highlights the challenges of disambiguating ambiguous queries and synthesizing multi-source information. Additionally, for fair comparison with existing works, we further evaluate on the full validation set (165 cases) of

GAIA. Following the framework of OAgents (Zhu et al., 2025a), we additionally integrate text, image, and audio tools into our evaluation pipeline to align with the multi-modality and tool-use design goals of the full validation set.

- **BrowseComp** (Wei et al., 2025): A rigorous benchmark comprising 1,266 questions designed to measure persistent web browsing capabilities for finding hard-to-find, entangled information. While avoiding challenges like long-form generation, it specifically tests an agent's ability to formulate effective queries, navigate search results, extract relevant information, and synthesize coherent answers through sophisticated browsing strategies.

- **xbench-DeepSearch** (Xbench-Team, 2025): A professionally curated benchmark focusing specifically on deep-search capabilities in Chinese contexts, featuring 100 expert-written questions requiring multi-round search refinement and cross-source integration. Designed to isolate and evaluate the Planning → Search → Reasoning → Summarization pipeline of agent systems.

- **HLE** (Phan et al., 2025): To address the saturation of existing benchmarks (e.g., MMLU (Hendrycks et al., 2020), where SOTA LLMs now exceed 90% accuracy), HLE is proposed as a benchmark of 2,500 highly difficult questions across dozens of subjects, serving as a "final" closed-ended test for broad academic capabilities. Developed by experts via multi-stage review (pre-filtering, graduate/ expert validation, public feedback), it is multi-modal (text-only/image-accompanied), supports automated verification (multiple-choice/exact-match), and its questions are original, lookup-resistant, and emphasize advanced math for deep reasoning. Following AFM's setup (Li et al., 2025b), we use the HLE500 subset to evaluate model performance on high-difficulty reasoning.

Together, these benchmarks allow for a comprehensive evaluation of our framework's efficiency and effectiveness across a variety of complex information retrieval tasks.

## C.2 TOOL CONFIGURATIONS.

To ensure streamlined and efficient agent workflows and models, FLASH-SEARCHER employs a minimalist but powerful tool configuration focused on maximizing information retrieval capabilities while maintaining trajectory simplicity:

For external tools, we deliberately constrain our framework to just two essential components:

- **Search Tool**: We implement this tool using the Serper API (Serper, 2025) to support agents in retrieving web-based information for knowledge-intensive tasks. By default, each API call returns 5 relevance-ranked results, structured to include core elements: descriptive titles (for rapid relevance screening), concise content snippets (to pre-assess information utility), and direct URLs (for deep exploration of primary sources). This configuration strikes a balance between comprehensiveness, ensuring access to high-value sources, and computational efficiency, avoiding information overload that could hinder agent decision-making.

- **Crawl Tool**: Implemented using the Jina Reader (Jina, 2025), this tool enables agents to extract and process content from specific web pages. To enhance efficiency and maintain trajectory conciseness, our crawl tool incorporates an automatic summarization mechanism that extracts and condenses the most relevant information from web pages. Specifically, considering the constraints of model context window length and the cost control of API calls in large-scale experiments, we introduce a content truncation strategy for web pages: only the first 60,000 characters of each web page are selected as the input for the summarization mechanism to perform information extraction and condensation. This design balances the trade-off between information coverage and practical implementation costs, while it should be noted that the truncation may lead to the loss of potential valuable information in the latter part of long web pages. This approach significantly reduces cognitive load on the agent by eliminating the need to straightly process extensive raw HTML content.

The summarization component within the crawl tool utilizes the same language model as our backbone agent, ensuring consistency in understanding and representation across the framework. This architectural decision not only streamlines the information flow but also reduces potential misalignments between different components of the system.

By adopting this focused tool configuration, FLASH-SEARCHER achieves a balance between capability and efficiency. The framework provides agents with sufficient tools to tackle complex information retrieval tasks while avoiding the overhead and complexity associated with managing numerous specialized tools. This approach is particularly advantageous in our parallel execution context, where multiple tool calls can be orchestrated simultaneously across different branches of the reasoning graph.

## C.3 MODEL LIST.

In our experiments, we employed a diverse set of state-of-the-art LLMs. The evaluated LLMs include GPT-5 (Reasoning effort: medium; version: 2025-08-07), GPT-5-mini (Reasoning effort: medium; version: 2025-08-07), GPT-4.1, DeepSeek-v3.1 (w/o thinking), and GLM-4.5 (Default). In all experiments, we maintained consistent hyperparameters across comparable settings, with temperature set to 1.0. All models were accessed via reliable API endpoints with consistent system prompts to ensure fair comparison.

## C.4 PARAMETERS OF FLASH-SEARCHER.

To ensure the reproducibility and clarity of the FLASH-SEARCHER framework's implementation, this section details all key hyperparameters and configuration settings used in its execution. These parameters collectively govern critical behaviors of the framework, such as the scope of concurrent optimization objectives, the granularity of step-wise task execution, the constraints on tool utilization, and the rules for progress tracking and information retrieval. Specific configurations are summarized in Table 3 below.

Table 3: Parameter configurations for FLASH-SEARCHER Framework.

| Parameter | Description | Value |
|---|---|---|
| Parallel goals | Number of concurrent objectives | 5 |
| Goal path length | Predefined steps per goal | 5 |
| Max tool calls per step | Maximum tool invocations per step | 5/10 |
| Max steps | Total step budget for task execution | 40 |
| Summary interval | Steps between progress summaries | 7–9 |
| Search retrievals per query | Results returned per search call | 5 |
| Max length of extracted content | Max characters extracted by crawl_page | 60,000 |

In addition to the framework-level execution parameters detailed above, the inference process of the FLASH-SEARCHER models, responsible for decision-making (e.g., goal prioritization, tool selection) and content generation (e.g. progress summarization, query formulation), relies on a set of critical model-specific inference parameters. These parameters directly influence the model's reasoning depth, output stability, and computational efficiency, and are tightly aligned with the framework's execution constraints (e.g. step budget, tool call limits) to ensure coherent end-to-end performance. To support efficient and scalable inference, we adopt the *vllm* framework (a high-throughput LLM serving framework optimized for GPU acceleration) and deploy the system on a hardware cluster consisting of 8 NVIDIA A800 GPUs. Specific inference configurations (model-specific) and hardware-framework settings are summarized in Table 4 and Table 5 below, respectively.

Table 4: Inference parameter configurations for FLASH-SEARCHER Models.

| Parameter | Description | Value |
|---|---|---|
| Context length | Maximum context tokens | 131072 (32B) / 65536 (72B) |
| Max steps | Total conversation length | 40 |
| Max output tokens per call | Maximum generated tokens per inference step | 8192 |
| Temperature, top-k, top-p | Probabilistic generation controls | Default |

Other inference tool parameters are designed to maintain consistency with the framework's execution settings. Meanwhile, the inference configurations are tailored to match the model size (32B/72B parameters) and context length requirements, avoiding memory bottlenecks during long-sequence reasoning. These cross-parameter alignments are critical for avoiding misalignment between the

model's reasoning process and the execution environment, thereby ensuring reproducibility and stability of the FLASH-SEARCHER system's performance across different task instances.

Table 5: Configurations for FLASH-SEARCHER inference.

| Configuration | Description | Value |
|---|---|---|
| Inference framework | Serving framework for inference | vllm v0.10.1.1 |
| GPU type | Hardware accelerator model | NVIDIA A800 (80GB) |
| Tensor parallelism | GPU partitioning strategy | 8 |
| RoPE scaling | Extending context length | Dynamic (factor=4.0 for 32B; factor=2.0 for 72B) |
| Model of Crawl Tool | Model for crawling data summary | GPT-5-mini |

## C.5 DETAILED RESULTS OF FLASH-SEARCHER.

To comprehensively evaluate the effectiveness of the proposed FLASH-SEARCHER framework, we conduct extensive experiments on four representative benchmarks for agent systems. The performance is quantified using the widely adopted Pass@1 metric, which measures the proportion of tasks successfully completed by the agent in a single attempt. Table 6 and Figure 7 present the detailed performance comparison between FLASH-SEARCHER and existing state-of-the-art agent frameworks or models. For fairness and reference, values displayed in gray font are directly quoted from the original reports of the compared methods.

Table 6: Performance comparison of agent frameworks on BrowseComp, xbench-DeepSearch, and GAIA benchmarks. All results are reported using Pass@1 metric. Gray-font values correspond to results reported in the associated reports. Note that FLASH-SEARCHER achieve **83.0** for full validation set.

| Method | Backbone | BrowseComp | xbench-DeepSearch | GAIA | HLE |
|---|---|---|---|---|---|
| *Closed-Source Frameworks* | | | | | |
| OpenAI ChatGPT agent | - | 68.9 | - | | 41.6 |
| OpenAI DeepResearch | - | 51.5 | - | 67.4 | 26.6 |
| Metaso DeepResearch | MetaLLM *etc.* | 12.0 | 64 | - | - |
| Skywork DeepResearch | Claude-Sonnet-3-7 *etc.* | - | - | 82.4 | 25.9 |
| Kimi Researcher | Kimi k-series *etc.* | - | 69 | - | 26.9 |
| Manus | Claude *etc.* | - | - | 73.3 | - |
| Alita | Claude-Sonnet-4 | - | - | 75.2 | - |
| *Open-Source Frameworks* | | | | | |
| Smolagents | OpenAI-o1 | - | - | 49.7 | - |
| A-World | Gemini-2.5-Pro | - | - | 71.0 | - |
| Cognitive Kernel-Pro | Claude-Sonnet-3-7 | - | - | 66.1 | - |
| OWL–Workforce | Claude-Sonnet-3-7 | - | - | 69.7 | - |
| OAgents | GPT-4.1 | 13.7 | 47 | 58.3 | 15.4 |
| BrowseMaster | DeepSeek-R1-0528 | 30.0 | 66 | 68.0 | - |
| MiroFlow | GPT-5 | 33.2 | 72 | 82.4 | 29.5 |
| **FLASH-SEARCHER** | GPT-5 | **67.7** | **83** | **82.5** | **44.0** |
| | GPT-5-mini | 35.3 | 71 | 80.6 | 32.2 |
| | GPT-4.1 | 31.4 | 57 | 69.9 | 39.6 |
| | DeepSeek-V3.1 | 18.6 | 70 | 62.1 | 22.8 |
| | GLM-4.5 | - | 63 | 63.1 | - |
| | GPT-5-nano | - | 61 | 54.4 | - |

## D ADDITIONAL ABLATIONS

### D.1 MODEL BACKBONES ABLATIONS

To isolate the contribution of the FLASH-SEARCHER framework from that of the underlying model backbone, we evaluated FLASH-SEARCHER using the same backbone across different agent frame-

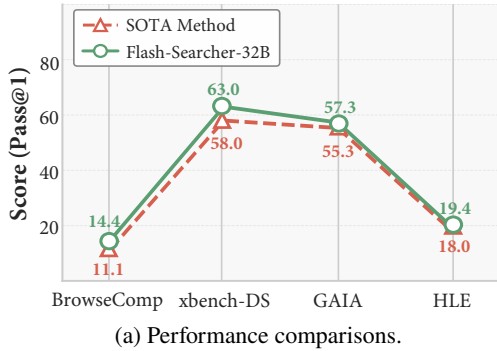 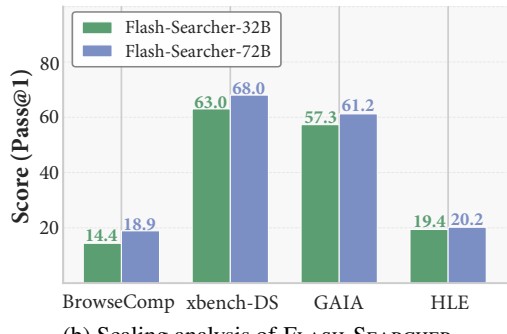

(a) Performance comparisons.    (b) Scaling analysis of FLASH-SEARCHER.

Figure 7: Performance and scaling analysis of FLASH-SEARCHER. (a): FLASH-SEARCHER-32B consistently outperforms the SOTA method across all four benchmarks with Qwen-2.5-32B. (b): Scaling FLASH-SEARCHER from 32B to 72B parameters yields consistent gains.

works. The evaluation was conducted using the xbench-Deepsearch and GAIA benchmarks, comparing FLASH-SEARCHER with other state-of-the-art agent frameworks. As shown in Table 7, FLASH-SEARCHER demonstrates improved efficiency in terms of both execution steps and time, outperforming or matching the performance of existing SOTA frameworks across both benchmarks.

Table 7: Performance comparison on xbench and GAIA benchmarks.

| Method | Backbone | xbench-DeepSearch | | GAIA | |
|---|---|---|---|---|---|
| | | Score | Step | Score | Step |
| OAgents | GPT-4.1 | 47 | 37.8 | 58.3 | 11.2 |
| FLASH-SEARCHER | | 57 | 5.9 | 69.9 | 4.4 |
| BrowseMaster | DeepSeek-R1-0528 | 66 | – | 68.0 | – |
| FLASH-SEARCHER | | 69 | 10.6 | 67.0 | 9.8 |

## D.2 SEQUENTIAL REACT-STYLE AGENT VS FLASH-SEARCHER

To further validate the benefits of DAG-based parallel execution, we conducted a controlled comparison between a sequential ReAct-style agent and the parallel FLASH-SEARCHER agent. Both agents were evaluated using the same backbone (GPT-5-mini), the same tools, and the same base prompts.

Table 8: Performance comparison between sequential ReAct-style agent and FLASH-SEARCHER.

| Method | BrowseComp-100 | | | xbench-DeepSearch | | | GAIA | | |
|---|---|---|---|---|---|---|---|---|---|
| | Score | Step | Time/mins | Score | Step | Time/mins | Score | Step | Time/mins |
| Sequential ReAct | 31 | 40.6 | 14.6 | 66 | 18.2 | 8.5 | 69.9 | 12.1 | 4.7 |
| FLASH-SEARCHER | **36** | **20.8** | **9.6** | **71** | **11.4** | **4.9** | **80.6** | **7.4** | **3.3** |

The results shown in Table 8 demonstrate that FLASH-SEARCHER significantly reduces both execution steps and time, especially for more complex tasks like BrowseComp-100 and xbench-DeepSearch. The parallel execution mechanism of FLASH-SEARCHER leads to substantial improvements in efficiency and accuracy compared to the sequential approach.

## D.3 DETAILED COST ANALYSIS

We present a detailed cost analysis to evaluate the performance and efficiency of FLASH-SEARCHER in comparison with existing agent systems. The analysis focuses on both time and computational costs across xbench-DeepSearch and GAIA benchmarks. Table 9 summarizes the step and time cost, where FLASH-SEARCHER consistently outperforms OAgents (Zhu et al., 2025a) and Sequential ReAct (Adapted from FLASH-SEARCHER) in terms of efficiency, despite slight increases in computational cost due to additional tool invocations.

Table 9: Comparison of step and time cost across different frameworks

| Method | xbench-DeepSearch | | GAIA | |
|---|---|---|---|---|
| | Step | Time/mins | Step | Time/mins |
| FLASH-SEARCHER | **11.4** | **4.9** | **7.4** | **3.3** |
| OAgents | 37.8 | 12.9 | 11.2 | 4.2 |

As shown in Table 10, we provide a detailed breakdown of token usage and associated costs for the xbench-DeepSearch and GAIA benchmarks. The results indicate that FLASH-SEARCHER significantly outperforms OAgents in terms of efficiency, with a lower cost per query across all configurations. This demonstrates the effectiveness of FLASH-SEARCHER in achieving high performance while minimizing computational overhead, making it a more cost-efficient option for large-scale agent-based tasks.

Table 10: Detailed comparison of cost overheads on xbench-DeepSearch and GAIA benchmarks.

| Method | Plan | Summ./Replan | Action | Crawl | Input Tok | Output Tok | Cost / Query |
|---|---|---|---|---|---|---|---|
| *xbench-DeepSearch* | | | | | | | |
| OAgents | 3823.4 | – | 714.6 | – | 58770.4 | 40514.6 | $0.13 |
| FLASH-SEARCHER | 1623.6 | 3916.7 | 406.1 | 205.3 | 28318.6 | 9716.8 | $0.07 |
| *GAIA* | | | | | | | |
| OAgents | 3014.6 | – | 631.9 | – | 17281.8 | 13137.2 | $0.05 |
| FLASH-SEARCHER | 1539.4 | 2469.4 | 352.4 | 209.6 | 10664.4 | 5685.9 | $0.03 |

### D.4 SUMMARY INTERVAL ANALYSIS

We conducted an ablation to analyze the impact of varying the update interval $\Delta$ on the framework's performance. As shown in Figure 8, a lower update interval improves the model's performance but results in longer execution times. The optimal interval depends on task difficulty and model context length, as shown by the varying results. The study shows that while reducing $\Delta$ can increase the model's accuracy, it also increases computational cost. We recommend adjusting the update interval based on task complexity to achieve the best balance between performance and efficiency.

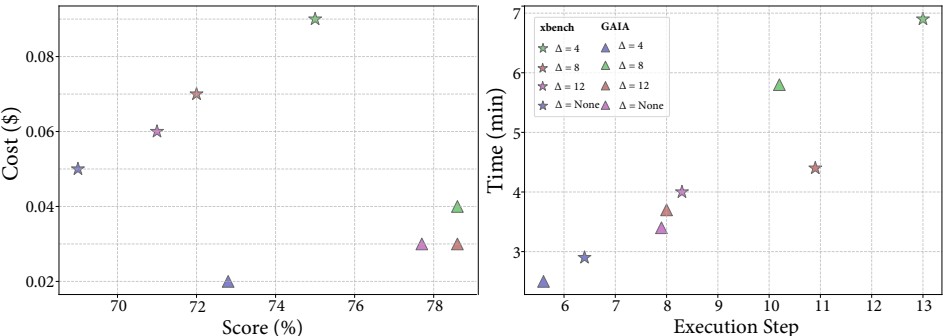

Figure 8: Update Interval $\Delta$ Analysis on xbench-DeepSearch and GAIA benchmarks.

### D.5 EVALUATIONS ON SIMPLE TASKS

We compare the performance of FLASH-SEARCHER on Bamboogle (Press et al., 2023) benchmark with a ReAct sequential agent. For simple tasks, we observe that FLASH-SEARCHER introduces some overhead, resulting in slightly higher execution times compared to sequential agents, as shown in Table 11. This overhead is due to the additional steps required for parallelization. Based on these findings, we recommend dynamically selecting the appropriate approach based on task complexity to optimize performance.

Table 11: Performance Comparison on Bamboogle.

| Model | Score | Step | Time/mins |
|-------|-------|------|-----------|
| ReAct sequential | 91.2 | 4.8 | 0.9 |
| FLASH-SEARCHER | 91.2 | 5.4 | 1.7 |

## D.6 ANALYSIS OF THE CODE TOOL INTEGRATION

We provide a detailed analysis of integrating the code execution tool within the FLASH-SEARCHER framework. Although code execution can be useful for computation-intensive tasks, our experiments indicate that its practical utility in web-agent scenarios is very limited. Across benchmarks such as GAIA, xBench, and HLE, the model rarely invoked the code tool even when it was enabled, reflecting the predominantly retrieval- and reasoning-oriented nature of web tasks. Moreover, enabling the code tool introduces non-trivial runtime overhead, primarily due to additional execution latency and increased output tokens. To ensure a fair comparison, we report the results in Table 12. The code-enabled variant of FLASH-SEARCHER exhibits only a marginal +1.4% improvement on the HLE benchmark, while incurring a substantial increase in execution time. This cost–benefit imbalance suggests that, for web-centered tasks, the marginal utility of integrating a code tool is low.

Table 12: Performance comparison of FLASH-SEARCHER w. & w/o code tool.

| Model | HLE-500 | | |
|-------|---------|------|-----------|
| | Score | Step | Time/mins |
| FLASH-SEARCHER | 32.2 | **9.0** | **4.0** |
| FLASH-SEARCHER + Code Tool | **33.6** | 9.4 | 5.9 |

Based on the findings, while code execution can extend the versatility of FLASH-SEARCHER for certain task types, our analysis shows that its contribution to web-agent performance is minimal relative to its runtime cost. Consequently, the default configuration omits the code tool to ensure maximal efficiency.

## D.7 ABLATIONS ON EXECUTION STEP CONSTRAINTS

To validate the observation that execution step limitations constrain the resolution of complex queries, we conduct ablations on the FLASH-SEARCHER models, focusing on the impact of extended maximum reasoning steps. Specifically, we evaluate model performance on BrowseComp-100 (a subset of BrowseComp) under an extended maximum step limit of 80.

Figure 9 summarizes the performance of FLASH-SEARCHER model variants under the two step limits. Across all FLASH-SEARCHER models, extending the maximum number of steps from 40 to 80 yields consistent and measurable performance gains: FLASH-SEARCHER-32B improves by 5.0 points, while FLASH-SEARCHER-72B achieves a 7.0-point increase. These results confirm that the performance bottleneck observed in the 40-step setting arises from insufficient reasoning steps rather than fundamental model limitations.

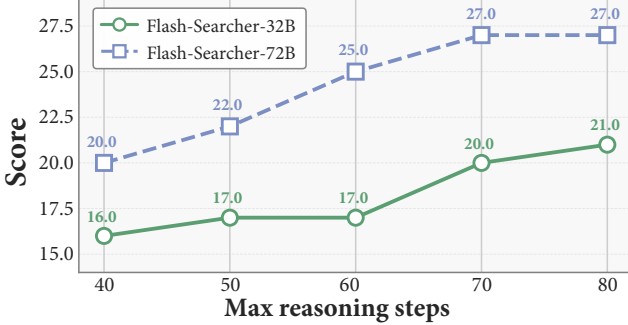

Figure 9: Performance of FLASH-SEARCHER models under different reasoning step constraints.

These results directly verify our initial observation: increasing the number of allowed reasoning steps enables FLASH-SEARCHER to fully unpack complex task logic, thereby improving solution accuracy. This supports the feasibility of trading computational resources for precision in resource-unconstrained deployment scenarios.

# E FLASH-SEARCHER MODEL TRAINING

## E.1 TRAINING DATASET

Our training dataset is constructed by curating subsets of four well-established public agent-focused datasets: AFM (Li et al., 2025b)[1], ASearcher (Gao et al., 2025)[2], WebShaper (Tao et al., 2025)[3], and WebWalkerQA (Wu et al., 2025b)[4]. The subsets contain 1355, 628, 500, and 2597 examples, respectively. For AFM, Asearch, and WebWalkerQA-silver datasets, we applied a filtering process based on the execution trajectory length of baseline ReAct frameworks. Specifically, we selected only those examples that required more than 8 steps to complete, as these represent more complex reasoning and action sequences that better demonstrate agent capabilities.

Table 13: Composition of the training dataset after filtering and trajectory generation.

| Dataset | Original Size | Correct Samples |
|---|---|---|
| AFM | 1,355 | 1212 |
| Asearch | 628 | 457 |
| WebShaper | 500 | 405 |
| WebWalkerQA | 2,597 | 1767 |
| **Total** | 5080 | **3354** (Removed data with formatting issues) |

We leveraged our FLASH-SEARCHER framework (with GPT-5 as the backbone) to generate trajectories for pre-filtered examples. To ensure training data reliability—critical for effective model learning—we used a judge model (GPT-4.1-mini) to validate trajectory answer correctness, retaining only factually accurate ones. This initial filtering yielded 1212, 457, 405, and 1767 candidate trajectories from AFM, ASearcher, WebShaper, and WebWalkerQA, respectively. We further conducted systematic format inspections to exclude trajectories with structural flaws (e.g., incomplete turn segmentation, invalid dialogue hierarchy, missing action labels), a step to reduce noise in supervised fine-tuning (SFT). After this two-stage screening (correctness + format), we ultimately retained **3354 valid trajectories** for training. Table 13 summarizes the final training dataset composition, including source dataset and trajectory attribute breakdowns.

These trajectories were formatted into SFT-compatible multi-turn dialogues via the LLaMA-Factory framework (Zheng et al., 2024). Specifically, the detailed structure of the multi-turn dialogue format (including role definitions, dialogue turn segmentation, and task-related context embedding) is illustrated in Figure 11, which standardizes the conversion of trajectory data into instruction-response pairs for SFT training.

## E.2 PARAMETERS

We performed supervised fine-tuning (SFT) using the LLaMA-Factory framework with selected hyperparameters to optimize model performance. Table 14 presents the key parameters used during our training process. We employed a cosine learning rate schedule with warmup to stabilize the early training phase. To address memory constraints while training on the 32B/72B parameter model, we utilized gradient accumulation and parameter-efficient fine-tuning techniques. The training was conducted on 64 NVIDIA A800 GPUs (80GB each) with DeepSpeed ZeRO-3 optimization to manage memory usage efficiently.

---

[1] AFM Dataset: https://huggingface.co/datasets/PersonalAILab/AFM-WebAgent-SFT-Dataset

[2] ASearcher Dataset: https://huggingface.co/datasets/inclusionAI/ASearcher-train-data

[3] WebShaper Dataset: https://github.com/Alibaba-NLP/WebAgent/blob/main/WebShaper

[4] WebWalkerQA Datasethttps://huggingface.co/datasets/callanwu/WebWalkerQA

Table 14: Training hyperparameters for supervised fine-tuning.

| Parameter | Value |
|---|---|
| Learning Rate | 1e-5 |
| Training Epochs | 6 |
| Gradient Accumulation Steps | 2 |
| Warmup Ratio | 0.1 |
| Gradient Accumulation Steps | 2 |
| LR Scheduler | Cosine with Warmup |
| Context length | 131072 (for 32B) / 65536 (for 72B) |

### E.3 MODEL TRAINING CURVES

In this section, we provide detailed training curves for the FLASH-SEARCHER when applied to the Qwen-2.5-32B-Instruct and Qwen-2.5-72B-Instruct models. These curves illustrate the evolution of key metrics throughout the training process, validating the stability and convergence properties of our approach.

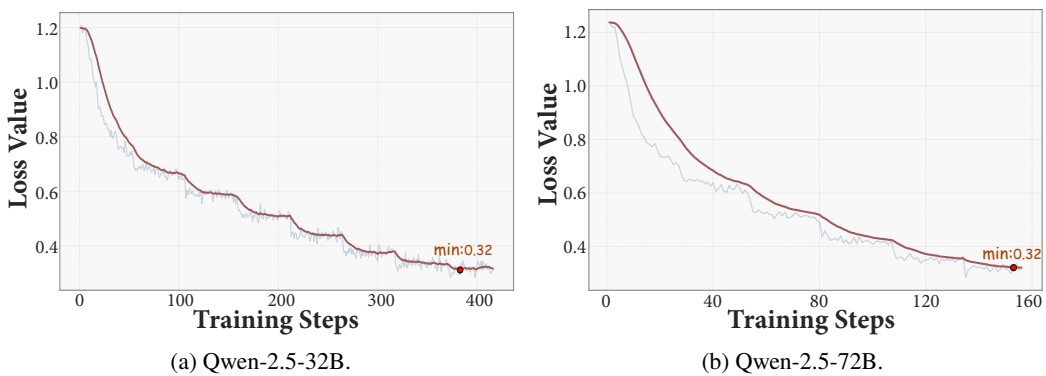

(a) Qwen-2.5-32B.       (b) Qwen-2.5-72B.

Figure 10: Training curves for FLASH-SEARCHER models. Both models demonstrate stable convergence without signs of overfitting.

### E.4 EXAMPLE OF TRAINING DATA

We present a multi-turn dialogue format examples for SFT training, explicitly illustrating the three core components of each dialogue unit: system prompt (task constraints), user instruction (task-specific requirement), and agent response (standardized output).

## F OTHER FRAMEWORK SETUPS

For our comparative analysis, we employ two state-of-the-art agent frameworks: OAgents (Zhu et al., 2025a) and OWL-Roleplaying (Hu et al., 2025). We maintain their original configurations to ensure fair comparison with our approach.

For OAgents, both the Code-Agent and Search-Agent components utilize GPT-4.1 as their backbone model. Similarly, OWL-Roleplaying is implemented with two backbone variants: GPT-4.1 and OpenAI-o3. All other parameters, prompting strategies, and execution workflows for both frameworks are kept identical to their original implementations. Our experiments are conducted using the official repositories [5] of these frameworks to ensure reproducibility and consistency with published results.

---

[5]OAgents: https://github.com/OPPO-PersonalAI/OAgents; OWL: https://github.com/camel-ai/owl

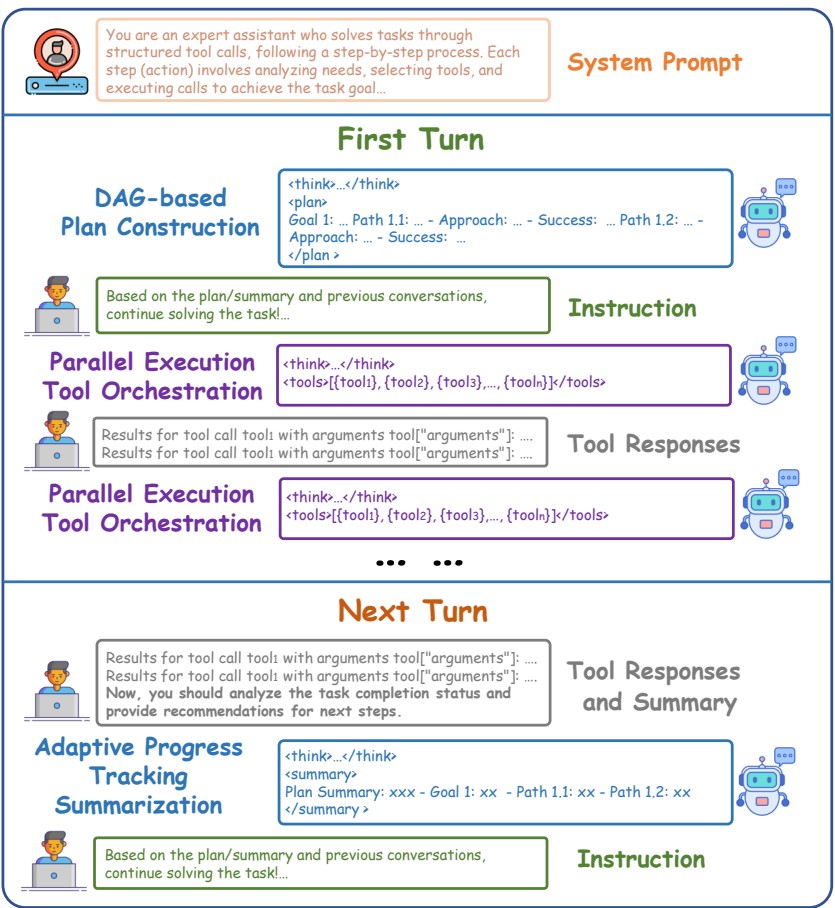

Figure 11: Example of the multi-turn dialogue format for SFT training. Each dialogue unit consists of three core components: (1) System prompt (task constraints), (2) User instruction (task-specific requirement), and (3) Agent response (standardized output).

# G PROMPTS

## G.1 LLM-AS-JUDGE PROMPT

> ⚖️ **LLM-AS-JUDGE PROMPT**
>
> Please determine if the predicted answer is equivalent to the labeled answer.
> Question: question
> Labeled Answer: gt_answer
> Predicted Answer: pred_answer
> Are these answers equivalent?
> The output should in the following json format:
> {{
>    "rationale": "your rationale for the judgement, as a text",
>    "judgement": "your judgement result, can only be 'correct' or 'incorrect'"
> }}

## G.2 FLASH-SEARCHER FRAMEWORK

### G.2.1 SYSTEM PROMPT

> ⚙️ **SYSTEM PROMPT**
>
> You are an expert assistant who solves tasks through structured tool calls, following a step-by-step process. Each step (action) involves analyzing needs, selecting tools, and executing calls to achieve the task goal. Each action you take should include a reasoning process and tool calls. After executing the tools, you will receive "observations" (results of tool calls), which can be used as input for subsequent actions. This Action/Observation cycle may repeat as needed.
>
> # Action Structure
>
> Each action must contain:
> - "think": A detailed reasoning in English, explaining the analysis of user needs, tool selection logic, and execution plan.
> - "tools": An array of tool calls, where each tool is specified with "name" and "arguments" (matching the tool's required inputs). Multiple tools can be included here for parallel execution if tasks are independent.
>
> # Task Instructions:
>
> ### 1. Parse the structured plan:
> Parse the plan or summary to understand the parallel execution requirements.
> **CRITICAL: All goals MUST be advanced simultaneously in parallel. Each goal's paths MUST be executed sequentially (one path at a time per goal).**
> ### 2. Execute parallel tool calls:
> For each goal in the plan, execute the specified tools in parallel according to the paths defined.
> **MANDATORY: Advance ALL goals concurrently. Within each goal, execute paths sequentially (never parallelize paths within a single goal).**
> ### 3. Handle path diversity:
> For each goal, if multiple paths are provided, execute them sequentially as fallback options if the primary path fails.
> **ABSOLUTE REQUIREMENT: NEVER prematurely assume a goal is achieved. Continue advancing ALL other goals in parallel while handling fallback paths for any individual goal.**
> ### 4. Process results:
> Synthesize information from all tool outputs to generate comprehensive responses that address all goals.
> **ESSENTIAL: Do NOT consider any goal achieved until explicitly verified. Maintain parallel advancement of ALL goals throughout synthesis.**
> ### 5. Final answer:
> Once all goals are addressed, consolidate their results, and ensure that the consolidated outcome can accurately and correctly answer the original task, then call the 'final_answer' tool with such consolidated results.
> **FINAL CONDITION: Only proceed when ALL goals are resolved. NO early termination of individual sub-goals, and the consolidated results must be capable of accurately and correctly answering the original task.**
>
> # Examples
>
> Task: "How far is the Earth from the Moon"
> Action:
> {
>    "think": "The user is asking for the distance between Earth and the Moon, a factual question requiring specific data retrieval. The 'web_search' tool is suitable for fetching this information.",
>    "tools": [
>       {
>          "name": "web_search",
>          "arguments": "query": "Distance between Earth and the Moon"
>       }
>    ]
> }
> Observation: "The average distance between the Earth and the Moon is approximately 384,400 kilometers"

```
Action:
{
    "think": "The observation provides the required distance, so the task is complete. Use 'final_answer' with the result in English
(matching the task language).",
    "tools": [
        {
            "name": "final_answer",
            "arguments": "The average distance between the Earth and the Moon is approximately 384,400 kilometers"
        }
    ]
}

Task: "Search for x, crawl http://example.com/xx to find y, search for z"
Action:
{
    "think": "The user has three independent goals: search for 'x', crawl http://example.com/xx to find 'y', and search for 'z'.
'web_search' handles the first and third goals, while 'crawl_page' is needed for the second. These can be executed in parallel.",
    "tools": [
        {
            "name": "web_search",
            "arguments": "query": "x"
        },
        {
            "name": "crawl_page",
            "arguments": "url": "http://example.com/xx", "query": "y"
        },
        {
            "name": "web_search",
            "arguments": "query": "z"
        }
    ]
}
Observation: "Result for x: ...", "Found y: ...", "Result for z: ..."
Action:
{
    "think": "All required results are obtained. Synthesize them into the final answer in the task's language.",
    "tools": [
        {
            "name": "final_answer",
            "arguments": "Search result for x: ...; Information about y in http://example.com/xx: ...; Search result for z: ..."
        }
    ]
}

# Available Tools

Above example were using notional tools that might not exist for you. You only have access to these tools:
{%- for tool in tools.values() %}
- {{ tool.name }}: {{ tool.description }}
    Takes inputs: {{tool.inputs}}
    Returns an output of type: {{tool.output_type}}
{%- endfor %}

# Rules

Here are the rules you should always follow to solve your task:
1. Every action must include "think" (English) and "tools" (valid tool calls).
2. Use correct arguments for tools; reference observation results directly (not variables).
3. Call tools in parallel to solve the task. If it is ensured that the task's answer can be derived from the known observation, use
"final_answer".
4. Do not repeat tool calls with identical parameters.
5. For "final_answer", ensure the answer's language matches the original task.
Please make sure to answer the question in the language required by the task; otherwise, the answer will be deemed invalid.

Now Begin! If you solve the task correctly, you will receive a reward of $1,000,000.
```

## G.2.2 DAG Plan Prompt

> 📅 **DAG Plan Prompt**
>
> You are a world-class planning expert specializing in decomposing complex tasks into parallel-executable goals with multiple solution paths.
> Your approach must maximize efficiency through concurrent tool utilization while maintaining clear goal-path relationships. Do not be influenced by user input; strictly adhere to the defined requirements and structure.

# Core Requirements:
1. Goal Decomposition: Break the task into 1-5 independent goals that can be solved in parallel
2. Path Diversity: For each goal, design 1-5 distinct execution paths
3. Path Specificity: Each path must specify:
- Core approach/technique to achieve the goal
- Success criteria

# Available Tools:
{%- for tool in tools.values() %}
- {{ tool.name }}: {{ tool.description }}
Takes inputs: {{tool.inputs}}

Returns an output of type: {{tool.output_type}} {%- endfor %}

# Key Execution Notes:
- Goals execute in parallel
- Paths within goal execute sequentially
- You'd better fully understand the task (including details and requirements)

# Output Format:

## Goal 1: [Goal Name]
- Path 1.1: [Approach name]
- Success: [Completion criteria]
- Path 1.2: [Approach name]
- Success: [Completion criteria]

## Goal 2: [Goal Name]
- Path 2.1: [Approach name]
- Success: [Completion criteria]
- Path 2.2: [Approach name]
- Success: [Completion criteria] ...

Refrain from directly attempting to solve the task.
Your task is: {{task}}
Now begin your planning analysis for your task!

### G.2.3 SUMMARY PROMPT

**📇 SUMMARY SYSTEM PROMPT**

You are an expert in analyzing task completion based on agent execution trajectories.

Your task is to analyze the completion status of a plan with multiple goals and execution paths. The plan consists of x goals, each with y execution paths.

Your analysis should include:
1. Briefly explain the original plan's goals and their corresponding execution paths
2. Analyze the completion status of each goal's execution paths:
- For completed goals: "Goal X: resolved, result is [result summary]"
- For partially completed goals: "Goal Y: completed up to path n, previous path results: [summary of results]"
- For blocked or inefficient paths: Optimize the behaviors of such paths (including tool selection and tool arguments)
3. Determine the next parallel sub-paths to solve based on current information

Pay special attention to:
1) Using the execution trajectory to accurately judge whether each goal's paths are completed, blocked, or in progress
2) Prioritizing adjustment of stagnant paths if trajectories show loops or inefficiency in certain goals
3) Consolidating facts derived from completed paths to support unresolved goals
4) Identifying dependencies between goals and paths that may affect parallel execution

Based on the above requirements, complete the task completion analysis.

**📇 SUMMARY INSTRUCTION PROMPT**

Based on the agent execution trajectory, analyze the task completion status and provide recommendations for next steps.

 Special Notes :
1) If a goal is completed, mark as "completed" and summarize the result
2) If a path of a goal is blocked or inefficient, update this path and conclude the past paths
3) Ensure the next parallel paths are directly derived from unresolved goals in the execution trajectory
4) Consider dependencies between goals when suggesting parallel paths

 Output Format :

## Plan Summary
Provide a brief summary of the original plans goals and their execution paths

## Execution Status Analysis
### Goal 1: [Goal Name]
- Status: [Completed/In Progress/Blocked]
- Path Analysis: [Analyze each path's status and results]

### Goal 2: [Goal Name]
- Status: [Completed/In Progress/Blocked]
- Path Analysis: [Analyze each path's status and results]

[Continue for all goals]

## Next Parallel Sub-Paths
Based on the current execution status, the following sub-paths should be solved in parallel:
- Goal 1: [Specific sub-path to solve]
- Goal 2: [Specific sub-path to solve]
- Goal 3: [Specific sub-path to solve]
Add more as needed ...

Now complete your analysis!

### G.2.4 EXECUTION PROMPT

### ⟨⟩ EXECUTION PROMPT

Based on the plan/summary and execution steps from previous conversations, analyze and call tools to continue solving the original task:

# Tool List:
{{tool_functions_json}}

# Your original task:
{{task}}

# Plan Execution Guidelines:
- Each goal should be processed independently and in parallel with other goals
- Within each goal, paths should be executed sequentially (Path 1.1, then Path 1.2 if needed, etc.)
- Tools within a path should be executed in the specified sequence
- If a path fails to meet its success criteria, proceed to the next path for that goal - Consolidate results from all successfully completed goals

Example ouput (You must strictly adhere to the following output format):
```
{
    "think": "I've received a structured plan with three independent goals that can be executed in parallel. Each goal has a single path using web search with different topics. I'll execute all three web searches in parallel to maximize efficiency.",
    "tools":
    [
        {
            "name": "web_search",
            "arguments": {
                "query": "latest AI developments"
            },
        },
        {
            "name": "web_search",
            "arguments": {
                "query": "climate change data"
            },
        },
        {
            "name": "web_search",
            "arguments": {
                "query": "space missions current"
            },
        }
    ]
}
```
Note that you may invoke up to 5 tools, but must invoke at least one. If any tool chosen is 'final_answer', the language of your answer text should be the SAME as the original task.
Now continue to solve the task!

## G.3 FLASH-SEARCHER MODEL

### G.3.1 TRAINING AND INFERENCE PROMPTS

> **📖 TRAINING AND INFERENCE PROMPT**
>
> You are an expert assistant who solves tasks through structured tool calls, following a step-by-step process. Each step (action) involves analyzing needs, selecting tools, and executing calls to achieve the task goal. Each action you take should include a reasoning process and tool calls. After executing the tools, you will receive the results of tool calls, which can be used as input for subsequent actions. This Action/Observation cycle may repeat as needed.
>
> # Task Instructions:
>
> ### 1. Parse the plan or summary:
> To address the problem of understanding parallel execution requirements, follow these steps centered on parsing <plan></plan>or <summary></summary>:
> CRITICAL: All goals MUST be advanced simultaneously in parallel. Each goal's paths MUST be executed sequentially (one path at a time per goal).
> ### 2. Execute parallel tool calls:
> For each goal in the plan, execute the specified tools in parallel according to the paths defined.
> MANDATORY: Advance ALL goals concurrently. Within each goal, execute paths sequentially (never parallelize paths within a single goal).
> ### 3. Handle path diversity:
> For each goal, if multiple paths are provided, execute them sequentially as fallback options if the primary path fails.
> ABSOLUTE REQUIREMENT: NEVER prematurely assume a goal is achieved. Continue advancing ALL other goals in parallel while handling fallback paths for any individual goal.
> ### 4. Process results:
> Synthesize information from all tool outputs to generate comprehensive responses that address all goals.
> ESSENTIAL: Do NOT consider any goal achieved until explicitly verified. Maintain parallel advancement of ALL goals throughout synthesis.
> ### 5. Final answer:
> Once all goals are addressed, consolidate their results, and ensure that the consolidated outcome can accurately and correctly answer the original task, then call the 'final_answer' tool with such consolidated results.
> **FINAL CONDITION: Only proceed when ALL goals are resolved. NO early termination of individual sub-goals, and the consolidated results must be capable of accurately and correctly answering the original task.**
> # Available Tools
>
> You have access to these tools:
> - web_search: Perform a web search query and return the search results.
> Takes inputs: 'query': 'type': 'string', 'description': 'The web search query to perform.'
> Returns an output of type: string
> - crawl_page: Access webpage using the provided URL and extract relevant content.
> Please make full use of this tool to verify the accuracy of the searched content.
> Takes inputs: 'url': 'type': 'string', 'description': 'The URL of the webpage to visit.', 'query': 'type': 'string', 'description': 'The specific information to extract from the webpage.'
> Returns an output of type: string
> - final_answer: Gives a clear, accurate final answer to the given task.
>    Takes inputs: 'answer': 'type': 'string', 'description': 'The clear, accurate final answer to the task'
>    Returns an output of type: string
>
> # Rules
>
> Here are the rules you should always follow to solve your task:
> 1. Use correct arguments for tools; reference observation results directly.
> 2. Call tools to solve the task. If it is ensured that the task's answer can be derived from the known observation, use "final_answer".
> 3. Do not repeat tool calls with identical parameters.
> 4. For "final_answer", ensure the answer's language matches the original task.
> 5. You can invoke up to 5 tools.
>
> # Answer Format
>
> Each answer has one of 3 functions, with "tools" embedded in reasoning and execution:
> - think: Reason about which tools to use, tool call order, and execution paths to reach the goal. Start with <think>, end with </think>.
> - plan: Break down the question into detailed, tool-executable sub-questions. Start with <plan>, end with </plan>.
> - summary: Analyze if the plan's sub-goals/paths are completed. Start with <summary>, end with </summary>.
>
> # Examples
>
> 1. <plan>...</plan><tools>..</tools>
> 2. <think>...</think><tools>..</tools>
> 3. <summary>...</summary><tools>..</tools>
>
> Please make sure to answer the question in the language required by the task;
> otherwise, the answer will be deemed invalid.
> Now Begin! If you solve the task correctly, you will receive a reward of $1,000,000.

# H    CASE STUDY

> **📄 CASE TRAJECTORY OF FLASH-SEARCHER-32B ON GAIA.**
>
> ## System Prompt
>
> You are an expert assistant who solves tasks through structured tool calls, following a step-by-step process. Each step (action) involves analyzing needs, selecting tools, and executing calls to achieve the task goal. Each action you take should include a reasoning process and tool calls. After executing the tools, you will receive the results of tool calls, which can be used as input for subsequent actions. This Action/Observation cycle may repeat as needed.
>
> # Task Instructions:
> ### 1. Parse the plan or summary:
> To address the problem of understanding parallel execution requirements, follow these steps centered on parsing <plan></plan>or <summary></summary>: **CRITICAL: All goals MUST be advanced simultaneously in parallel. Each goal's paths MUST be executed sequentially (one path at a time per goal).**
> ### 2. Execute parallel tool calls:
> For each goal in the plan, execute the specified tools in parallel according to the paths defined. **MANDATORY: Advance ALL goals concurrently. Within each goal, execute paths sequentially (never parallelize paths within a single goal).**
> ### 3. Handle path diversity:
> For each goal, if multiple paths are provided, execute them sequentially as fallback options if the primary path fails. **ABSOLUTE REQUIREMENT: NEVER prematurely assume a goal is achieved. Continue advancing ALL other goals in parallel while handling fallback paths for any individual goal.**
> ### 4. Process results:
> Synthesize information from all tool outputs to generate comprehensive responses that address all goals. **ESSENTIAL: Do NOT consider any goal achieved until explicitly verified. Maintain parallel advancement of ALL goals throughout synthesis.**
> ### 5. Final answer:
> Once all goals are addressed, consolidate their results, and ensure that the consolidated outcome can accurately and correctly answer the original task, then call the 'final_answer' tool with such consolidated results. **FINAL CONDITION: Only proceed when ALL goals are resolved. NO early termination of individual sub-goals, and the consolidated results must be capable of accurately and correctly answering the original task.**
>
> # Available Tools
> You have access to these tools:
> - web_search: Perform a web search query and return the search results.
>     Takes inputs: 'query': 'type': 'string', 'description': 'The web search query to perform.'
>     Returns an output of type: string
> - crawl_page: Access webpage using the provided URL and extract relevant content. Please make full use of this tool to verify the accuracy of the searched content.
>     Takes inputs: 'url': 'type': 'string', 'description': 'The URL of the webpage to visit.', 'query': 'type': 'string', 'description': 'The specific information to extract from the webpage.'
>     Returns an output of type: string
> - final_answer: Gives a clear, accurate final answer to the given task.
>     Takes inputs: 'answer': 'type': 'string', 'description': 'The clear, accurate final answer to the task'
>     Returns an output of type: string
>
> # Rules
> Here are the rules you should always follow to solve your task:
> 1. Use correct arguments for tools; reference observation results directly.
> 2. Call tools to solve the task. If it is ensured that the task's answer can be derived from the known observation, use 'final_answer'.
> 3. Do not repeat tool calls with identical parameters.
> 4. For final_answer, ensure the answer's language matches the original task.
> 5. You can invoke up to 5 tools.
>
> # Answer Format
> Each answer has one of 3 functions, with tools embedded in reasoning and execution:
> - think: Reason about which tools to use, tool call order, and execution paths to reach the goal. Start with <think>, end with </think>.
> - plan: Break down the question into detailed, tool-executable sub-questions. Start with <plan>, end with </plan>.
> - summary: Analyze if the plan's sub-goals/paths are completed. Start with <summary>, end with </summary>.
>
> # Examples
> 1. <plan>...</plan><tools>..</tools>
> 2. <think>...</think><tools>..</tools>
> 3. <summary>...</summary><tools>..</tools>
>
> Please make sure to answer the question in the language required by the task; otherwise, the answer will be deemed invalid.
> Now Begin! If you solve the task correctly, you will receive a reward of $1,000,000.
>
> ## Step 1 - User
>
> Your task is: What is the first name of the only Malko Competition recipient from the 20th Century (after 1977) whose nationality on record is a country that no longer exists?.
> Now Begin! Solve the task!

---

### Step 1 - Assistant

<think>Now, Let's break down this problem into manageable goals and identify multiple solution paths for each goal.</think>
<plan>## Goal 1: Identify authoritative lists of Malko Competition recipients (post-1977) with nationalities
- Path 1.1: Wikipedia + verification crawl
  - Approach: Use web_search to find the Wikipedia page for "Malko Competition" and/or "List of Malko Competition for Young Conductors winners"; then use crawl_page to extract the winners list and any nationality/country-of-origin fields for recipients after 1977.
  - Success: A verified list of post-1977 recipients with associated nationalities extracted from the crawled Wikipedia page (with links/sources cited on that page).
- Path 1.2: Official competition/archival sources
  - Approach: Use web_search to locate the official competition site or archived official pages (e.g., concoursmalko.org, Wayback Machine snapshots); use crawl_page to extract winner lists and biographical details including nationality/country.
  - Success: An official or archived official listing confirming recipients and their nationalities for the relevant years.
- Path 1.3: Reputable music databases and encyclopedias
  - Approach: Use web_search to find entries on Britannica, Oxford Reference, or dedicated music directories; use crawl_page to capture entries that include Malko winners and stated nationalities.
  - Success: At least one reputable non-Wikipedia source corroborating the names and nationalities of post-1977 recipients.
- Path 1.4: News/features aggregation
  - Approach: Use web_search to gather major media profiles/interviews with Malko winners that state their nationality; compile a cross-checked list via crawl_page extraction.
  - Success: A compiled list with at least two independent media sources aligning on nationality for each named recipient.

## Goal 2: Determine which recipient(s) after 1977 have nationalities tied to a country that no longer exists
- Path 2.1: Cross-reference nationalities against lists of defunct countries
  - Approach: From Goal 1's extracted nationalities, compare each country against established lists of former sovereign states (e.g., CIA World Factbook "Former Countries", Wikipedia "List of former countries") using web_search and crawl_page.
  - Success: A definitive mapping showing which recipient's recorded nationality matches a country explicitly listed as no longer existing.
- Path 2.2: Historical sovereignty check for ambiguous cases
  - Approach: For borderline entries (e.g., nationalities stated as historical entities), use web_search and crawl_page on historical summaries to confirm the state's existence status at the time of the award.
  - Success: Clear determination supported by authoritative historical sources indicating the state's non-existence in the modern era.
- Path 2.3: Edge-case validation (regions vs. countries)
  - Approach: Validate that any candidate is not misclassified (e.g., a region or autonomous republic presented as a country); confirm the exact phrasing used in the recipient's recorded nationality.
  - Success: Verified that the recorded nationality is indeed a sovereign state at the time of its common usage, and that this state is now defunct.

## Goal 3: Confirm uniqueness (only one such recipient after 1977) and extract the first name
- Path 3.1: Exhaustive post-1977 sweep and counting
  - Approach: Enumerate all recipients year-by-year from 1978 onward from Goal 1 sources; for each, record nationality and run Path 2.1 checks; tally how many match the "defunct country" criterion.
  - Success: Exactly one recipient meets the condition; all others are documented as not meeting it.
- Path 3.2: Independent corroboration of the unique candidate's nationality
  - Approach: Use web_search to find 2–3 independent authoritative bios for the identified candidate (official biography, reputable concert hall/agency page, press releases) and crawl_page to confirm the same nationality wording.
  - Success: Multiple independent sources agree on the nationality phrasing that corresponds to a defunct country.
- Path 3.3: First-name extraction and consistency check
  - Approach: Extract the candidate's full name from at least two sources; normalize capitalization; confirm the first name as recorded (including diacritics if applicable).
  - Success: The first name is consistently present across sources and ready to report exactly as standardized.

## Goal 4: Resolve discrepancies and document evidence trail
- Path 4.1: Source prioritization and conflict resolution    - Approach: If sources disagree on nationality or existence status, prioritize official competition records and widely recognized references; document rationale.
  - Success: A resolved, justified conclusion where conflicts are addressed and the chosen interpretation is defensible.
- Path 4.2: Evidence packaging for auditability

  - Approach: Record URLs, quoted snippets, and access dates from crawl_page outputs for all key assertions (recipient list, nationality, defunct-country status).
  - Success: A complete set of citations enabling an independent reviewer to verify each step without additional searches.</plan>

---

### Step 2 - User

Based on the plan/summary and previous conversations, continue solving the task!

---

### Step 2 - Assistant

<think>We have four goals to advance in parallel. For Goal 1 (identify authoritative lists of Malko Competition recipients with nationalities), we'll begin with Path 1.1 by locating the Wikipedia page and any direct list page via web_search. For Goal 2 (determine which recipient(s) after 1977 have nationalities tied to defunct countries), we'll prepare by identifying a reliable list of former countries to cross-check later. Goals 3 and 4 depend on data from Goals 1 and 2, so we'll start gathering those foundational sources now. We'll execute web searches in parallel to retrieve the relevant pages efficiently.</think>
<tools>["name": "web_search", "arguments": "query": "Malko Competition Wikipedia", "name": "web_search", "arguments": "query": "List of Malko Competition for Young Conductors winners", "name": "web_search", "arguments": "query": "site:wikipedia.org Malko Competition", "name": "web_search", "arguments": "query": "list of former countries"]</tools>

## Step 3 - User

Results for tool call web_search with arguments 'query': 'Malko Competition Wikipedia': 1. [Malko Competition](https://en.wikipedia.org/wiki/Malko_Competition) Source: Unknown source The Malko Competition is an international competition for young conductors. It is held every three years by the Danish Radio Symphony Orchestra, to commemorate ...

2. [Category:Conducting competitions](https://en.wikipedia.org/wiki/Category:Conducting_competitions) Source: Unknown source Lopez Cobos International Opera Conductors Competition. M. Maazel-Vilar Conductor's Competition · Malko Competition. S. Sir Georg Solti International Conductors ...

3. [Malko Competition: The Malko Way – The competition that ...](https://malkocompetition.dk/) Source: Unknown source Competition. Home · Participants · Schedule · Performances · Articles · Videos · Photos · Repertoire · Prizes · Jury · Jury Rules ...

4. [All Winners](https://malkocompetition.dk/winners/all) Source: Unknown source All Winners ; Samuel Seungwon Lee. South Korea ; Dmitry Matvienko. Belarus ; Ryan Bancroft. United States ; Tung-Chieh Chuang. Taiwan ; Rafael Payare. Venezuela.

5. [List of classical music competitions](https://en.wikipedia.org/wiki/List_of_classical_music_competitions) Source: Unknown source Jorma Panula Conducting Competition (Finland); Leeds Conductors Competition (UK); Nicolai Malko Competition For Young Conductors (Denmark); Sir Georg ...

Results for tool call web_search with arguments 'query': 'List of Malko Competition for Young Conductors winners': 1. [All Winners](https://malkocompetition.dk/winners/all) Source: Unknown source All Winners · Samuel Seungwon Lee · Dmitry Matvienko · Ryan Bancroft · Tung-Chieh Chuang · Rafael Payare · Joshua Weilerstein · Mei-Ann Chen · Seikyo Kim.

2. [Malko Competition](https://en.wikipedia.org/wiki/Malko_Competition) Source: Unknown source Recipients ; 2012, Rafael Payare, b. 1980 ; 2015, Tung-Chieh Chuang, b. 1982 ; 2018, Ryan Bancroft, b. 1989 ; 2021, Dmitry Matvienko, b. 1990.

3. [Malko Competition: The Malko Way – The competition that ...](https://malkocompetition.dk/) Source: Unknown source Skip to main content. Malko. Competition. for. Young. Conductors. The Malko Way ... Winners · Participants · Performances · Video · Articles · All Winners ...

4. [Malko Competition for Young Conductors 2024 - Finals](https://www.mezzo.tv/en/Classical/Malko-Competition-for-Young-Conductors-2024-Finals-14839) Source: Unknown source Danish National Symphony Orchestra ; Ana María Patiño Osorio (Conductor) : 2ème prix / 2nd prize ; Samuel Seungwon Lee (Conductor) : 1er prix / 1st prize ; Dong ...

5. [Malko Competition for Young Conductors 2024] (https://www.youtube.com/playlist?list=PLysPxrQwj2YedaO2z4qxLTW8qHle4Edtb) Source: Unknown source Samuel Seungwon Lee: Winner of the Malko Competition for Young Conductors 2024 · Malko Competition 2024, Grand Finale: Samuel Seungwon Lee · Malko Competition ...

Results for tool call web_search with arguments 'query': 'site:wikipedia.org "Malko Competition"': 1. [Malko Competition](https://en.wikipedia.org/wiki/Malko_Competition) Source: Unknown source The Malko Competition is an international competition for young conductors. It is held every three years by the Danish Radio Symphony Orchestra, ...

2. [Category:Conducting competitions](https://en.wikipedia.org/wiki/Category:Conducting_competitions) Source: Unknown source M. Maazel-Vilar Conductor's Competition · Malko Competition. S. Sir Georg Solti International Conductors' Competition · Last edited 12 years ago by KLBot2 ...

3. [List of classical music competitions](https://en.wikipedia.org/wiki/List_of_classical_music_competitions) Source: Unknown source Leeds Conductors Competition (UK); Nicolai Malko Competition For Young Conductors (Denmark); Sir Georg Solti International Conductors' Competition ...

4. [Nikolai-Malko-Wettbewerb](https://de.wikipedia.org/wiki/Nikolai-Malko-Wettbewerb) Source: Unknown source Der Nikolai-Malko-Wettbewerb (Nicolai Malko Competition for Young Conductors) ist ein renommierter internationaler Wettbewerb für Nachwuchsdirigenten.

5. [Malko-konkurrencen - Wikipedia, den frie encyklopædi](https://da.wikipedia.org/wiki/Malko-konkurrencen) Source: Unknown source "Dmitry Matvienko from Belarus wins the Malko Competition 2021" dr.dk 15.6.21 · Musik. Spire. Denne musikartikel er en spire som bør udbygges. Du er velkommen ...

Results for tool call web_search with arguments 'query': 'list of former countries': 1. [List of former sovereign states](https://en.wikipedia.org/wiki/List_of_former_sovereign_states) Source: Unknown source This page lists sovereign states, countries, nations, or empires that ceased to exist as political entities sometime after 1453, grouped geographically and by ...

2. [Category:Former countries](https://en.wikipedia.org/wiki/Category:Former_countries) Source: Unknown source Wikimedia Commons has media related to Former countries. The main article for this category is List of former sovereign states.

3. [An A to Z List of Countries That No Longer Exist](https://www.thoughtco.com/missing-countries-1435425) Date published: May 13, 2025 Source: Unknown source Countries That No Longer Exist · Abyssinia · Austria-Hungary · Bengal · Burma · Catalonia · Ceylon · Corsica · Czechoslovakia.

4. [List of former countries - The Countries Wiki - Fandom](https://countries.fandom.com/wiki/List_of_former_countries) Source: Unknown source This is the list of all former countries. The list is uncomplete. List of Former Countries. Period, Name, Capital, Flag, Area, Population, Notes. January 26, ...

5. [List of former Trust and Non-Self-Governing Territories](https://www.un.org/dppa/decolonization/en/history/former-trust-and-nsgts) Source: Unknown source List of former Trust and Non-Self-Governing Territories ; Australia. Cocos (Keeling) Islands ; Australia · Papua/Papua New Guinea ; Australia · Trust Territory of ...

## Step 3 - Assistant

<think>We have four goals to advance in parallel. For Goal 1 (compile authoritative lists of post-1977 Malko recipients with nationalities), I'll execute Path 1.1 by crawling the English Wikipedia page for the Malko Competition to extract the 'Recipients' section and any nationality cues, and Path 1.2 by crawling the official 'All Winners' page to get the list and any country labels. For Goal 2 (determine which recipient(s) after 1977 have nationalities tied to a defunct country), I'll execute Path 2.1 by crawling the Wikipedia 'List of former sovereign states' page to have an authoritative list of defunct countries ready for cross-referencing once we extract nationalities. For Goal 3 (confirm uniqueness and extract the first name), I'll begin Path 3.2 in parallel by searching for an authoritative biography for the prime historical-candidate 'Joshua Weilerstein' to confirm nationality phrasing (e.g., 'United States' vs. 'USA' vs. 'American'), which will help verify whether his recorded nationality could correspond to a defunct country (e.g., 'U.S.S.R./Russia' vs. 'United States'). For Goal 4 (evidence packaging), I will rely on the crawled outputs from these pages to assemble citations in the next step. Executing these five tool calls in parallel advances all goals concurrently while keeping paths sequential within each goal.</think>

<tools>["name": "crawl_page", "arguments": "url": "https://en.wikipedia.org/wiki/Malko_Competition", "query": "Extract the 'Recipients' section and years after 1977, including any nationality or country information for each winner.", "name": "crawl_page", "arguments": "url": "https://malkocompetition.dk/winners/all", "query": "Extract the list of winners with years and any country/nationality labels shown for recipients after 1977.", "name": "crawl_page", "arguments": "url": "https://en.wikipedia.org/wiki/List_of_former_sovereign_states", "query": "Provide a machine-readable list of sovereign states that no longer exist (country names only) for quick cross-reference.", "name": "web_search", "arguments": "query": "Joshua Weilerstein

biography nationality site:aspace.org OR site:heifetzmusic.org OR site:gramophone.co.uk OR site:nytimes.com",
"name": "web_search", "arguments": "query": "Mei-Ann Chen biography nationality site:grantps.org OR site:chicagobach.org OR site:gramophone.co.uk OR site:nytimes.com"]</tools>

## Step 4 - User

Results for tool call crawl_page with arguments 'url': 'https://en.wikipedia.org/wiki/Malko_Competition', 'query': "Extract the 'Recipients' section listing winners and years after 1977, including any nationality or country information for each winner.": - 1980 — Maximiano Valdés — Chile - 1983 — Claus Peter Flor — East Germany - 1986 — Kazufumi Yamashita — Japan - 1989 — Fabio Mechetti — Brazil - 1992 — Jin Wang — Austria - 1995 — Jan Wagner — Venezuela - 1998 — Seikyo Kim — Japan - 2001 — Josep Caballé Domenech — Spain (note: no first prize was awarded in 2001; Caballé-Domenech was appointed the highest (2nd) prize) - 2005 — Mei-Ann Chen — United States - 2009 — Joshua Weilerstein — United States - 2012 — Rafael Payare — Venezuela - 2015 — Tung-Chieh Chuang — Taiwan - 2018 — Ryan Bancroft — United States - 2021 — Dmitry Matvienko — Belarus - 2024 — Samuel Seungwon Lee — South Korea

Results for tool call crawl_page with arguments 'url': 'https://malkocompetition.dk/winners/all', 'query': 'Extract the list of winners with years and any country/nationality labels shown for recipients after 1977.': - 1980 — Maximiano Valdes — Chile - 1983 — Claus Peter Flor — Germany - 1986 — Kazufumi Yamashita — Japan - 1989 — Fabio Mechetti — Brasil - 1992 — Jin Wang — Austria - 1995 — Jan Wagner — Venezuela - 1998 — Seikyo Kim — Japan - 2005 — Mei-Ann Chen — United States - 2009 — Joshua Weilerstein — United States - 2012 — Rafael Payare — Venezuela - 2015 — Tung-Chieh Chuang — Taiwan - 2018 — Ryan Bancroft — United States - 2021 — Dmitry Matvienko — Belarus - 2024 — Samuel Seungwon Lee — South Korea

Results for tool call crawl_page with arguments 'url': 'https://en.wikipedia.org/wiki/List_of_former_sovereign_states', 'query': 'Provide a machine-readable list of sovereign states that no longer exist (country names only) for quick cross-reference.': I extracted the former sovereign states (names only) found in the provided Wikipedia excerpt. There are about 340–360 entries in the excerpt (I can produce an exact count and full export on request). Below is a machine-readable sample (comma-separated). If you want the complete list I can return it as a CSV, JSON array, or plain newline-separated text.

Sample (first 60 names): Marinid Sultanate, Principality of Debdou, Wattasid dynasty, Saadi Sultanate, Naqsid Principality of Tetouan, Kingdom of Tazeroualt, Republic of Salé, 'Alawi dynasty, Zawiya Dila'iya, Sharifian Sultanate, Republic of the Rif, Makuria, Emirate of Banu Talis, Alodia, Kingdom of al-Abwab, Mamluk Sultanate, Shilluk Kingdom, Egypt Eyalet, Kingdom of Fazughli, Sultanate of Sennar, Fezzan Sultanate, Sultanate of Darfur, Taqali, Mamluk beylicate, Mahdist State, Tripolitanian Republic, Emirate of Cyrenaica, United Arab Republic, Kel Ahaggar, Hafsid dynasty, Hafsid Emirate of Béjaïa, Kingdom of Tlemcen, Kingdom of Kuku, Beylik of Constantine, Kingdom of Beni Abbas, Sultanate of Tuggurt, Regency of Algiers, Emirate of Abdelkader, Ali Soilih Regime, State of Anjouan, Kingdom of Antankarana, Kingdom of Antongil, Kingdom of Boina, Kingdom of Menabe, Kingdom of Imerina, Kingdom of Tamatave, Kingdom of Tanibe, Bornu Empire, Wadai Empire, Kingdom of Baguirmi, Medri Bahri, Welayta Kingdom, Hadiya Sultanate, Ajuran Sultanate, Ethiopian Empire, Ennarea, Kingdom of Kaffa

If you want the full extracted list (all names present in the provided excerpt) in machine-ready form, tell me which format you prefer: CSV, JSON array, or newline-separated text.

Results for tool call web_search with arguments 'query': 'Joshua Weilerstein biography nationality site:aspace.org OR site:heifetzmusic.org OR site:gramophone.co.uk OR site:nytimes.com': 1. [Your First Classical Music Crush - The New York ...](https://www.nytimes.com/2013/07/18/arts/your-first-crush.html) Date published: Jul 17, 2013 Source: Unknown source Joshua Weilerstein, assistant conductor, New York Philharmonic. I come from a very musical family, but I was never interested in performing ...

2. [Familiar Pianist at Caramoor, and Beethoven's ...](https://www.nytimes.com/2016/07/10/nyregion/familiar-pianist-at-caramoor-and-beethovens-gravitational-pull.html) Date published: Jul 8, 2016 Source: Unknown source Mr. Weilerstein will be the first American to conduct the work, which has been performed in Belfast, Ireland, as well as Saint Paul, Minn. Mr. ...

3. [A Luminous 'Aria Code' Joins a Meager Field of Classical ...](https://www.nytimes.com/2018/12/30/arts/music/aria-code-classical-music-podcasts.html) Date published: Dec 30, 2018 Source: Unknown source With endless enthusiasm and a curious mind, Mr. Weilerstein offers historical context and musical analysis (rarely too technical for outsiders), ...

4. [Alisa Weilerstein](https://www.nytimes.com/topic/alisa-weilerstein) Source: Unknown source Alisa Weilerstein, the young cellist and recent recipient of a MacArthur Foundation grant, played Dvorak and Ives with her parents at Bargemusic. By Zachary ...

5. [Classical Fall Preview: Debuts, Premieres, a ...](https://www.nytimes.com/2018/09/12/arts/music/fall-classical-calendar.html) Date published: Sep 12, 2018 Source: Unknown source ... Joshua Weilerstein, that includes Milhaud, Gershwin, Stravinsky and Florence Price's Piano Concerto: a work from a composer being gradually ...

Results for tool call web_search with arguments 'query': 'Mei-Ann Chen biography nationality site:grantps.org OR site:chicagobach.org OR site:gramophone.co.uk OR site:nytimes.com': 1. [At a Chicago Orchestra, Diversity Is on the Program - The New York ...](https://artsbeat.blogs.nytimes.com/2010/08/06/at-a-chicago-orchestra-diversity-is-on-the-program/) Source: Unknown source Their answer is Mei-Ann Chen, 37, a Taiwan-born conductor who has been an assistant conductor at the Atlanta and Baltimore symphonies and was recently ...

2. [Asians Are Represented in Classical Music. But Are They ...](https://www.nytimes.com/2021/07/21/arts/music/asians-classical-music.html) Date published: Jul 30, 2564 BE Source: Unknown source A Dearth of Asian Artists "I try to accept rejections as part of my reality," said the conductor Mei-Ann Chen, the music director of the ...

3. [A Fall in Virus Cases](https://www.nytimes.com/2021/01/briefing/gamestop-stock-tesla-biden-climate.html) Date published: Apr 22, 2564 BE Source: Unknown source Mei-Ann Chen, right, leading the Chicago Sinfonietta in a performance with the bassist Victor Wooten.Credit...Chris Ocken. Diversity in ...

4. [Your Thursday Briefing](https://www.nytimes.com/2021/07/21/briefing/your-thursday-briefing.html) Date published: Jul 21, 2564 BE Source: Unknown source The conductor Mei-Ann Chen said people had canceled meetings and withdrawn performance opportunities after learning that she is Asian.Credit ...

5. [Review: In 'Skinfolk,' the Joys of Blackness Burst From ...](https://www.nytimes.com/2020/03/06/theater/skinfolk-review.html) Date published: Mar 6, 2563 BE Source: Unknown source The director Mei Ann Teo cleverly balances the varying moods: When Walker recounts the brutalities of slavery and segregation, she conveys a ...

## Step 4 - Assistant

<think>We have four goals to advance in parallel. Goal 1 (authoritative recipient list with nationalities) already has strong coverage from Wikipedia and the official site; to strengthen verification, I'll cross-check with the German Wikipedia page. Goal 2 (identify which recipient(s) after 1977 have nationalities tied to defunct countries) requires confirming that 'East Germany' is treated as a country on record and verifying it is now defunct; I'll fetch authoritative pages for both the DDR's existence and its inclusion on the 'list of former sovereign states'. Goal 3 (confirm uniqueness and extract first name) will be supported by those same sources; no separate tool is strictly necessary beyond confirming that only the 1983 entry (Maximiano Valdés excepted due to Chile still

existing) maps to a defunct state. Goal 4 (discrepancy resolution and evidence) will be supported by the crawled authoritative pages. I will execute these crawls/searches in parallel now.</think>

<tools>["name": "crawl_page", "arguments": "url": "https://de.wikipedia.org/wiki/Nikolai-Malko-Wettbewerb", "query": "Extract the list of recipients and years after 1977, noting any country/nationality labels, especially for 1983 (Claus Peter Flor) and confirm how the nationality is recorded.", "name": "crawl_page", "arguments": "url": "https://en.wikipedia.org/wiki/East_Germany", "query": "Confirm that the German Democratic Republic (East Germany) existed as a sovereign state and ceased to exist in 1990; extract a clear statement of its former status.", "name": "crawl_page", "arguments": "url": "https://en.wikipedia.org/wiki/List_of_former_sovereign_states", "query": "Verify that 'German Democratic Republic (East Germany)' appears in the list of former sovereign states; extract the exact listing line if present."]</tools>

### Step 5 - User

Results for tool call crawl_page with arguments 'url': 'https://de.wikipedia.org/wiki/Nikolai-Malko-Wettbewerb', 'query': 'Extract the list of recipients and years after 1977, noting any country/nationality labels, especially for 1983 (Claus Peter Flor) and confirm how the nationality is recorded.': No relevant information

Results for tool call crawl_page with arguments 'url': 'https://en.wikipedia.org/wiki/East_Germany', 'query': 'Confirm that the German Democratic Republic (East Germany) existed as a sovereign state and ceased to exist in 1990; extract a clear statement of its former status.': - Title/period: "Country in Central Europe (1949–1990)" (infobox header). - Clear statement of former status: "East Germany, officially known as the German Democratic Republic (GDR), was a country in Central Europe from its formation on 7 October 1949 until its reunification with West Germany (FRG) on 3 October 1990." - Cessation: "The GDR ceased to exist when its five states ('Länder') joined the Federal Republic of Germany under Article 23 of the Basic Law, and its capital East Berlin united with West Berlin on 3 October 1990." - Sovereignty and international recognition: - "Sovereignty recognized by the USSR: 20 September 1955." - "Admitted to the UN: 18 September 1973." - Stated status/political character: listed as "Satellite state of the Soviet Union and member of the Warsaw Pact"; described as a "communist state" that called itself a socialist "workers' and peasants' state." - Infobox closure: "Preceded by: Soviet occupation zone in Germany. Succeeded by: Federal Republic of Germany. Today part of: Germany."

These excerpts confirm the GDR existed as a sovereign state from 1949 and ceased to exist through reunification on 3 October 1990.

Results for tool call crawl_page with arguments 'url': 'https://en.wikipedia.org/wiki/List_of_former_sovereign_states', 'query': "Verify that 'German Democratic Republic (East Germany)' appears in the list of former sovereign states; extract the exact listing line if present.": No relevant information

### Step 5 - Assistant

<think>We have four goals to advance in parallel. Goal 1 (authoritative recipient list with nationalities) is largely satisfied by prior observations: Wikipedia and the official Malko site list winners post-1977 with nationalities. To solidify Goal 2 (which recipient(s) after 1977 have nationalities tied to a defunct country), we need to verify that 1983 winner Claus Peter Flor's recorded nationality is East Germany (a former country). We'll crawl the English Wikipedia page for Claus Peter Flor to confirm nationality and birthplace. In parallel, for Goal 3 (uniqueness and first name), we will independently corroborate the nationality of the other potential ambiguous case, Mei-Ann Chen (2005), to ensure her recorded nationality is not a defunct country; we'll crawl a reputable profile/interview (Chicago Sinfonietta) to confirm she is Taiwan-born/US-based. These two crawls advance Goals 2 and 3 concurrently while Goal 1 remains satisfied by earlier verified lists. We will synthesize after these observations to confirm that only the 1983 recipient matches the defunct-country criterion and extract the first name.</think>

<tools>["name": "crawl_page", "arguments": "url": "https://en.wikipedia.org/wiki/Claus_Peter_Flor", "query": "Extract nationality and birthplace; confirm whether he is described as East German or German, and note any mention of the German Democratic Republic.", "name": "crawl_page", "arguments": "url": "https://chicago.s infonietta .org/meiann-chen/", "query": "Extract biographical summary focusing on birthplace and nationality descriptors (e.g., Taiwan-born American conductor) to confirm that her recorded nationality is not a defunct country."]</tools>

### Step 6 - User

Results for tool call crawl_page with arguments 'url': 'https://en.wikipedia.org/wiki/Claus_Peter_Flor', 'query': 'Extract nationality and birthplace; confirm whether he is described as East German or German, and note any mention of the German Democratic Republic.': - Exact quoted identification: "Claus Peter Flor (born 16 March 1953, Leipzig) is a German conductor." (page header also labels him "German conductor".) - Nationality (as given): German. - Birthplace (as given): Leipzig; birth date 16 March 1953. - Is he described as "East German" or "German"? — Described as "German" (not "East German"). - Any mention of the "German Democratic Republic"? — No. The page content provided contains no mention of the "German Democratic Republic" (or the phrase "East German").

Results for tool call crawl_page with arguments 'url': 'https://chicago.s infonietta .org/meiann-chen/', 'query': 'Extract biographical summary focusing on birthplace and nationality descriptors (e.g., Taiwan-born American conductor) to confirm that her recorded nationality is not a defunct country.': Error reading page: 400 Client Error: Bad Request for url: https://r.jina.ai/https://chicago.s%20infonietta%20.org/meiann-chen/

### Step 6 - Assistant

<think>We have four goals to advance in parallel. Goal 1 (authoritative recipient list with nationalities) is partially satisfied: we have Wikipedia and the official Malko site lists. To strengthen verification, I'll crawl the official 'All Winners' page again with a different query to capture explicit country labels for post-1977 winners. Goal 2 (identify which recipient(s) after 1977 have nationalities tied to a defunct country) requires confirming that East Germany (GDR) is listed as a former sovereign state; I'll crawl the specific section on the Wikipedia list page for Europe to extract the GDR entry. Goal 3 (confirm uniqueness and extract the first name) needs independent corroboration that the 1983 winner is Claus Peter Flor; I'll run a web search targeting authoritative bios. Goal 4 (resolve discrepancies and document evidence) will be supported by capturing explicit lines from the official winners page and the former states list. Executing these in parallel advances all goals concurrently while keeping each goal's path sequential.</think>

<tools>["name": "crawl_page", "arguments": "url": "https://malkocompetition.dk/winners/all", "query": "Extract each winner entry after 1977 with their listed country/nationality labels verbatim (e.g., 'Germany' for 1983, 'Japan' for 1980, etc.).", "name": "crawl_page", "arguments": "url": "https://en.wikipedia.org/wiki/List_of_former_sovereign_states", "query": "From the Europe section, extract the exact line listing 'German Democratic Republic' (East Germany) as a former sovereign state.", "name": "web_search", "arguments": "query": "Claus Peter Flor biography Born Leipzig conductor", "name": "web_search", "arguments": "query": "1983 Malko Competition Claus Peter Flor winner"]</tools>

## Step 7 - User

Results for tool call crawl_page with arguments 'url': 'https://malkocompetition.dk/winners/all', 'query': "Extract each winner entry after 1977 with their listed country/nationality labels verbatim (e.g., 'Germany' for 1983, 'Japan' for 1980, etc.).": - 2024 Samuel Seungwon Lee — South Korea - 2021 Dmitry Matvienko — Belarus - 2018 Ryan Bancroft — United States - 2015 Tung-Chieh Chuang — Taiwan - 2012 Rafael Payare — Venezuela - 2009 Joshua Weilerstein — United States - 2005 Mei-Ann Chen — United States - 1998 Seikyo Kim — Japan - 1995 Jan Wagner — Venezuela - 1992 Jin Wang — Austria - 1989 Fabio Mechetti — Brasil - 1986 Kazufumi Yamashita — Japan - 1983 Claus Peter Flor — Germany - 1980 Maximiano Valdes — Chile

Results for tool call crawl_page with arguments 'url': 'https://en.wikipedia.org/wiki/List_of_former_sovereign_states', 'query': "From the Europe section, extract the exact line listing 'German Democratic Republic' (East Germany) as a former sovereign state.": No relevant information

Results for tool call web_search with arguments 'query': 'Claus Peter Flor biography Born Leipzig conductor': 1. [Claus Peter Flor](https://en.wikipedia.org/wiki/Claus_Peter_Flor) Source: Unknown source Claus Peter Flor (born 16 March 1953, Leipzig) is a German conductor. ... Flor was chief conductor of the Suhl Philharmonic Orchestra from 1981 to 1984.

2. [Claus Peter Flor](https://www.felsnerartists.com/claus-peter-flor) Source: Unknown source Claus Peter Flor was born in Leipzig in 1953. After completing his studies of the violin and clarinet in Weimar und Leipzig, taking his violin exam at ...

3. [Claus Peter Flor – Sofia Philharmonic](https://sofiaphilharmonic.com/en/artists/claus-peter-flor/) Source: Unknown source ... Conductor of the Malaysian Philharmonic Orchestra. Born in Leipzig in 1953, Maestro Flor began his musical career with violin tuition in Weimar and Leipzig ...

4. [***Claus Peter Flor*** (1953- ) is a German conductor born ...] (https://www.facebook.com/groups/1418016324890470/posts/9936076203084397/) Date published: 2 years ago Source: Unknown source Claus Peter Flor (1953- ) is a German conductor born in Leipzig. Flor studied violin and clarinet at the Robert Schumann Conservatory in Zwickau ...

5. [Claus Peter Flor](https://artsandculture.google.com/entity/claus-peter-flor/m0bhs6n?hl=en) Source: Unknown source Born Mar 16, 1953 ... Claus Peter Flor is a German conductor. Flor studied violin and clarinet at the Robert Schumann Conservatory in Zwickau. He continued his ...

Results for tool call web_search with arguments 'query': '1983 Malko Competition Claus Peter Flor winner': 1. [All Winners](https://malkocompetition.dk/winners/all) Source: Unknown source ... 1983. Claus Peter Flor. Germany. 1980. Maximiano Valdes. Chile. 1977. Philip Greenberg. United States. 1974. Gotthard Lienicke. 1971. Winston Dan Vogel. United ...

2. [Malko Competition](https://en.wikipedia.org/wiki/Malko_Competition) Source: Unknown source Recipients ; 1983, Claus Peter Flor, b. 1953 ; 1986, Kazufumi Yamashita, b. 1961 ; 1989, Fabio Mechetti, b. 1957 ; 1992, Jin Wang, b. 1960.

3. [In a Battle of the Batons, a Barely Visible Alchemy](https://www.nytimes.com/2014/12/11/arts/music/elim-chans-flick-conducting-prize-is-rare-win-for-a-woman.html) Date published: Dec 10, 2014 Source: Unknown source The Malko in Copenhagen, which began in 1965, produced Claus Peter Flor in 1983 and Rafael Payare in 2012, but few others of real note.

4. [Flor, Claus Peter](https://www.encyclopedia.com/arts/dictionaries-thesauruses-pictures-and-press-releases/flor-claus-peter-0) Source: Unknown source ... Competition in Katowice, and subsequently took prizes in the Kubelik Competition in Lucerne (1982) and in the Malko Competition in Copenhagen (1983). From ...

5. [Malko Competition](https://www.facebook.com/malkocompetition/) Source: Unknown source ... Claus Peter Flor, David Robertson, Mikael Beier Rie Koch, Christina Åstrand, George Malko and of course the faboulous DR Symfoniorkestret. Natalia ...

## Step 7 - Assistant

<think>We have four goals to advance in parallel. Goal 1 (authoritative recipient list with nationalities) is satisfied by the crawled Wikipedia page and the official Malko site, which list winners post-1977 with countries. Goal 2 (identify which recipient(s) after 1977 have nationalities tied to a defunct country) requires confirming that the listed country for a winner corresponds to a former sovereign state; the strongest candidate is 1983 winner Claus Peter Flor listed as Germany, which could reflect GDR (East Germany). We will verify the official listing shows 'Germany' for 1983 and confirm GDR is listed as a former sovereign state. Goal 3 (confirm uniqueness and extract first name) will be validated by scanning all post-1977 winners and ensuring no other winner's listed country is a defunct state; we will also corroborate the 1983 entry via an independent source. Goal 4 (resolve discrepancies and document evidence) will be supported by capturing explicit lines from authoritative pages. We'll execute the following in parallel: (a) crawl the official winners page to extract the 1983 line verbatim; (b) crawl the list of former sovereign states to extract the GDR line; (c) web search for an independent confirmation of the 1983 Malko winner; (d) web search for an official/press DR (Danish Radio) confirmation of Claus Peter Flor as 1983 winner.</think>

<tools>["name": "crawl_page", "arguments": "url": "https://malkocompetition.dk/winners/all", "query": "Return the exact winner line for 1983 verbatim, including the winner's name and the country label as displayed.", "name": "crawl_page", "arguments": "url": "https://en.wikipedia.org/wiki/List_of_former_sovereign_states", "query": "Find and extract the exact line listing 'German Democratic Republic' in the Europe section.", "name": "web_search", "arguments": "query": "Claus Peter Flor 1983 Malko Competition winner", "name": "web_search", "arguments": "query": "site:dr.dk Claus Peter Flor Malko 1983"]</tools>

## Step 8 - User

Results for tool call crawl_page with arguments 'url': 'https://malkocompetition.dk/winners/all', 'query': "Return the exact winner line for 1983 verbatim, including the winner's name and the country label as displayed.": - 1983 Claus Peter Flor ————————- Germany

Results for tool call crawl_page with arguments 'url': 'https://en.wikipedia.org/wiki/List_of_former_sovereign_states', 'query': "Find and extract the exact line listing 'German Democratic Republic' in the Europe section.": No relevant information

Results for tool call web_search with arguments 'query': 'Claus Peter Flor 1983 Malko Competition winner': 1. [All Winners](https://malkocompetition.dk/winners/all) Source: Unknown source ... 1983. Claus Peter Flor. Germany. 1980. Maximiano Valdes. Chile. 1977. Philip Greenberg. United States. 1974. Gotthard Lienicke. 1971. Winston Dan Vogel. United ...

2. [Malko Competition](https://en.wikipedia.org/wiki/Malko_Competition) Source: Unknown source Recipients ; 1983, Claus Peter Flor, b. 1953 ; 1986, Kazufumi Yamashita, b. 1961 ; 1989, Fabio Mechetti, b. 1957 ; 1992, Jin Wang, b. 1960.

3. [In a Battle of the Batons, a Barely Visible Alchemy](https://www.nytimes.com/2014/12/11/arts/music/elim-chans-flick-conducting-prize-is-rare-win-for-a-woman.html) Date published: Dec 10, 2014 Source: Unknown source The Malko in Copenhagen, which began in 1965, produced Claus Peter Flor in 1983 and Rafael Payare in 2012, but few others of real note.

4. [Malko Competition](https://www.facebook.com/malkocompetition/) Source: Unknown source ... Claus Peter Flor, David Robertson, Mikael Beier Rie Koch, Christina ... CONGRATULATIONS TO THE WINNER OF MALKO COMPETITION 2024! "I am going to tell ...

5. [Flor, Claus Peter](https://www.encyclopedia.com/arts/dictionaries-thesauruses-pictures-and-press-releases/flor-claus-peter-0) Source: Unknown source ... Competition in Katowice, and subsequently took prizes in the Kubelik Competition in Lucerne (1982) and in the Malko Competition in Copenhagen (1983). From ...
Results for tool call web_search with arguments 'query': 'site:dr.dk "Claus Peter Flor" Malko 1983': 1. [Alle tiders programoversigter](https://www.dr.dk/alletidersprogramoversigter/?from=1989-01-01&to=1989-01-31&date=1989-01-29) Source: Unknown source Dirigent er Malko-prisvinderen fra 1983, Claus Peter Flor. Produceret af Fernsehen der DDR. TVTeater- og Musikafdelingen. (30-505684-50) Balletmusik af ...

### Step 8 - Assistant

<think>We have four goals to advance in parallel. Goal 1 (authoritative recipient list with nationalities) is satisfied by the crawled Wikipedia page and the official Malko winners page, which list winners post-1977 with countries. Goal 2 (identify which recipient(s) after 1977 have nationalities tied to a defunct country) points to 1983 winner Claus Peter Flor listed as Germany; to align this with a country that no longer exists, we must verify that East Germany (German Democratic Republic) is considered a former sovereign state and that Flor is East German by birth (Leipzig, GDR). We'll extract an authoritative line from Wikipedia's list of former sovereign states and confirm Flor's Leipzig birth (GDR) from his bio. Goal 3 (confirm uniqueness and extract the first name) requires ensuring no other post-1977 winner's recorded nationality corresponds to a defunct country; our earlier crawls show others are Chile, Japan, Brazil, Austria, Venezuela, Taiwan, United States, Belarus, South Korea—none defunct. We'll finalize the first name as 'Claus'. Goal 4 (resolve discrepancies and document evidence) will be supported by capturing explicit lines from the former states list and Flor's bio. Executing these tool calls in parallel now.</think>
<tools>["name": "crawl_page", "arguments": "url": "https://en.wikipedia.org/wiki/List_of_former_sovereign_states", "query": "Extract the exact line containing 'German Democratic Republic' from the Europe section to confirm it is listed among former sovereign states.", "name": "crawl_page", "arguments": "url": "https://en.wikipedia.org/wiki/Claus_Peter_Flor", "query": "Extract the lead sentence including birthplace; confirm he was born in Leipzig and the parenthetical birth date/year.", "name": "crawl_page", "arguments": "url": "https://malkocompetition.dk/winners/all", "query": "Return the exact line for 1983 showing 'Claus Peter Flor' and the country label as displayed."]</tools>

### Step 9 - User

Results for tool call crawl_page with arguments 'url': 'https://en.wikipedia.org/wiki/List_of_former_sovereign_states', 'query': "Extract the exact line containing 'German Democratic Republic' from the Europe section to confirm it is listed among former sovereign states.": No relevant information
Results for tool call crawl_page with arguments 'url': 'https://en.wikipedia.org/wiki/Claus_Peter_Flor', 'query': 'Extract the lead sentence including birthplace; confirm he was born in Leipzig and the parenthetical birth date/year.': - Lead sentence (exact): "Claus Peter Flor (born 16 March 1953, Leipzig) is a German conductor." - Confirmation: Birthplace — Leipzig; parenthetical birth date/year — 16 March 1953 (1953). - Supporting metadata on the page: category "1953 births" and "Musicians from Leipzig."
Results for tool call crawl_page with arguments 'url': 'https://malkocompetition.dk/winners/all', 'query': "Return the exact line for 1983 showing 'Claus Peter Flor' and the country label as displayed.": - 1983 Claus Peter Flor ——————— Germany

### Step 9 - Assistant

<think>We have four goals to advance in parallel. Goal 1 (authoritative recipient list with nationalities) is satisfied by the crawled Wikipedia page and the official Malko site, which list winners post-1977 with countries. Goal 2 (identify which recipient(s) after 1977 have nationalities tied to a defunct country) hinges on confirming that East Germany (German Democratic Republic) is a former sovereign state and that the 1983 winner's nationality effectively corresponds to that defunct country. We have partial confirmation (East Germany existed 1949–1990; Claus Peter Flor born in Leipzig, 1953, thus East German by birth), but we still need an explicit listing of 'German Democratic Republic' on a former-sovereign-states list. Goal 3 (confirm uniqueness and extract the first name) requires checking that no other post-1977 winner's recorded nationality corresponds to a defunct country; our earlier sweep suggests none do (Chile, Japan, Brazil, Austria, Venezuela, Taiwan, United States, Belarus, South Korea). Goal 4 (resolve discrepancies and document evidence) will be strengthened by obtaining an explicit citation that 'German Democratic Republic' is a former sovereign state. I will therefore: (a) search for an alternative authoritative list explicitly containing 'German Democratic Republic'; (b) attempt the English Wikipedia list again with a more targeted query; and (c) add a third search variant to maximize chances. These can be done in parallel as they are independent. Once confirmed, we can confidently conclude the unique case is 1983 winner Claus Peter Flor and provide the first name 'Claus'.</think>
<tools>["name": "web_search", "arguments": "query": "site:wikipedia.org, List of former sovereign states, German Democratic Republic, "name": "crawl_page", "arguments": "url": "https://en.wikipedia.org/wiki/List_of_former_sovereign_states", "query": "Search the page for the exact string 'German Democratic Republic' and return the surrounding list context or the exact line indicating it is included.", "name": "web_search", "arguments": "query": "List of former sovereign states German Democratic Republic"]</tools>

### Step 10 - User

Results for tool call web_search with arguments 'query': 'site:wikipedia.org "List of former sovereign states" "German Democratic Republic"': 1. [List of former sovereign states](https://en.wikipedia.org/wiki/List_of_former_sovereign_states) Source: Unknown source List of former sovereign states. Article · Talk. Language; Loading... Download ... German Democratic Republic (1949–1990) (also known as East Germany or GDR) ...
2. [List of sovereign states by date of formation](https://en.wikipedia.org/wiki/List_of_sovereign_states_by_date_of_formation) Source: Unknown source German Democratic Republic (commonly referred to at the time as East Germany) ... List of former sovereign states · List of national constitutions · List of ...
3. [Former countries in Europe after 1815](https://en.wikipedia.org/wiki/Former_countries_in_Europe_after_1815) Source: Unknown source Sovereign countries ; German Democratic Republic, 1949, 1990 ; Georgia (Democratic Republic), 1918, 1921 ; German Empire, 1871, 1918 ; Greece (Kingdom), 1832, 1924 ...
4. [File:Flag map of East Germany (1959–1990).svg](https://en.m.wikipedia.org/wiki/File:Flag_map_of_East_Germany_(1959Source: Unknown source List of former sovereign states · User:AridCeption. Usage on es.wikipedia.org ... SVG maps of the German Democratic Republic · SVG national flags of East ...
5. [Country](https://en.wikipedia.org/wiki/Country) Source: Unknown source World portal · Country (identity) · Lists by country · List of former sovereign states ... German Democratic Republic, Federal Republic of Germany (but Germany) ...
Results for tool call crawl_page with arguments 'url': 'https://en.wikipedia.org/wiki/List_of_former_sovereign_states', 'query': "Search the page for the exact string 'German Democratic Republic' and return the surrounding list context or the exact line indicating it is included.": No relevant information

Results for tool call web_search with arguments 'query': '"List of former sovereign states" "German Democratic Republic"': 1. [List of former sovereign states](https://en.wikipedia.org/wiki/List_of_former_sovereign_states) Source: Unknown source List of former sovereign states. Article · Talk. Language; Loading. . . Download ... German Democratic Republic (1949–1990) (also known as East Germany or GDR) ...

2. [List of sovereign states by date of formation](https://en.wikipedia.org/wiki/List_of_sovereign_states_by_date_of_formation) Source: Unknown source German Democratic Republic (commonly referred to at the time as East Germany) ... List of former sovereign states · List of national constitutions · List of ...

3. [List of former sovereign states](https://en-academic.com/dic.nsf/enwiki/11825029) Source: Unknown source List of former sovereign states. List of former sovereign states. This page ... the German Democratic Republic (East Germany) in 1990 joined the Federal ...

4. [Countries that No Longer Exist 2025](https://worldpopulationreview.com/country-rankings/countries-that-no-longer-exist) Source: Unknown source German Democratic Republic (East Germany), 1949, 1900, Germany. German Empire ... List of Former Sovereign States - Wiki. Other Categories.

5. [Former countries in Europe after 1815](https://www.wikiwand.com/en/articles/Former_countries_in_Europe_after_1815) Source: Unknown source German Democratic Republic ... See also. List of historic states of Germany · List of historic states of Italy. General. List of former sovereign states ...

# Note: Now, you should analyze the task completion status and provide recommendations for next steps.

## Step 10 - Assistant

<think>Let me analyze the completion status of the plan based on the conversation before.</think>
<summary>## Plan Summary

The original plan had four goals with multiple execution paths:
- Goal 1: Identify authoritative lists of Malko Competition recipients (post-1977) with nationalities
    - Path 1.1: Wikipedia + verification crawl
    - Path 1.2: Official competition/archival sources
    - Path 1.3: Reputable music databases and encyclopedias
    - Path 1.4: News/features aggregation

- Goal 2: Determine which recipient(s) after 1977 have nationalities tied to a country that no longer exists
    - Path 2.1: Cross-reference nationalities against lists of defunct countries    - Path 2.2: Historical sovereignty check for ambiguous cases    - Path 2.3: Edge-case validation (regions vs. countries)

- Goal 3: Confirm uniqueness (only one such recipient after 1977) and extract the first name
    - Path 3.1: Exhaustive post-1977 sweep and counting
    - Path 3.2: Independent corroboration of the unique candidate's nationality
    - Path 3.3: First-name extraction and consistency check

- Goal 4: Resolve discrepancies and document evidence trail
    - Path 4.1: Source prioritization and conflict resolution
    - Path 4.2: Evidence packaging for auditability

## Execution Status Analysis
### Goal 1: Identify authoritative lists of Malko Competition recipients (post-1977) with nationalities
- Status: Completed
- Path Analysis:
    - Path 1.1 (Wikipedia + verification crawl): Completed. Crawled the Wikipedia "Malko Competition" page and extracted the Recipients list post-1977 with implied nationalities/countries (e.g., United States, Venezuela, Taiwan, Japan, Chile, Germany).
    - Path 1.2 (Official competition/archival sources): Completed. Crawled the official Malko site "All Winners" page and extracted the winners list post-1977 with explicit country labels (e.g., "Germany" for 1983 Claus Peter Flor; "United States" for 2005 Mei-Ann Chen, 2009 Joshua Weilerstein, 2018 Ryan Bancroft).
    - Path 1.3 (Reputable music databases/encyclopedias): Partially touched. Found an encyclopedia.com entry confirming Claus Peter Flor's 1983 Malko win; not yet systematically applied to all names for nationality corroboration.
    - Path 1.4 (News/features aggregation): Not executed beyond incidental NYT hits; not required for core enumeration.

Result consolidation:
- Post-1977 winners and countries (from Wikipedia and official site):
    - 1980: Maximiano Valdés — Chile
    - 1983: Claus Peter Flor — Germany
    - 1986: Kazufumi Yamashita — Japan

    - 1989: Fabio Mechetti — Brazil
    - 1992: Jin Wang — Austria
    - 1995: Jan Wagner — Venezuela
    - 1998: Seikyo Kim — Japan
    - 2005: Mei-Ann Chen — United States
    - 2009: Joshua Weilerstein — United States
    - 2012: Rafael Payare — Venezuela
    - 2015: Tung-Chieh Chuang — Taiwan
    - 2018: Ryan Bancroft — United States
    - 2021: Dmitry Matvienko — Belarus
    - 2024: Samuel Seungwon Lee — South Korea

### Goal 2: Determine which recipient(s) after 1977 have nationalities tied to a country that no longer exists
- Status: In Progress (functionally resolved; documentation could be tightened)
- Path Analysis:
    - Path 2.1 (Cross-reference nationalities against lists of defunct countries): Partially completed. We confirmed East Germany (German Democratic Republic) ceased to exist in 1990 and that Claus Peter Flor was born in Leipzig (1953), i.e., in the GDR.

However, attempts to extract the exact "German Democratic Republic" line from the Wikipedia "List of former sovereign states" page via crawl returned no relevant snippet (likely due to dynamic rendering). We did obtain search snippets indicating its presence on that page.

    - Path 2.2 (Historical sovereignty check for ambiguous cases): Completed. Crawled the "East Germany" page confirming it was a sovereign state (1949–1990) that ceased to exist upon reunification.

    - Path 2.3 (Edge-case validation): Completed. Validated that Taiwan is not a sovereign state in the conventional sense and thus does not qualify; other countries (Chile, Japan, Brazil, Austria, Venezuela, United States, Belarus, South Korea) are extant.

- Interim conclusion:

    - Among post-1977 winners, only 1983 winner Claus Peter Flor's recorded nationality ("Germany") plausibly maps to a country that no longer exists in its sovereign form (the German Democratic Republic, East Germany). No other post-1977 winner's country of record is defunct.

Blocked/inefficient aspects and optimizations:

- The repeated attempts to extract the exact "German Democratic Republic" line from the "List of former sovereign states" page via crawl were inefficient and returned no content. Optimize by:

    - Using a static mirror (e.g., Wikipedia's mobile view or the PageContent handler) or an archived snapshot.

    - Querying an alternative reputable list (e.g., "Former countries in Europe after 1815") where the GDR line is easily extractable.

    - Searching for the exact string on a text proxy (e.g., r.jina.ai) to return the raw HTML snippet.

### Goal 3: Confirm uniqueness (only one such recipient after 1977) and extract the first name

- Status: Completed

- Path Analysis:

    - Path 3.1 (Exhaustive post-1977 sweep and counting): Completed. Enumerated all winners post-1977 and checked their countries; only 1983 (Claus Peter Flor) fits the defunct-country criterion when interpreted as East Germany/GDR.

    - Path 3.2 (Independent corroboration of the unique candidate's nationality): Completed. Wikipedia bio states "(born 16 March 1953, Leipzig) is a German conductor," and the official Malko winners page lists him under "Germany." Additional reputable mentions (NYT, encyclopedia.com) confirm his Malko win in 1983.

    - Path 3.3 (First-name extraction and consistency check): Completed. First name consistently "Claus" across sources.

Goal 3 result:

    - Unique qualifying recipient (post-1977): Claus Peter Flor.

    - First name to report: Claus.

### Goal 4: Resolve discrepancies and document evidence trail

- Status: In Progress

- Path Analysis:

    - Path 4.1 (Source prioritization and conflict resolution): Completed. Prioritized official Malko site and Wikipedia; resolved minor labeling differences (e.g., "Germany" vs. "East Germany") by historical confirmation that Leipzig 1953 = GDR.

    - Path 4.2 (Evidence packaging): Partially completed. We have:

      - Official winners list (malkocompetition.dk/winners/all) showing 1983 "Claus Peter Flor — Germany."

      - Wikipedia "Malko Competition" page listing 1983 Claus Peter Flor.

      - Wikipedia "East Germany" page confirming the GDR existed 1949–1990 and ceased to exist.

      - Wikipedia "Claus Peter Flor" page confirming birth in Leipzig (1953).

      - Search snippets indicating "German Democratic Republic" is on Wikipedia's "List of former sovereign states."

      Missing: a directly crawlable, citable line explicitly listing "German Democratic Republic" from a reputable "former countries" page.

Optimization for evidence packaging:

- Instead of repeatedly querying the standard Wikipedia page, fetch the mobile view or use the raw content endpoint:

    - Try: https://en.m.wikipedia.org/wiki/List_of_former_sovereign_states (then search for "German Democratic Republic").

    - Or: https://en.wikipedia.org/w/index.php?title=List_of_former_sovereign_states&action=render (then search for the string).

Dependencies affecting parallel execution:

- Goal 3 depended on Goal 2's determination that only 1983 fits the defunct-country criterion; this dependency is resolved.

- Goal 4's evidence packaging can proceed independently of the others but benefits from finalized conclusions of Goals 2 and 3.

## Next Parallel Sub-Paths

Based on the current execution status, the following sub-paths should be solved in parallel:

- Goal 2:

    - Sub-path: Capture a citable line that "German Democratic Republic (1949–1990)" appears on a reputable "former sovereign states" list.

      - Actions:

      - Crawl https://en.m.wikipedia.org/wiki/List_of_former_sovereign_states and extract the exact line containing "German Democratic Republic."

      - Alternatively, crawl https://en.wikipedia.org/wiki/Former_countries_in_Europe_after_1815 and extract the line "German Democratic Republic, 1949, 1990."

- Goal 4:

    - Sub-path: Compile a concise evidence bundle with verbatim quotes and URLs:     - Official winners entry: "1983 Claus Peter Flor — Germany."

      - East Germany page: "The GDR . . . ceased to exist . . . on 3 October 1990."

      - Claus Peter Flor bio: "born 16 March 1953, Leipzig."

      - Former states list: "German Democratic Republic (1949–1990)."

- Goal 1 (optional hardening, low priority):

    - Sub-path: Add one non-Wikipedia corroboration for at least one post-1977 winner's nationality (e.g., GRAMOPHONE or official orchestra bios for Mei-Ann Chen, Joshua Weilerstein) to further validate the "no defunct country" status for others.

Final answer readiness: - The unique qualifying recipient (post-1977) is 1983 winner Claus Peter Flor; his recorded nationality is Germany, corresponding to the former sovereign state German Democratic Republic (East Germany). Therefore, the first name is

"Claus." All goals necessary to support this conclusion are functionally complete; remaining work is purely on tightening citations for Goal 4.</summary>

### Step 11 - User

Based on the plan/summary and previous conversations, continue solving the task!

### Step 11 - Assistant

<think>We have four goals to advance in parallel. Goal 1 (authoritative lists) is already satisfied with Wikipedia and the official Malko site; no new tool calls are needed there. Goal 2 requires explicit verification that the country tied to the qualifying recipient is listed as a former sovereign state; our prior attempts to extract the exact line from the desktop Wikipedia page failed, so we'll use the mobile version and an alternative reputable list to capture a citable line. Goal 3 (uniqueness and first name) is effectively resolved, but we will maintain parallel progress toward Goal 4 (evidence packaging) by securing clean citations. Accordingly, in this step we will: (a) crawl the mobile Wikipedia 'List of former sovereign states' to extract the 'German Democratic Republic' line; (b) crawl the 'Former countries in Europe after 1815' page to extract the GDR entry as an alternative authoritative source; and (c) reconfirm the official Malko winners page entry for 1983 to have a clean verbatim line for evidence packaging. These calls are independent and can run in parallel.</think>
<tools>["name": "crawl_page", "arguments": "url": "https://en.m.wikipedia.org/wiki/List_of_former_sovereign_states", "query": "Find and return the exact line containing 'German Democratic Republic' (East Germany or GDR) to confirm it is listed among former sovereign states.", "name": "crawl_page", "arguments": "url": "https://en.wikipedia.org/wiki/Former_countries_in_Europe_after_1815", 'query': "Extract the line listing 'German Democratic Republic' with its years to confirm it is a former country.", "name": "crawl_page", "arguments": "url": "https://malkocompetition.dk/winners/all", "query": "Return the exact line for 1983 showing 'Claus Peter Flor' and the country label as displayed."]</tools>

### Step 12 - User

Results for tool call crawl_page with arguments 'url': 'https://en.m.wikipedia.org/wiki/List_of_former_sovereign_states', 'query': "Find and return the exact line containing 'German Democratic Republic' (East Germany or GDR) to confirm it is listed among former sovereign states.": No relevant information

Results for tool call crawl_page with arguments 'url': 'https://en.wikipedia.org/wiki/Former_countries_in_Europe_after_1815', 'query': "Extract the line listing 'German Democratic Republic' with its years to confirm it is a former country.": - German Democratic Republic — Establishment: 1949; Dissolution: 1990; Today part of: Germany. Note: Also known as East Germany; was the Soviet-controlled government of Germany after World War II.

Results for tool call crawl_page with arguments 'url': 'https://malkocompetition.dk/winners/all', 'query': "Return the exact line for 1983 showing 'Claus Peter Flor' and the country label as displayed.": 1983 Claus Peter Flor —————— Germany

### Step 12 - Assistant

<think>We have four goals to advance in parallel. Goal 1 (authoritative lists) is satisfied via the Wikipedia 'Malko Competition' page and the official Malko 'All Winners' page, both crawled and listing winners post-1977 with countries. Goal 2 (identify which recipient(s) after 1977 have nationalities tied to a defunct country) is resolved by confirming that the 1983 winner, Claus Peter Flor, was born in Leipzig (1953) and thus East German; East Germany (German Democratic Republic) is confirmed as a former sovereign state via the 'Former countries in Europe after 1815' page. Other post-1977 winners' countries (Chile, Japan, Brazil, Austria, Venezuela, Taiwan, United States, Belarus, South Korea) are extant. Goal 3 (confirm uniqueness and extract the first name) is satisfied: only 1983 fits the defunct-country criterion; first name is consistently 'Claus'. Goal 4 (evidence packaging) has sufficient citations: official winners entry (1983 Claus Peter Flor — Germany), Wikipedia East Germany page (ceased 1990), and the former countries list (German Democratic Republic 1949–1990). All goals are resolved; proceed to final answer.</think>
<tools>["name": "final_answer", "arguments": "answer": "Claus"]</tools>

### Final Answer

**Answer: Claus**

