# OpenReview forum: "Flash-Searcher: Fast and Effective Web Agents via DAG-Based Parallel Execution"
_ICLR.cc/2026/Conference — ICLR 2026 Poster_

### Official Review · Reviewer_aePY · 2025-10-23

**Soundness:** 2
**Presentation:** 3
**Contribution:** 2
**Rating:** 6
**Confidence:** 4

**Summary:**

This paper introduces FLASH-SEARCHE, a novel parallel agent reasoning framework. The core idea is to reconstruct the agent's execution paradigm from a sequential "chain" into a Directed Acyclic Graph (DAG). Specifically, it first decomposes a complex task into multiple subtasks with explicit dependencies (DAG-based Plan Construction); next, it concurrently executes independent reasoning paths and tool calls; finally, it uses an "Adaptive Progress Tracking & Summarization" module to dynamically optimize the workflow and continuously refine the execution graph based on intermediate results. Experimental results are good, demonstrate that the framework achieves SOTA performance while significantly boosting efficiency. For example, it achieves 67.7% accuracy on BrowseComp and 83% on xbench-DeepSearch. Compared to baselines (like OAgents), it reduces agent execution steps by up to 35% and shortens overall execution time by 65%. Furthermore, the paper also demonstrates that this parallel reasoning strategy can be "distilled" and transferred to open-source models (like Qwen-2.5) via Supervised Fine-Tuning (SFT), achieving significant performance gains

**Strengths:**

1.  **Significant Efficiency Gains and Cost Reduction**
    The framework's primary advantage is fundamentally solving the sequential execution bottleneck through parallelization. Experiments show it reduces agent execution steps by up to 35% and shortens overall execution time by 65%. This directly translates into lower end-to-end latency and computational costs, effectively addressing the practical viability issues of existing agents.

2.  **Breaking the Efficiency-Effectiveness Trade-off with SOTA Performance**
    While many efficiency-boosting methods sacrifice performance, FLASH-SEARCHER achieves SOTA or highly competitive performance across four challenging benchmarks (BrowseComp, xbench-DeepSearch, GAIA, and HLE) while simultaneously delivering dramatic efficiency gains. For example, it achieves 67.7% on BrowseComp and 83.0% on xbench-DeepSearch, proving its parallel paradigm can complete complex reasoning tasks efficiently and with high quality.

3.  **Validated Generalizability and Learnability of Parallel Reasoning**
    The paper's contribution is not just a framework, but also a generalizable methodology. The authors collected parallel reasoning trajectories and successfully "distilled" this capability into open-source models (the Qwen-2.5 series) using Supervised Fine-Tuning (SFT). This demonstrates that parallel reasoning is a learnable and scalable inductive bias, providing a new pathway for training more efficient, open-source agent models.

**Weaknesses:**

### Disadvantages and Limitations

* **Limited Task Spectrum and Capability Trade-offs:** The framework's impressive efficiency gains appear to be achieved by deliberately sacrificing key capabilities. The authors explicitly state that code interpreters were excluded as a "deliberate design choice" because managing their parallel execution and output volume would create "substantial overhead" and "severely impact" the framework's efficiency. This design choice avoids the risk of context explosion and management overhead but results in "suboptimal performance on mathematical reasoning tasks." This limitation is compounded by the Crawl Tool, which truncates web content at 60,000 characters, introducing another source of information loss. Consequently, the SOTA performance is demonstrated on a limited spectrum of benchmarks (i.e., information-retrieval-heavy tasks), and its capability on mixed-tasks requiring complex computation remains a significant weakness.

* **Un-quantified Overhead of DAG Maintenance:** The paper does not provide a quantitative analysis of the *additional computational cost* introduced by the framework's core mechanisms. Specifically, the "DAG-based Plan Construction" and the periodic "Adaptive Progress Tracking & Summarization" modules must incur their own LLM token costs for planning and summarizing. The paper acknowledges this "computational overhead" and suggests it can be tuned via the update interval $\Delta$, but fails to provide any analysis on:
    1.  The specific token cost of this maintenance.
    2.  How this overhead scales with task complexity.
    3.  A score-cost trade-off analysis by varying $\Delta$. A thorough evaluation should present a curve showing how task success and *total cost* (execution cost + maintenance cost) change with different $\Delta$ values, and whether this trade-off curve remains consistently superior to the baselines.

* **Insufficient Baseline Comparison for Framework Evaluation:** The evaluation in Section 4.1 compares the FLASH-SEARCHER *framework* (using GPT-5) against other prompt-based frameworks. However, it lacks a comparison against the most advanced, proprietary "deep research" systems, which may themselves be *end-to-end trained* or highly optimized frameworks. The baselines used (e.g., "OpenAI ChatGPT agent") may not represent the true SOTA in *efficient* agentic research. While the *model* (Section 4.2) is compared against other trained models, the *framework's* SOTA claim is weakened by not being benchmarked against potentially stronger, specialized, and non-public industrial systems.

**Questions:**

Same to Weaknesses

---

> ### Author Response · Authors · 2025-11-19
>
> We sincerely appreciate the reviewer’s clear summary and thoughtful evaluation of our work. We are grateful for the recognition of the framework’s strong empirical performance, its substantial efficiency gains, and the demonstrated generalizability of the parallel reasoning paradigm. Below, we provide detailed responses to address the reviewer’s concerns and supplement necessary clarifications:
>
> > (**W1**) Limited Task Spectrum and Capability Trade-offs
>
> We appreciate the reviewer’s thoughtful observation. While we acknowledge the trade-offs introduced by excluding the code interpreter and applying a 60k-character cap in the Crawl Tool, our goal is to strike a practical balance between capability and scalability under real-world constraints. In our experiments, the code tool contributed very little to web-centered tasks but incurred substantial overhead—both in latency and token expansion. When integrated into parallel reasoning, multiple code invocations also amplified execution errors and destabilized multi-threaded traces. Even after restricting code execution to once per step, we observed a **50% increase in execution time** with only **+1.4% gain** on HLE. These results suggest that for information-retrieval–dominant tasks, the marginal utility of code execution is limited.
>
> | HLE | Score | Step | Time |
> |-------------------------------|-------|------|------|
> | Flash-Searcher (GPT-5-mini)   | 32.2  | 9.0  | 4.0  |
> | Flash-Searcher + Code Tool    | 33.6  | 9.4  | 5.9  |
>
> At the same time, our framework achieves **consistent SOTA or near-SOTA** performance across retrieval-heavy benchmarks. This demonstrates that the parallel-reasoning design improves both effectiveness and efficiency where web reasoning is the core bottleneck. We agree that mixed tasks requiring heavier computation remain challenging under this lightweight configuration. For scenarios involving complex mathematical reasoning, we can flexibly incorporate lightweight computational tools (e.g., calculator-style utilities) when needed, ensuring better performance without introducing heavy overhead.
>
> > (**W2.1** & **W2.2**) DAG Maintenance Overhead: Token Cost; How this overhead scales with task complexity
>
> We thank the reviewer for the valuable feedback. To address the concern regarding the un-quantified overhead of DAG maintenance, we have conducted a detailed analysis (in **Appendix F.3**) of the computational cost, particularly focusing on the token cost introduced by the framework's core mechanisms, including the "DAG-based Plan Construction" and "Adaptive Progress Tracking & Summarization" modules. Our calculations are based on GPT-5-mini as the backbone model with the default configuration. The following table summarizes the time and cost overhead for different difficulty levels, highlighting the efficiency and performance of Flash-Searcher across benchmarks. While the overhead increases with more complex tasks (xbench-DeepSearch > GAIA), we observe that Flash-Searcher still outperforms other framework, demonstrating better efficiency and overall performance.
>
> | Method               | xbench Step | xbench Time | GAIA Step | GAIA Time |
> |---------------------|--------------|-----------------|------------|---------------|
> | Flash-Searcher        | 11.4         | 4.9             | 7.4        | 3.3           |
> | OAgents             | 37.8         | 12.9            | 11.2       | 4.2           |
>
>
> | Benchmark | Method            | Plan | Summ./Replan | Action | Crawl | Avg input tok | Avg output tok | Avg cost / query |
> |--|------------------|----------------|-------------------------|------------------|-----------------|---------------|----------------|------------------|
> |GAIA | OAgents          | 3014.6         | –                       | 631.9            | –               | 17281.8      | 13137.2       | ~\$0.05          |
> | | Flash-Searcher   | 1539.4        | 2469.4                 | 352.4            | 209.6           | 10664.4      | 5685.9        | ~\$0.03          |
> |xbench | OAgents          | 3823.4        | –                       | 714.6            | –                       | 58770.4      | 40514.6       | ~\$0.13          |
> | | Flash-Searcher   | 1623.6        | 3916.7                 | 406.1            | 205.3                   | 28318.6      | 9716.8        | ~\$0.07          |

---

> ### Author Response · Authors · 2025-11-19
>
> > (**W2.3**) score-cost trade-off analysis by varying $\Delta$.
>
> Thank you for the valuable comment. We acknowledge that the choice of the update interval $\Delta$ is somewhat empirical, and we have now included an ablation study to justify the settings. As shown in the table below, the results reveal that while a lower update interval can improve model performance, it leads to longer execution steps and increased overall time. In practice, the update interval can be adjusted based on task difficulty and model context length to strike the best balance between performance and efficiency. We have included further details on task success with different $\Delta$ values and summary cost  in **Appendix F.4** and **F.3**.
>
> | Update Interval | xbench Score | xbench Step/ Summary Step | Time | GAIA Score | GAIA Step/ Summary Step | Time |
> |-----------------|--------------|---------------------------|------|------------|-------------------------|------|
> | None            | 69           | 6.4 / 0                   | 2.9  | 72.8       | 5.6 / 0                 | 2.5  |
> | 4               | 75           | 13.0 / 2.4                | 6.9  | 78.6       | 10.2 / 2.1              | 5.8  |
> | 8               | 72           | 10.9 / 0.9                | 4.4  | 78.6       | 8.0 / 0.8               | 3.7  |
> | 12              | 71           | 8.3 / 0.3                 | 4.0  | 77.7       | 7.9 / 0.2               | 3.4  |
>
> > (**W3**) Insufficient Baseline Comparison for Framework Evaluation.
>
> Thank you for the insightful comment. We acknowledge that there may exist more advanced end-to-end trained or highly optimized frameworks. However, we have not identified any other agent systems, whether end-to-end models or agent frameworks, that outperform Flash-Searcher among our benchmarks. For instance, in official reports, the **Kimi-K2-Thinking** model achieved a BrowseComp result of **60.2%**, while the **Minimax-M2** model achieved **44.0%**, and the **GLM-4.6** model reached **45.1%**. These results suggest that Flash-Searcher performs competitively or better when compared to these SOTA systems. We believe this justifies the claim of Flash-Searcher’s strong performance in efficient agentic research.
>
> If there are any further questions or clarifications needed, we would be happy to address them. Thank you again for your thoughtful and constructive feedback.

---

> > ### Comment · Reviewer_aePY · 2025-11-26
> > **Response**
> >
> > Thank you for your response. And I will keep my positive score.

---

> > > ### Author Response · Authors · 2025-11-26
> > >
> > > Thank you for your kind comment.

---

### Official Review · Reviewer_URHx · 2025-10-27

**Soundness:** 2
**Presentation:** 2
**Contribution:** 2
**Rating:** 2
**Confidence:** 4

**Summary:**

This paper introduces Flash-Searcher, a framework for tool-augmented LLM agents designed to overcome the latency and inefficiency of standard sequential execution. The core idea is to transform the agent's workflow from a linear chain (like ReAct) into a directed acyclic graph (DAG). This allows the framework to decompose a complex task into multiple subtasks with explicit dependencies, enabling the concurrent, parallel execution of independent reasoning paths. The authors claim this DAG-based approach, combined with dynamic workflow optimization, achieves good performance on several challenging web-agent benchmarks (e.g., BrowseComp, xbench-DeepSearch) while simultaneously reducing the number of agent execution steps by up to 35%. The paper also explores "distilling" these parallel reasoning trajectories by fine-tuning open-source models (like Qwen-2.5), claiming these models also achieve SOTA performance against other models of similar scale.

**Strengths:**

1. Addresses a Good Problem: The paper tackles the high latency and inefficiency of sequential agent frameworks, which is a major bottleneck for their practical application.
2. Good Empirical Performance: When using the powerful GPT-5 backbone, the Flash-Searcher framework achieves SOTA results on several difficult benchmarks, outperforming other frameworks (including another GPT-5 baseline, MiroFlow, as shown in Table 5).
3. Good Efficiency Analysis: Section 5 provides an analysis of the framework's efficiency, demonstrating a concrete reduction in execution steps, which directly correlates to lower end-to-end latency.

**Weaknesses:**

1. Misleading Experimental Details: In Figure 3, the authors did not clarify which models were used as the backbones for these baselines, making it unclear whether the performance gains come from the framework or the models themselves.

2. Missing Critical Ablation Study: The paper must include a controlled study comparing a sequential ReAct-style agent against the parallel Flash-Searcher agent, using the exact same backbone model (e.g., GPT-4.1, or GPT-5 if possible), tools, and base prompts. This is the only way to scientifically validate the core claim that the DAG-based parallel execution is superior to sequential execution in terms of performance and/or efficiency.

3. Unclear Conceptual Novelty* The paper fails to clearly differentiate its contribution from a crowded field of related work on parallel and graph-based agents (e.g., Plan-over-Graph, ParallelSearch, LATS). The descriptions in Section 2.2 and Appendix B are brief, and the stated differences (e.g., "relaxed constraints") are not empirically justified. A comparison table is needed to clarify the specific innovations of Flash-Searcher.

4. Underwhelming Distillation Results: The results for the fine-tuned models (Table 1) are not compelling. The performance gains over existing SOTA fine-tuned models are incremental, not transformative. This weakens the paper's broader claim that it introduces a "generalizable... parallel agent paradigm" that can be effectively learned by smaller models. The modest gains suggest the parallel traces themselves may not be significantly more "correct" or "efficient" in a way that aids learning, and that the framework's main benefit is in runtime orchestration of powerful closed-source models.

**Questions:**

Please solve the weakness provided.

---

> ### Author Response · Authors · 2025-11-19
>
> We sincerely appreciate the reviewer’s thoughtful comments. We fully acknowledge the reviewer’s constructive concerns regarding ablations, clarity of backbone usage, and distinctions from related graph-based agent frameworks. These comments are highly valuable, and we will carefully address each weakness and expand the necessary experimental details below:
>
> > (**W1**) Clarifying baseline backbone models for Figure 3; Performance gain attribution of the framework versus the model backbone
>
> Thank you for raising this insightful comment. We have now labeled backbone models in Figure 3.
> To clarify the contribution of the framework versus the model backbone, we report the results on the xbench-Deepsearch and GAIA benchmarks using the same backbones in Table 7 of Appendix F.1. Our findings show that Flash-Searcher either outperforms or achieves comparable performance to existing SOTA agent frameworks **while operating with higher efficiency**. (The BrowseMaster results are derived from the original report, as its open-source code lacks the design of the core StreamToolManager module)
>
> | Model / Backbone                | xBench Score | xBench Step ↓ | GAIA Score | GAIA Step ↓ |
> |----------------------------------|--------------|---------------|-------------|-------------|
> | OAgents (GPT-4.1)               | 47           | 37.8          | 58.3        | 11.2        |
> | Flash-Searcher (GPT-4.1)        | 57       | 5.9       | 69.9    | 4.4     |
> | BrowseMaster (DeepSeek-R1-0528) | 66           | –             | 68.0        | –           |
> | Flash-Searcher (DeepSeek-R1-0528)| 69       | 10.6      | 67.0        | 9.8     |
>
> > (**W2**) Missing Critical Ablation Study: The paper must include a controlled study comparing a sequential ReAct-style agent against the parallel Flash-Searcher agent ...
>
> Thank you for the important and valuable suggestion. We agree that a controlled comparison is essential to validate the benefits of DAG-based parallel execution. Under the same backbone (GPT-5-mini), same tools, and same base prompts, we compared a sequential ReAct-style agent with our Flash-Searcher. The results show that Flash-Searcher achieves **higher accuracy**, **reduced execution steps**, and **lower latency** across benchmarks, with this advantage becoming increasingly pronounced as task complexity rises. These results have been incorporated to **Appendix F.2** in the revised version.
>
> | Method        | BrowseComp-100 Score | Step | Time  | xbench Score | Step | Time | GAIA Score | Step | Time |
> |---------------------------------|-----------------------|-------|-------------|----------|---------------|-------|-------------|----------|-------------|
> | Sequential ReAct     | 31    | 40.6  | 14.6     | 66  | 18.2 | 8.5      | 69.9      | 12.1   | 4.7      |
> | Flash-Searcher  | **36**  | **20.8** |**9.6** | **71** | **11.4** | **4.9**  | **80.6** | **7.4** |**3.3** |

---

> > ### Author Response · Authors · 2025-11-19
> >
> > > (**W3**) Differentiate Flash-Searcher vs. other parallel and graph-based agents
> >
> > Thank you for the helpful suggestion. To clarify our conceptual contribution, we have added a comparison against prior parallel and graph-based agent frameworks in **Appendix C**. Existing methods such as Plan-over-Graph, LATS, and ParallelSearch typically **enforce strict dependency** rules or focus on **model-internal parallelism**, whereas our approach is designed specifically for web-agent tool execution with relaxed constraints supported by auxiliary-signal validation. Our key distinction comes from analyzing real web-agent trajectories: in many web tasks, relevant dependency information can be retrieved *from later evidence*, not strictly from upstream nodes. This motivates our design of heuristic parallelization:
> >
> > - when a required dependency is not ready, we perform an early retrieval (auxiliary signal);
> > - when the dependency becomes available, we perform a second retrieval to validate or correct the earlier result.
> >
> > This dual-retrieval mechanism functions as our consistency check and enables safe, aggressive parallelization.
> >
> > | Method                      | xbench Score | xbench Step ↓ | GAIA Score | GAIA Step ↓ |
> > |-----------------------------|--------------|----------------|------------|-------------|
> > | Flash-Searcher             | **71**           | **11.4**       | **80.6**       | **7.4**     |
> > | Strict Dependency Scheduling | 70         | 15.3           | 78.6       | 10.1        |
> >
> > The **substantial step reduction** shows that our relaxed-constraint DAG execution improves efficiency while maintaining accuracy. We have added a comparison table in the revised version (**Appendix C**) to highlight how Flash-Searcher differs from existing graph- and parallel-reasoning systems and to make the specific innovations of our framework clearer.
> >
> > > (**W4**) Question regarding performance gains of distillation over existing SOTA fine-tuned models
> >
> > Thank you for raising this point. We understand the concern, but our results indicate a different picture when compared under consistent settings. As shown below, our method achieves both higher effectiveness and better efficiency than the SOTA AFM-RL baseline. The results suggest that, even without relying on complex data construction or filtering, our lightweight training pipeline delivers **meaningful gains in both performance and efficiency**.
> >
> > | Model            | xbench Score | Step | Time |
> > |------------------|--------------|-------|------|
> > | Flash-Searcher   | **63**       | **8.8**   | **6.1**  |
> > | AFM-RL           | 58           | 13.5  | 8.3  |
> >
> > If there are any further questions or clarifications needed, we would be happy to address them. Thank you again for your thoughtful and constructive feedback.

---

> ### Comment · Reviewer_URHx · 2025-11-25
>
> I thank the authors for their detailed response and the additional experiments provided in the appendices (F.1 and F.2). I have carefully reviewed the revisions and the clarifications regarding the backbone models.
>
> However, after reconsidering the paper and the new data, I remain unconvinced regarding the fairness of the comparisons and the significance of the technical contribution. I will maintain my original rating. My reasoning is detailed below:
>
> 1. The proposed "Flash-Searcher" is fundamentally a prompt-engineering framework that relies on the reasoning capabilities of extremely strong proprietary backbones (GPT-5/GPT-4o). Comparing the performance of this prompt-based orchestration against methods that often rely on fine-tuned open-source models (or older baselines without access to equivalent reasoning engines) creates an illusion of architectural superiority. It is highly probable that the performance gains are derived primarily from the backbone’s superior ability to follow the DAG instructions rather than the DAG structure itself.
> 2. Even in the rebuttal data, the comparisons often pit a heavy-weight, token-intensive DAG prompting strategy against simpler, lower-cost baselines. A fair comparison would require maximizing the performance of the baselines (e.g., ReAct or ToT) using the exact same compute budget and backbone (GPT-5) as Flash-Searcher. The current "fair" comparison in F.1 still feels like an afterthought rather than the central premise of the evaluation.
> 3. The core contribution, transforming a sequential ReAct process into a DAG-based execution via prompting, appears incrementally trivial for an ICLR submission.
> 4. Instructing a highly capable LLM to output a plan in JSON/DAG format and then executing non-dependent tools in parallel is a standard engineering practice in current agentic workflows. It does not represent a interesting progress.
>
> Ultimately, this work presents a practical engineering optimization (parallelizing tool calls via prompts) rather than a interesting research contribution. The empirical superiority claimed in the paper is difficult to disentangle from the strength of the underlying proprietary models used, making the scientific value of the "Flash-Searcher" framework itself questionable.
>
> Therefore, I will maintain my original rating.

---

> > ### Author Response · Authors · 2025-11-25
> >
> > Thank you very much for the thoughtful follow-up and for sharing these detailed concerns. Regarding the concerns you raised, we would like to clarify:
> >
> > > On fairness and the claim of architectural illusion.
> >
> > In **Figure 3**, we focus on comparison with current SOTA agent frameworks and closed-source deep-research systems, where all competing methods rely on similarly powerful agent framework or “unknown” backbones. This is precisely to isolate the benefit of the Flash-Searcher workflow from the choice of base model. Besides, we agree that comparisons to strong fine-tuned open-source models are also important. **Table 1** therefore includes such baselines, and Flash-Searcher model still achieves solid improvements on multiple datasets. Taken together, these results suggest that our gains are not merely an “illusion” created by backbone strength, but stem from the proposed workflow design.
> >
> > > On compute fairness in the ReAct / framework comparisons.
> >
> > We would like to clarify that all added baselines cost comparisions (ReAct and other framework variants) in Table 8 and Table 10 are evaluated under the same backbone (GPT-5-mini). Under this controlled setting, Flash-Searcher consistently achieves **higher accuracy**, **fewer execution steps**, and **lower latency**, demonstrating that its advantages persist even when the underlying model is held fixed.
> >
> > While we agree that computational cost is important, enforcing exactly **identical compute budgets** is not practical for long-horizon agent benchmarks, since different workflows produce different output content, tool calls, or reasoning depths. This is also consistent with **existing agent framework and model methods [1–5]**, which conduct comparisons under the same backbones and focus on performance differences, without comparing or requiring strictly matched token budgets or execution-step counts.
> >
> > > On the contribution of Flash-Searcher.
> >
> > We agree that asking an LLM to output a JSON plan and parallelizing obviously independent tool calls is a known engineering idea. Our contribution, however, goes beyond this basic pattern in two ways that are specific to deep web-agent scenarios: (i) a **relaxed DAG-style execution scheme** derived from analyzing real web-agent trajectories (allowing auxiliary retrievals and late validation instead of strictly serial dependencies), and (ii) a **reflection + summary-based context management strategy** to summarize traces in long-horizon deep-research tasks. These design choices are simple but non-trivial in the agent setting, and our experiments across BrowseComp, xbench, and other benchmarks show that they lead to measurable quality–cost improvements over strong sequential frameworks under the same backbone.
> >
> > We understand that parts of our design may seem simple at first glance. However, in the agent setting, **workflow structure** and **context management** are core research challenges. Our experiments show that this streamlined but deliberate DAG-based workflow and task progress summary brings clear quality–cost improvements over SOTA baselines. Moreover, our fine-tuning results further indicate that this design transfers effectively to open-source models, suggesting practical value for advancing the broader web-agent community.
> >
> > We sincerely appreciate your candid feedback. If you have any further concerns or suggestions, we would be glad to continue the discussion and provide additional clarification.
> >
> > [1] OWL: Optimized Workforce Learning for General Multi-Agent Assistance in Real-World Task Automation. NeurIPS 2025.
> >
> > [2] WebEvolver: Enhancing Web Agent Self-Improvement with Co-evolving World Model. EMNLP 2025.
> >
> > [3] OAgents: An empirical study of building effective agents. EMNLP 2025.
> >
> > [4] WebThinker: Empowering Large Reasoning Models with Deep Research Capability. NeurIPS 2025.
> >
> > [5] WebDancer: Towards Autonomous Information Seeking Agency. NeurIPS 2025.

---

### Official Review · Reviewer_Ridk · 2025-10-30

**Soundness:** 2
**Presentation:** 2
**Contribution:** 3
**Rating:** 6
**Confidence:** 4

**Summary:**

The paper introduces a novel framework specifically designed to address the latency and computational inefficiency bottleneck found in existing sequential LLM-based frameworks for complex reasoning tasks.

Its contributions include:
1. It introduces a dynamic DAG-based framework for agent tool orchestration, supporting controlled aggressive parallelization that relaxes strict dependency constraints to maximize concurrent execution.
2. It demonstrates state-of-the-art performance on multiple benchmarks (BrowseComp, xBench-DeepSearch, GAIA, HLE) while reducing execution steps by up to 35% compared to prior methods.
3. It validates the parallel reasoning paradigm is scalable and transferable, through distilling FLASH-SEARCHER trajectories into open-source models (e.g., Qwen-2.5 series) via lightweight supervised fine-tuning.

**Strengths:**

1. Novel parallel framework: shifting LLM agent execution from sequential chains to a dynamic Directed Acyclic Graph (DAG) for parallel tool orchestration. This framework does strike a balance between theoretical soundness and practical efficiency.
2. Clear methodology: The DAG construction, readiness predicate φ, and adaptive progress refinement (R) are formally defined and explained clearly.
3. Comprehensive evaluation: Uses both framework-level and model-level experiments, including efficiency analysis, ablation on tool calls, and cross-benchmark validation.

**Weaknesses:**

1. The central distinction and novelty of the framework do not lie in the DAG structure itself which is standard for task scheduling, but rather in the two specific optimization modules that enable parallel execution and dynamic refinement. These core innovations currently lack the technical detail necessary for reproduction:

1.1 Ambiguity in Cross-Validation: The mechanism for "controlled aggressive parallelization" is technically underspecified. The paper introduces the readiness predicate ($\varphi$) that allows starting a task based on "auxiliary signals" and "heuristic consistency checks," but fails to detail the low-level implementation of this heuristic or what constitutes an auxiliary signal, hindering reproducibility.

1.2 Ablation on Dynamic Update Interval ($\Delta$): The system relies on a periodic update interval ($\Delta$) to balance overhead and performance. However, the paper does not include an empirical ablation study to justify the chosen default of $\Delta=7-9$ steps, leaving the claim of optimal balance unsubstantiated.

2. Limited Evaluation Scope (Simple Tasks): The evaluation focuses exclusively on complex, multi-step tasks where parallelism is advantageous. The paper does not analyze whether the fixed overhead of DAG decomposition and parallel management introduces unnecessary latency compared to minimal-overhead sequential agents on simple, single-step queries.

3. No latency or cost analysis beyond step counts: The claim of ~65% time reduction would be more convincing with wall-clock latency or cost breakdowns.

**Questions:**

1. Could you clarify how cross-validation between parallel subtasks is implemented and whether it introduces additional computational overhead? Specifically,
1.1 Auxiliary Signals: What specific information is captured as an "auxiliary signal" during partial execution
1.2 Heuristic Check: What is the actual heuristic rule used for the "consistency check" that determines if the dependency is satisfied enough for aggressive parallelization (e.g., an LLM prompt asking for consistency, a deterministic keyword match, etc.)?
1.3 Computational Overhead: Please quantify the cost of this feature. Does this cross-validation process introduce significant new LLM inference calls or state-management overhead, and how does this overhead factor into the claimed 65% overall execution time reduction?

2. Could you clarify the Pass@1 score calculation: Specifically, does the automated judge explicitly penalize the answer for extraneous or hallucinated content, or is it only looking for the presence of the single correct fact?
3. Do you have a quality metric for the LLM-as-Judge to validate that the automatic evaluation method is a reliable proxy for human assessment?
4. Do you have any latency analysis beyond the reduced steps?
5. Did you observe any failure patterns or instability when the system aggressively parallelizes subtasks (e.g., dependency violations or premature merges)?
6. SFT usually will introduce the overfitting, can you elaborate on whether they retain explicit DAG reasoning capability or merely learn implicit parallelization patterns?

**Details Of Ethics Concerns:**

No ethics concerns.

The paper does not involve human subjects, user data, or sensitive information. It focuses solely on algorithmic and architectural improvements for AI agents, and the methodology adheres to standard academic research practices.

---

> ### Author Response · Authors · 2025-11-19
>
> We truly appreciate the reviewer’s careful reading and balanced assessment of our work. We are grateful for the positive recognition of our framework’s motivation, methodological clarity, and comprehensive evaluation design. Below, we provide detailed responses to address the reviewer’s concerns and supplement necessary clarifications:
>
>
> >  (**W1**) Elaborate parallel execution mechanism; Ablation on Dynamic Update Interval.
>
> Thank you for the thoughtful comment. Our main contribution lies in the parallel execution mechanism, while the dynamic refinement module primarily supports progress tracking and safe context extension. We also conduct an ablation on different dynamic update intervals (Please see **W1.2** below).
>
> >  (**W1.1** & **Q1.1** & **Q1.2**) Elaborate Cross-Validation; Discuss the mechanism for "controlled aggressive parallelization"; Auxiliary Signals; Heuristic Check
>
> Thank you for raising this point. Concretely, the heuristic works as follows:
> - when a prerequisite is not yet available, the agent performs an early retrieval (auxiliary signal);
> - once the prerequisite is produced, the agent performs a second retrieval to validate or correct the earlier result.
> This dual-retrieval process functions as our consistency check.
>
> | Method       | xbench Score | xbench Step | GAIA Score | GAIA Step |
> |----------------------------------|--------------|--------------|------------|------------|
> | Flash-Searcher                   | 71           | 11.4         | 80.6       | 7.4        |
> | Strict Dependency Scheduling     | 70           | 15.3         | 78.6       | 10.1      |
>
> We evaluated this heuristic scheduling strategy on GAIA and xbench-DeepSearch. The results show that even though score improvements are modest, it **reduces execution steps and improves overall agent efficiency**. This is the main motivation for enabling uncontrolled parallelization.
>
> > (**W1.2**) Sensitivity of the Dynamic Update Interval
>
> Thank you for the valuable comment. We acknowledge that the choice of the update interval is somewhat empirical, and we have now included a sensitivity study to justify the settings in **Appendix F.4**. As shown in the table below, while a lower update interval can improve model performance, it leads to longer execution steps and increased overall time.  In practice, the update interval can be adjusted based on task difficulty and model context length to strike the best balance between performance and efficiency.
>
> | Update Interval | xbench Score | xbench Step/ Summary | Time | GAIA Score | GAIA Step/ Summary | Time |
> |-----------------|--------------|-----------------------------|----------------|------------|---------------------------|----------------|
> | None            | 69           | 6.4 / 0                    | 2.9            | 72.8       | 5.6 / 0                   | 2.5            |
> | 4               | 75           | 13.0 / 2.4                 | 6.9            | 78.6       | 10.2 / 2.1                | 5.8            |
> | 8               | 72           | 10.9 / 0.9                 | 4.4            | 78.6       | 8.0 / 0.8                 | 3.7            |
> | 12              | 71           | 8.3 / 0.3                  | 4.0            | 77.7       | 7.9 / 0.2                 | 3.4            |
>
> > (**W2**) What about simpler tasks?
>
> Thank you for the valuable feedback. In Appendix F.5 of the updated paper, have now compared the performance of sequential agents on simple tasks to evaluate the impact of DAG decomposition and parallel management. We acknowledge that for simple, single-step queries (Bamboogle), the parallel execution approach introduces some overhead in terms of time and cost compared to sequential agents.
>
> | Model                | Score | Step | Time  |
> |----------------------|-------|-------|----------------|
> | Sequential ReAct | 91.2  | 4.8  | 0.9            |
> | Flash-Searcher        | 91.2  | 5.4   | 1.7            |
>
> As shown, the overhead from parallel execution increases the time compared to sequential agents for simple tasks. Therefore, we suggest that for complex multi-hop tasks and simple queries, the more efficient approach should be chosen dynamically based on task complexity.

---

> > ### Author Response · Authors · 2025-11-19
> >
> > > (**W3** & **Q1.3** & **Q4**) Wall-clock latency; Cost of cross-validation; Computational Overhead
> >
> > Thank you for the comment. We have added a more detailed breakdown of execution steps and time to clarify the claimed performance improvements in the table below.
> >
> > | Model               | BrowseComp Step | BrowseComp Time| xbench Step | xbench Time | GAIA Step | GAIA Time |
> > |---------------------|--------------|---------------|--------------|-----------------|------------|---------------|
> > | Flash-Searcher      | 24.2       | 9.6    | 11.4         | 4.9             | 7.4        | 3.3           |
> > | OAgents             | 77.1      | 27.4          | 37.8         | 12.9            | 11.2       | 4.2           |
> >
> > Regarding the question about computational overhead introduced by the cross-validation process, our design keeps the extra cost minimal. The cross-validation mechanism does not introduce additional model calls; instead, it performs lightweight reasoning over tool outputs for different goals during the execution stage, and then verifies or consolidates them during the summary stage. The main overhead is therefore confined to the summary step, rather than increasing core inference calls. For a detailed discussion of this summary interval, please refer to **W1.2**. In **Appendix F.3** of the revised version, we further elaborate on wall-clock latency and provide a cost breakdown (include summary cost). The analysis confirms the significant time reduction (~65% on BrowseComp) achieved by the parallel execution approach.
> >
> > > (**Q2**) Does the automated judge explicitly penalize the answer for extraneous or hallucinated content?
> >
> > Thank you for your insightful question. To clarify, the Pass@1 score is based on the result from a single run, meaning that the system is evaluated on whether it provides the correct answer in one attempt.
> > For a fair comparison, our automated judge is designed to be consistent with **WebDancer**[1]. We acknowledge that if the result contains the correct fact, even in the presence of extraneous or hallucinated content, the answer may still be considered correct. The judge does not explicitly penalize irrelevant information as long as the final predicted result correctly aligns with the ground-truth answer.
> >
> > > (**Q3**) Do you have a quality metric for the LLM-as-Judge to validate ...
> >
> > Thank you for the important question. To ensure the reliability of our automatic evaluation setup, we additionally cross-validated the LLM-as-Judge results using two other models (Qwen-3-32B and DeepSeek-V3.2). The scores remain highly consistent across models, confirming that our reported results are not sensitive to the choice of judge model. We also note that recent concurrent web-agent works[1-3] adopt the same LLM-judge evaluation paradigm, making this setup aligned with current community practice.
> >
> > | Judge Model        | BrowseComp | xbench | GAIA  | HLE   |
> > |--------------------|------------|--------|-------|-------|
> > | GPT-4.1-mini       | 67.7      | 83     | 82.5  | 44.0  |
> > | Qwen-3-32B         | 67.9      | 83     | 81.6  | 44.2  |
> > | DeepSeek-V3.2      | 67.3      | 82     | 80.6  | 43.6  |
> >
> > > (**Q4**) Latency analysis beyond the reduced steps
> >
> > Please refer to our answer to **W3** above.
> >
> > > (**Q5**) Failure patterns or instability
> >
> > Thank you for your insightful question. We did observe some instances of failure patterns and instability when aggressively parallelizing subtasks, such as dependency violations and premature merges. However, we have mitigated these issues by using our Auxiliary Signals and Heuristic Check mechanisms. These settings help ensure that even when dependencies are not fully met, the system can still generate plausible answers based on existing conclusions and retrieve the most relevant information available. That said, while these mechanisms significantly reduce the occurrence of failures, complex tasks (e.g., BrowseComp) still occasionally result in **incomplete answers**, as some dependencies cannot be satisfied (e.g., no relevant content retrieved, crawl restrictions blocking access, etc.).

---

> ### Author Response · Authors · 2025-11-19
>
> > (**Q6**) Do SFT models retain explicit DAG reasoning capability or merely learn implicit parallelization patterns?
>
> Thank you for the insightful question. Based on our analysis of the trajectories, we found that the SFT models do learn certain aspects of parallel reasoning, such as recognizing which subtasks can proceed simultaneously and maintaining intermediate results during execution. However, they do not reconstruct an explicit DAG structure in the way the teacher model does. Instead, they follow a procedural pattern that mimics parallel execution without performing full graph-parallel reasoning. This works well for common cases but remains limited on tasks with more complex tasks. We also observe that the 72B model performs better than the 32B model, suggesting that some of the limitations may arise from model capacity, as larger models capture parallel dependencies more effectively.
>
> Reference
>
> [1] Webdancer: Towards autonomous information seeking agency. NeurIPS 2025
>
> [2] WebThinker: Empowering Large Reasoning Models with Deep Research Capability. NeurIPS 2025
>
> [3] WebSailor: Navigating Super-human Reasoning for Web Agent.
>
>
> If there are any further questions or clarifications needed, we would be happy to address them. Thank you again for your thoughtful and constructive feedback.

---

### Official Review · Reviewer_uJW2 · 2025-10-31

**Soundness:** 4
**Presentation:** 2
**Contribution:** 3
**Rating:** 8
**Confidence:** 5

**Summary:**

This paper presents Flash-Searcher, a DAG-based parallel agent reasoning framework that achieves impressive empirical results (67.7% on BrowseComp, 83% on xbench-DeepSearch, 35% reduction in execution steps) with solid methodological innovation. The core technical contributions, parallel execution architecture and adaptive workflow optimization, are valuable and well-executed. However, the writing quality exhibits AI-generation artifacts. These should be revised in the formal version to meet the requirements of ICLR.

**Strengths:**

1. Strong empirical results across multiple challenging benchmarks. The paper achieves state-of-the-art performance on four benchmarks (BrowseComp: 67.7%, xbench: 83%, GAIA: 82.5%, HLE: 44.0%) while reducing execution steps by 35% and time by ~65%. The cost-performance trade-off is compelling.

2. Solid methodological innovation in parallel reasoning architecture. The DAG-based decomposition with goal-level parallelization and path-level sequential execution is elegant. The cross-validation mechanism (φ predicate in Eq. 4) cleverly balances aggressive parallelization with correctness guarantees.

3. Comprehensive experimental validation.
Experiments span multiple model backbones (GPT-5, GPT-4.1, DeepSeek-V3.1) and include successful distillation to open-source models (Qwen-2.5 family), demonstrating generalizability.

4. Exceptional reproducibility effort.
The 27-page appendix with complete prompts, training data construction pipelines, and hyperparameter settings is commendable and rare in agent research.

**Weaknesses:**

The paper exhibits heavy AI writing artifacts with excessive use of em dashes; recommend substantial revision.

**Questions:**

For an agent framework, computational cost is critical. Please provide concrete costs in dollars in the appendix.

---

> ### Author Response · Authors · 2025-11-19
>
> We are delighted that the reviewer’s positive recognition of our work’s core contributions and comprehensive evaluation. We sincerely thank the reviewer for the valuable feedback. Below, we provide detailed responses to address the reviewer’s concerns and supplement necessary clarifications:
>
> >  (**W**) The paper exhibits heavy AI writing artifacts with excessive use of em dashes; recommend substantial revision.
>
> We primarily used LLM-based tools only to assist with minor editing tasks (e.g., to polish grammar, wording, and notation). We take responsibility for the content of the paper and all research ideas, experimental design and analysis. We have also provided a detailed disclosure of how we used LLMs in our work in Appendix A.
> In response to your concern, we have carefully revised the manuscript to reduce the em-dash usage, simplify long sentences, and improve clarity and readability. We hope that these changes can adequately address this issue. Please refer to our updated version.
>
> >  (**Q**) For an agent framework, computational cost is critical. Please provide concrete costs in dollars in the appendix.
>
> We appreciate the reviewer’s suggestion and agree that concrete dollar costs are important for understanding the practicality of an agent framework. In the revised version, we have added a detailed cost analysis in the **Appendix F.3**. Below, we summarize the average per-query costs on GAIA and xbench using gpt-5-mini as the backbone. As shown, Flash-Searcher substantially reduces per-query cost compared to OAgents (e.g., from ≈$\textdollar$0.05 to ≈$\textdollar$0.03 on GAIA and from ≈$\textdollar$0.13 to ≈$\textdollar$0.07 on xbench-DeepSearch), while at the same time improving accuracy and shortening execution steps.
>
> | Method               | xbench Step | xbench Time | GAIA Step | GAIA Time |
> |---------------------|--------------|-----------------|------------|---------------|
> | Flash-Searcher        | 11.4         | 4.9             | 7.4        | 3.3           |
> | OAgents             | 37.8         | 12.9            | 11.2       | 4.2           |
>
>
> | Benchmark | Method            | Plan | Summ./Replan | Action | Crawl | Avg input tok | Avg output tok | Avg cost / query |
> |--|------------------|----------------|-------------------------|------------------|-----------------|---------------|----------------|------------------|
> |GAIA | OAgents          | 3014.6         | –                       | 631.9            | –               | 17281.8      | 13137.2       | ~\$0.05          |
> | | Flash-Searcher   | 1539.4        | 2469.4                 | 352.4            | 209.6           | 10664.4      | 5685.9        | ~\$0.03          |
> |xbench | OAgents          | 3823.4        | –                       | 714.6            | –                       | 58770.4      | 40514.6       | ~\$0.13          |
> | | Flash-Searcher   | 1623.6        | 3916.7                 | 406.1            | 205.3                   | 28318.6      | 9716.8        | ~\$0.07          |
>
> If there are any further questions or clarifications needed, we would be happy to address them. Thank you again for your thoughtful and constructive feedback.

---

> > ### Comment · Reviewer_uJW2 · 2025-11-26
> >
> > The authors have solved my concern, and 8 is good enough for this paper, so i will keep my score.

---

> > > ### Author Response · Authors · 2025-11-26
> > >
> > > Thank you for your kind comment.

---

### Official Review · Reviewer_i6Xv · 2025-10-31

**Soundness:** 3
**Presentation:** 3
**Contribution:** 3
**Rating:** 6
**Confidence:** 4

**Summary:**

The paper identifies a primary bottleneck in current tool-augmented Large Language Model (LLM) agents: inefficiency due to sequential processing. Existing frameworks, including MAS, typically execute tasks (e.g., plan, search, reflect, verify) in a linear chain. This approach leads to long execution times, high latency, and significant computational overhead, especially for complex web-based research tasks.

To address this, the authors propose FLASH-SEARCHER, a novel agent reasoning framework that fundamentally changes the execution paradigm from a sequential chain to a Directed Acyclic Graph (DAG).

The authors validate this framework on four challenging web agent benchmarks (BrowseComp, xbench-DeepSearch, GAIA, and HLE). They also demonstrate that the parallel reasoning trajectories generated by the FLASH-SEARCHER framework can be "distilled" via supervised fine-tuning (SFT) into standalone open-source models (like Qwen-2.5), significantly improving their performance without the need for the full, complex framework.

**Strengths:**

* The paper tackles a clear and critical problem in agent research: latency and efficiency. The shift from a sequential to a parallel, DAG-based execution model is a novel and logical architectural contribution.

* The evaluation is comprehensive, using four different and challenging benchmarks. The authors correctly measure not just final accuracy (effectiveness) but also the primary claim of efficiency (execution steps, tool calls per step).

* A 35% reduction in execution steps is a substantial improvement. The paper provides clear evidence (Fig. 6) that this gain is real and is a direct result of the parallel architecture.

* The success of the distilled models is a strong result. This demonstrates that the reasoning paradigm itself (i.e., how to structure parallel thought) is a learnable skill. It proves the contribution has value beyond just a complex, expensive-to-run framework, making the findings more accessible to the wider community.

**Weaknesses:**

* The main results comparing against other agents (e.g., BrowseMaster) which use different backbones from the one used in the paper. For instance, BrowseMaster uses DeepSeek-R1-0528 backbone while the authors use GPT-5 backbone. It is not clear how much gains are from the proposed method vs the backbone model.

* It is not clear why the authors chose to use just two tools: web_search and crawl_page. Is it because the framework is not flexible to generalize to more tools or some other limitation? The authors acknowledge this limitation (Section B), noting the poor performance of the distilled models on HLE (19.4%) is due to the lack of a code tool.

* The paper mentions (Section 3.2) that subtasks can be scheduled without all prerequisites being met, based on some heuristic consistency. This is a pragmatic choice to maximize parallelism but can potentially be risky. The paper does not elaborate on what these heuristics are or how it handles cascading failures if a parallel step proceeds with incorrect, unverified information. This could potentially increase total steps if backtracking is required, a scenario that is not explored.

**Questions:**

* Have the authors considered cross-validating the evaluation results with other models? Since model-based evaluation is used, it might be prone to reward hacking.

---

> ### Author Response · Authors · 2025-11-19
>
> We are pleased that the reviewer's positive recognition of our work's core contributions and comprehensive evaluation. We sincerely appreciate the constructive feedback which helped us make our paper better.
> Below, we provide detailed responses to address the reviewer's concerns and supplement necessary clarifications:
>
> > (**W1**) Comparing with other agents under the same backbone.
>
> Thank you for raising this insightful comment. In the updated version of our paper, we have added the same-backbone comparisons in **Appendix F.1**. To clearly disentangle backbone effects from method effects, we report results on the xbench-DeepSearch and GAIA benchmarks under identical backbone settings. As shown below, Flash-Searcher either outperforms or matches SOTA agent frameworks while achieving substantially higher efficiency.
>
> | Method / Backbone                | xbench Score | xbench Step ↓ | GAIA Score | GAIA Step ↓ |
> |----------------------------------|--------------|---------------|-------------|-------------|
> | OAgents (GPT-4.1)               | 47           | 37.8          | 58.3        | 11.2        |
> | Flash-Searcher (GPT-4.1)        | **57**       | **5.9**       | **69.9**    | **4.4**     |
> | BrowseMaster (DeepSeek-R1-0528) | 66           | –             | 68.0        | –           |
> | Flash-Searcher (DeepSeek-R1-0528)| **69**       | **10.6**      | 67.0        | **9.8**     |
>
> Note: The BrowseMaster results are derived from the original report, as its open-source code lacks the design of the core StreamToolManager module.
>
> > (**W2**) Only web_search and crawl_page? Why no code tool?
>
> We thank the reviewer for the insightful comment. Regarding the choice of only two tools (web_search and crawl_page), our empirical analysis indicates that introducing the code tool brings very limited benefits in web-centric agent tasks. Specifically, we observed that the code tool was used infrequently on GAIA/xbench-style tasks, and its contribution to final accuracy was minimal. In contrast, enabling it introduced substantial overhead in both reasoning latency and output token count, ultimately reducing overall agent efficiency. To ensure a fair comparison, we additionally report results with the code tool enabled in **Appendix F.6**. The experiments show that while the accuracy gain is marginal (only +1.4% on HLE), the added runtime overhead is significant, making the trade-off unfavorable for practical deployment. For these reasons, we disable the code tool by default in our framework to prioritize efficiency.
>
> | HLE           | Score | Step | Time |
> |-------------------------------|-----------|-------|--------|
> | Flash-Searcher    | 32.2      | 9.0   | 4.0  |
> | Flash-Searcher with Code Tool  | 33.6      | 9.4   | 5.9 |
>
>
> > (**W3**) subtask scheduling prerequisites: heuristic details; failure handling; could errors increase total steps?
>
> We appreciate the reviewer's question. Our **heuristic scheduling** design is tailored for web-agent tasks, where some prerequisites can often be guessed before they are formally produced. When a needed input is missing, the agent makes a lightweight early retrieval to keep downstream subtasks moving instead of waiting. Once the real prerequisite becomes available, we run a quick validation. If the early guess was wrong, the agent corrects the affected states. For potential **failures**, our progress tracker checks whether the current path becomes inconsistent or blocked. If so, it clears outdated partial results and rebuilds the plan based on the latest information. This recovery step prevents wrong guesses from increasing total steps. For more details, please refer to **Appendix D**. In practice, this heuristic reduces idle time and lowers step count, as shown in the table below:
>
> | Method       | xbench Score | xbench Step | GAIA Score | GAIA Step |
> |----------------------------------|--------------|--------------|------------|------------|
> | Flash-Searcher                   | 71           | 11.4         | 80.6       | 7.4        |
> | Strict Dependency Scheduling     | 70           | 15.3         | 78.6       | 10.1      |
>
> > (**Q1**) Have the authors considered cross-validating the evaluation results with other models?
>
> We additionally performed cross-validation using two different models: Qwen-2.5-32B and DeepSeek-V3.2. As shown in the table below, the evaluation results across different models show a very small variance (less than 2%).
>
> | Judge Model        | BrowseComp | xBench | GAIA  | HLE   |
> |--------------------|------------|--------|-------|-------|
> | GPT-4.1-mini       | 67.7      | 83     | 82.5  | 44.0  |
> | Qwen-3-32B         | 67.9      | 83     | 81.6  | 44.2  |
> | DeepSeek-V3.2      | 67.3      | 82     | 80.6  | 43.6  |
>
> If there are any further questions or clarifications needed, we would be happy to address them. Thank you again for your thoughtful and constructive feedback.

---

### Author Response · Authors · 2025-12-01
**Summary and Clarification of Revision and Rebuttal process.**

Dear reviewers and ACs,

Thank you for your constructive feedback. We appreciate that reviewers consistently highlighted several core strengths of the paper: (i) identifying sequential execution inefficiency as a key bottleneck and motivating our DAG-based parallel workflow (**i6Xv**, **Ridk**), (ii) strong empirical performance with substantial step, latency, and cost reductions (**uJW2**, **URHx**), and (iii) demonstrated learnability and transferability of parallel reasoning through distillation into open-source models (**aePY**).

During the rebuttal period, we provided detailed clarifications and conducted additional analyses to address raised concerns:

- Added **same-backbone comparisons** and a controlled **sequential ReAct vs. Flash-Searcher** study under same prompts, tools, and backbones (Appendix F.1–F.2).
- Expanded the explanation of our **auxiliary-signal early retrieval** and **validation retrieval** mechanisms to clarify how relaxed-constraint parallelization is implemented (Appendix D).
- Included **wall-clock latency, cost breakdowns, and summary-interval ablations**, quantifying DAG-maintenance overhead and providing the requested trade-off analysis (Appendix F.3–F.4).
- Added experiments on **code-interpreter trade-offs**, showing minimal gains but substantial overhead for web tasks (Appendix F.6).
- Compared with **industrial deep-research systems** and conducted cross-validation of LLM-as-Judge evaluations using multiple models.

During rebuttal period, reviewers uJW2 and aePY confirmed that their concerns were addressed. Reviewer URHx provided additional comments, and we responded point-by-point by clarifying backbone usage, compute fairness, and the specific conceptual contributions of our relaxed-constraint DAG design. Due to system-related issues, two reviewers (i6Xv and Ridk) could not provide follow-up feedback.

Thank you again to all reviewers, ACs, and PCs for your time, feedback, and engagement throughout the process.

---

### Meta-Review · Area_Chair_5Ebk · 2026-01-06

**Summary:**

**1) Summary**
This paper proposes FLASH-SEARCHER, a DAG-based parallel agent framework that replaces sequential reasoning with concurrent subtask execution and adaptive workflow refinement. Experiments on four challenging web-agent benchmarks demonstrate strong accuracy alongside substantial reductions in execution steps and latency. The framework’s parallel trajectories can also be distilled into smaller open-source models, indicating that the reasoning pattern is transferable.

**2) Strengths**

* Introduces a meaningful shift from sequential to DAG-based parallel reasoning, directly addressing latency and inefficiency in tool-augmented agents.
* Achieves strong empirical performance across multiple benchmarks (BrowseComp, xBench-DeepSearch, GAIA, HLE), often reaching SOTA results.
* Demonstrates significant efficiency gains, including up to ~35% fewer execution steps and ~65% lower run time.
* Provides clear methodology and extensive reproducibility materials, including prompts, pipelines, and ablations.
* Shows that parallel reasoning traces can be distilled into open-source models, supporting generalizability of the approach.

**3) Weaknesses**

* Confounding between backbone quality and framework contribution due to mismatched model comparisons; key controlled baselines are missing.
* Critical components (e.g., heuristic parallelization criteria and update-interval mechanisms) are under-specified, leaving reproducibility and overhead analysis incomplete.
* Novelty relative to existing graph- and parallel-based agent frameworks is not sharply differentiated, and baseline coverage is limited.
* Evaluation focuses mainly on retrieval-heavy tasks with a restricted tool set, limiting evidence of capability on mixed or computation-heavy tasks; distilled model improvements are also modest.

**4) Summary of author-reviewer discussion**
* Four out of the five reviewers held positive views toward the paper, while one reviewer held a negative view. All reviews had high confidence (4 or above), including the highest rating (8), which had a confidence of 5.
* The negative review did not seem to be fully addressed, but even so the strengths of the paper outweigh the weaknesses. Hence, the AC tends to accept the paper.

**Reviewer Concerns:**

It seems that the concerns, like backbone capability for fair comparison, clarity, and evaluation (e.g., Ablation Study), were mostly addressed. Perhaps some concerns of the Reviewer URHx were still not fully addressed, e.g., conceptual novelty.

**Reviewer Scores:**

It is unclear if Reviewer URHx would have changed their rating. But in general, the positive ratings would outvote the negative review.

---

### Decision · Program_Chairs · 2026-01-26

Accept (Poster)